# Neural G0: a quiescent-like state found in neuroepithelial-derived cells and glioma

Samantha A O'Connor[1,†], Heather M Feldman[2,†], Sonali Arora[2], Pia Hoellerbauer[2,3], Chad M Toledo[2,3], Philip Corrin[2], Lucas Carter[2], Megan Kufeld[2], Hamid Bolouri[2], Ryan Basom[4], Jeffrey Delrow[4] [iD], José L McFaline-Figueroa[5], Cole Trapnell[5], Steven M Pollard[6], Anoop Patel[2,7], Patrick J Paddison[2,3,*] & Christopher L Plaisier[1,**] [iD]

## Abstract

Single-cell RNA sequencing has emerged as a powerful tool for resolving cellular states associated with normal and maligned developmental processes. Here, we used scRNA-seq to examine the cell cycle states of expanding human neural stem cells (hNSCs). From these data, we constructed a cell cycle classifier that identifies traditional cell cycle phases and a putative quiescent-like state in neuroepithelial-derived cell types during mammalian neurogenesis and in gliomas. The Neural G0 markers are enriched with quiescent NSC genes and other neurodevelopmental markers found in non-dividing neural progenitors. Putative glioblastoma stem-like cells were significantly enriched in the Neural G0 cell population. Neural G0 cell populations and gene expression are significantly associated with less aggressive tumors and extended patient survival for gliomas. Genetic screens to identify modulators of Neural G0 revealed that knockout of genes associated with the Hippo/Yap and p53 pathways diminished Neural G0 in vitro, resulting in faster G1 transit, down-regulation of quiescence-associated markers, and loss of Neural G0 gene expression. Thus, Neural G0 represents a dynamic quiescent-like state found in neuroepithelial-derived cells and gliomas.

**Keywords** G0; glioma; neural stem cells; quiescence; scRNA-seq
**Subject Categories** Cancer; Chromatin, Transcription & Genomics; Neuroscience
**Mol Syst Biol. (2021) 17: e9522**

## Introduction

Most developing and adult tissues are hierarchically organized such that tissue growth and maintenance are driven by the production of lineage-committed cells from populations of tissue-resident stem and progenitor cells (Reya *et al*, 2001). Stem cells in adult tissues are typically found in a quiescent or reversible G0 state and must re-enter the cell cycle and divide to promote lineage commitment (Doetsch, 2003; Obernier *et al*, 2018). Stem cell progeny further balances lineage commitment with proliferation to produce adequate numbers of lineage-committed and terminally differentiated cells to keep pace with demand (Lin, 2008). While much is known about specific regulatory events governing organismal development and tissue homeostasis, we lack a detailed picture of how cells enter, maintain, and exit quiescent-like states.

Data from recent studies using single-cell analysis of specific developmental compartments have begun to unravel some of the mysteries around G0-like states, including hematopoiesis (Cabezas-Wallscheid *et al*, 2017; Hay *et al*, 2018), adult and fetal neurogenesis (Llorens-Bobadilla *et al*, 2015; Artegiani *et al*, 2017; Nowakowski *et al*, 2017), skeletal muscle regeneration (Scott *et al*, 2019), colon homeostasis (Grün *et al*, 2015), and a variety of other tissue types. The picture emerging from these studies indicates that in any given tissue, there is a continuum of highly regulated G0-like states in stem and progenitor cells and their progeny, which cause cells to enter long- or short-term states of quiescence (distinguishable from terminal differentiation/maturation states). For example, during adult mammalian neurogenesis, single-cell RNA-seq (scRNA-seq) analysis has led to a model where "dormant" quiescent neural stem cell (NSC) populations (e.g., in the subventricular zone or hippocampus) enter a "primed" state before entering the cell cycle and differentiating (Llorens-Bobadilla *et al*, 2015).

---

1  School of Biological and Health Systems Engineering, Arizona State University, Tempe, AZ, USA
2  Human Biology Division, Fred Hutchinson Cancer Research Center, Seattle, WA, USA
3  Molecular and Cellular Biology Program, University of Washington, Seattle, WA, USA
4  Genomics and Bioinformatics Shared Resources, Fred Hutchinson Cancer Research Center, Seattle, WA, USA
5  Department of Genome Sciences, University of Washington, Seattle, WA, USA
6  Edinburgh CRUK Cancer Research Centre, MRC Centre for Regenerative Medicine, The University of Edinburgh, Edinburgh, UK
7  Department of Neurosurgery, University of Washington, Seattle, WA, USA
   *Corresponding author. Tel: +1 206 667 4312; E-mail: paddison@fredhutch.org
   **Corresponding author. Tel: +1 480 965 6832; E-mail: plaisier@asu.edu
   †These authors contributed equally to this work

The application of scRNA-seq to dissociated primary glioma tumors has provided critical insights into intratumoral heterogeneity and developmental gene expression patterns for primary gliomas (Patel *et al*, 2014; Tirosh *et al*, 2016a; Darmanis *et al*, 2017; Venteicher *et al*, 2017; Filbin *et al*, 2018; Neftel *et al*, 2019). One key conclusion from these studies is that each tumor represents a complex, yet maligned, neurodevelopmental ecosystem that harbors diverse cell types which presumably contribute to tumor growth and homeostasis in specific ways (e.g., vascular mimicry, immune evasion, recreating NSC niches, and neural injury responses). However, these datasets have failed to produce models for transitions in and out of G0-like states. In contrast to NSC scRNA-seq studies where established cell-based markers are used to enrich for NSCs (e.g., GLAST$^+$/Prom1$^+$) (Llorens-Bobadilla *et al*, 2015), there are no pre-existing universal markers for glioblastoma (GBM) tumor cells that can neatly resolve subpopulations into quiescent, "primed", G1, or differentiated cellular states (Lathia *et al*, 2015). As a result, these studies generally create a catchall G1 category for cells with "low cell cycle index" or that lowly express S, G2, or M phase marker genes.

In addition, scRNA-seq cell cycle classifiers are not trained to identify G0-like populations. For example, the state-of-the-art cell cycle classifier from the Seurat scRNA-seq analysis pipeline (ccSeurat) by design only classifies cells into G1, S, and G2/M phases (Butler *et al*, 2018). ccSeurat was trained on a mouse embryonic stem cell (mESC) scRNA-seq dataset, where mESCs were Hoechst stained and sorted into G1, S, and G2/M populations and then subjected to scRNA-seq (Buettner *et al*, 2015; Scialdone *et al*, 2015). The ccSeurat cannot identify G0-like states because the training forced only these three states as the outcome, and mESCs do not transition into a natural state of quiescence.

Here, we performed scRNA-seq on *in vitro* grown human neural stem cells (hNSCs) derived from the developing mammalian telencephalon (Davis & Temple, 1994; Johe *et al*, 1996), which can recapitulate the expansion, specification, and maturation of each of the major cell types in the mammalian central nervous system (Pollard *et al*, 2006; Sun *et al*, 2008). We have previously used hNSCs as non-transformed, tissue-appropriate controls for functional genomic screens in patient-derived glioblastoma stem-like cells (GSCs)

(Danovi *et al*, 2013; Ding *et al*, 2013, 2017; Hubert *et al*, 2013; Toledo *et al*, 2014, 2015). We have observed that NSCs have longer doubling times of 40–50 h compared to 30–40 h for GSC isolates due to longer G0/G1 transit times. NSC scRNA-seq analysis led to the discovery of a transient Neural G0 subpopulation, which self-renewing NSCs pass in and out of and is enriched for genes expressed in quiescent NSCs and a broader set of neurodevelopment markers expressed in other neural progenitors and cell types poised for cell division. We constructed a classifier, which we apply to neurodevelopment and glioma patient data to determine the functional impact of this cell subpopulation. Finally, we identify genes that when perturbed diminish this G0-like state. Thus, our results reveal Neural G0 as a cellular state associated with quiescence in neuroepithelial-derived cell types.

## Results

### Identification of cell cycle phases and candidate G0/G1 subpopulations in hNSCs

We profiled 5,973 actively dividing U5-hNSCs (Bressan *et al*, 2017) using scRNA-seq to identify the single-cell gene expression states corresponding to cell cycle phases with a focus on G0/G1 subpopulations. Unsupervised cell clustering identified eight distinct clusters of cells (Fig 1A). A small cluster of cells that had significantly lower RNA levels was excluded; it was later included as an outgroup for classifier construction (i.e., "G1/other"; Appendix Fig S1D). Meaning was attributed to the remaining seven clusters based on (i) analyzing the set of marker genes significantly over-expressed within each cluster (avg log fold-change ≥ 0.3; adjusted *P*-value ≤ 0.05; Dataset EV1); (ii) comparison with predicted cell cycle phases from the ccSeurat single-cell classification method; (iii) distribution of single-cell cyclin and CDK expression across each cell cycle phase; and (iv) RNA velocity predicted progression of cells along the cell cycle phases. Through these comparisons, described below, we labeled the clusters based on the Whitfield *et al*, 2002 convention as follows: Neural G0 (17.3% of cells), G1 (36.7%), Late G1 (6.4%), S (7.2%), S/G2 (10.9%), G2/M (10.6%), and M/Early

---

**Figure 1. Gene expression map of cell cycle and candidate G0 and G1 subpopulations using single-cell RNA-seq in U5-hNSCs.** ▶

A  Eight transcriptional clusters were discovered from unsorted U5-hNSCs using an unsupervised graph-based clustering method. Single cells were embedded into a two-dimensional space for visualization using t-Distributed Stochastic Neighbor Embedding (tSNE).

B  The ccSeurat classifier was applied to the unsorted U5-hNSCs. The G1, S, and G2M phase calls were overlaid onto the tSNE embedding.

C  The number of cells for each U5-hNSC defined cell cycle phase colorized by the three ccSeurat classifier phases. Names for new U5-hNSC defined cell cycle phases are red.

D  RNA velocity stream plot for U5-hNSCs shows the directional flow of cells through the phases of the cell cycle (i.e., G1 → Late G1 → S → S/G2 → G2/M → M/Early G1 → G1) and into the novel Neural G0 phase.

E  Heat map of the relative expression (row-wise z-score) for the top 10 non-redundant genes for each prominent cluster in U5-NSCs and gene ontology analysis of the up-regulated genes defining each cluster.

F  Top marker genes for each cluster.

G  Cyclin and CDK marker genes found for each cluster.

H  Functional GO term enrichment for key cell cycle-related and "glial cell differentiation" terms. Full cluster-defining gene list is in Dataset EV1, and full gene ontology is in Dataset EV2.

I  Enrichment of knockdown cell cycle arrest phenotype genes for each cluster.

J  Ridge graph comparisons of cyclin expression across the U5-hNSC defined cell cycle phases. The x-axis is relative expression of the cyclin, and the y-axis is counts of cells per phase. The cell cycle phase with the peak expression is denoted by an arrowhead at the top of the plot.

K  Ridge graph comparison of cyclin expression across the Seurat-classified cell cycle phases. The x-axis is relative expression of the cyclin, and the y-axis is counts of cells per phase.

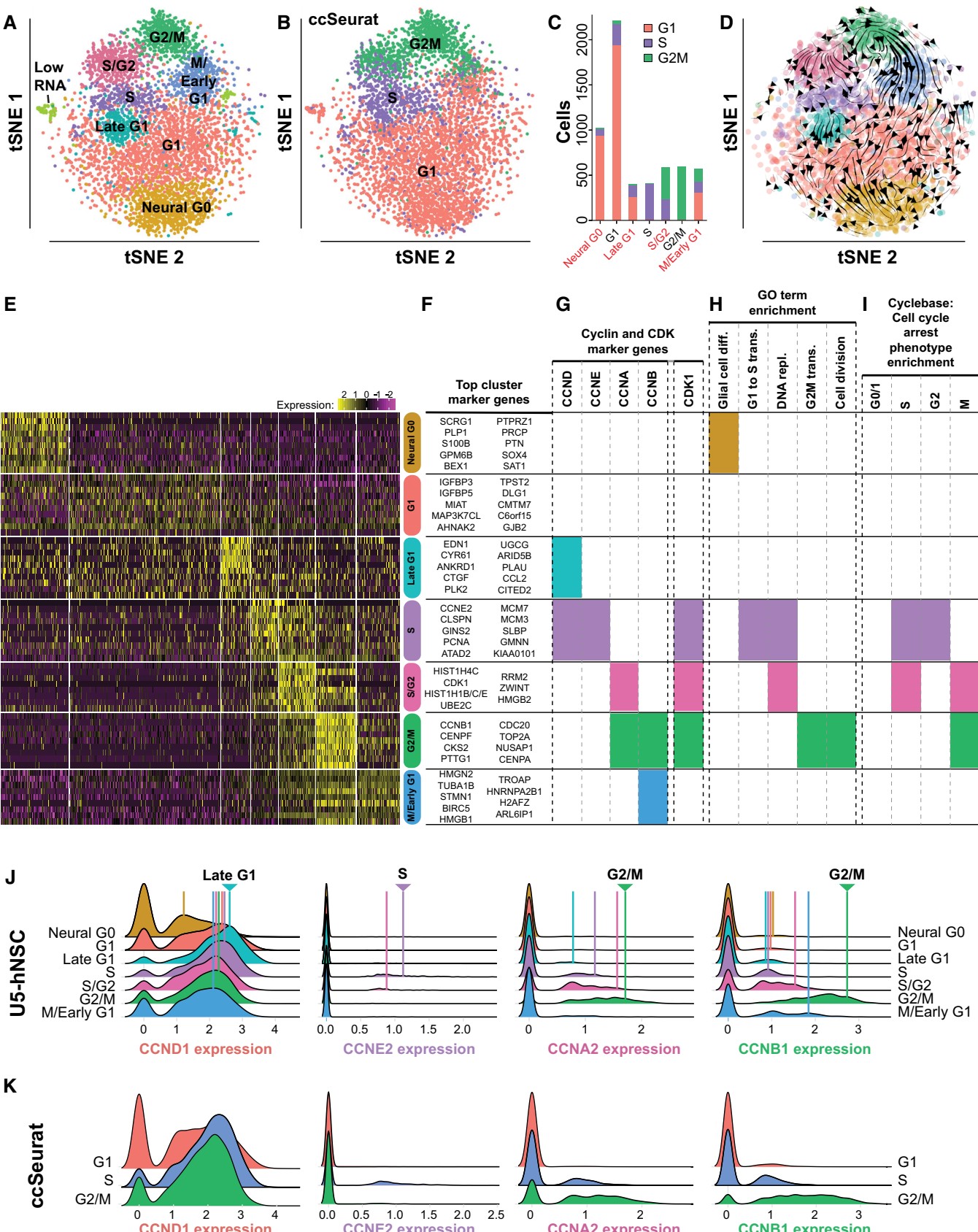

**Figure 1.**

G1 (8.4%) (Fig 1A). Neural G0 and Late G1 are novel states that were observed in U5-hNSCs.

## Characterizing U5-hNSC cell cycle phases

We assigned a cell cycle phase to the seven U5-hNSC clusters by analyzing the marker genes (Fig 1F, Appendix Fig S2), cyclin and CDK expression (Fig 1G, J and K), GO term functional enrichment (Fig 1H; Dataset EV2), and enrichment of genes associated with arrest in specific cell cycle phases (Fig 1I) (Santos *et al*, 2015). Cyclin expression patterns were consistent with prior knowledge (Appendix Fig S1A) where *CCND1* is a marker gene for the Late G1 and S clusters, *CCNE2* for the S cluster only, *CCNA2* for the S/G2 and G2/M clusters, *CCNA1* and *CCNB1* for the G2/M cluster only, and *CCNB2* for the G2/M and M/Early G1 clusters (Fig 1G and J). In addition, the cyclin-dependent kinase *CDK1* is a marker gene for the S, S/G2, and G2/M phases. The cyclin and *CDK1* expression patterns are highly consistent with the expected cell cycle expression pattern (Fig 1G and J) (Darzynkiewicz *et al*, 1996). Functional enrichment analysis of each cluster's marker genes linked Neural G0 with "glial cell differentiation", S with "G1 to S transition", S and S/G2 with "DNA replication", and G2/M with "G2M transition" and "cell division" (Fig 1H; Dataset EV2). Gene knockdowns that arrest cells in S and G2 phases were enriched in S cluster marker genes, arrest in S and M phases were enriched in S/G2 cluster marker genes, and arrest in M phase were enriched in G2/M cluster marker genes (Fig 1I; Dataset EV3) (Santos *et al*, 2015). The accumulation of these sources of evidence strongly supports the cell cycle identities we have attributed to the seven cell clusters observed in actively dividing U5-hNSC cells.

Four of the seven clusters were related to G0/G1: G1, Late G1, M/Early G1, and Neural G0. Despite being the largest cluster, the G1 cluster had the smallest number of enriched genes, which included IGFR1 signaling genes (e.g., *IGFBP3* and *IGFBP5*) and had significant reductions of genes expressed in S, S/G2, and G2/M clusters (Fig 1E). The Late G1 cluster was defined by genes important in G1 cell cycle progression, including *CCND1* and *MYC*, and was enriched for cholesterol biosynthesis genes, cell adhesion genes, and a subset of YAP target genes, such as *CTGF*, *CYRG1*, and *SERPINE1* (Fig 1E; Appendix Fig S1B; Datasets EV1 and EV2). The M/Early G1 cluster showed low but significant residual expression of M phase genes and enrichment for splicing factor genes, which could represent residual mRNA from G2/M (Fig 1E; Dataset EV1).

The Neural G0 cluster showed significant repression of 246 genes peaking in other phases of the cell cycle, including suppression of *CCND1* expression, which is an indicator of cell cycle exit (Sherr, 1995), and other cell cycle-regulated genes such as *AURKB*, *CCNB1/2*, *CDC20*, *CDK1*, and *MKI67*. Moreover, the 158 up-regulated genes defining this cluster were genes with key roles in neural development, including glial cell differentiation, neurogenesis, neuron differentiation, and oligodendrocyte differentiation (Fig 1E; Dataset EV1). These genes included transcription factors with known roles in balancing stem cell identity and differentiation, including *BEX1*, *HEY1*, *HOPX*, *OLIG2*, *SOX2*, *SOX4*, and *SOX9* (Sakamoto *et al*, 2003; Bergsland *et al*, 2006; Scott *et al*, 2010) (Fig 1E; Dataset EV1).

We further characterized the U5-hNSC cell cycle phases by comparison with ccSeurat (Butler *et al*, 2018). The current ccSeurat classifier assigns cells into the G1, S, and G2/M phases. When

ccSeurat is applied to the hNSC scRNA-seq data, the G1 phase cells match with the U5-hNSC phases Neural G0, G1, and M/Early G1 (Fig 1B and C). The ccSeurat M phase matches to S/G2, G2/M, and M/Early G1 (Fig 1B and C). The ccSeurat S phase most strongly matches to S, but also matches to all the other phases except for G2M. Overall, there is good agreement between the U5-hNSC and ccSeurat cell cycle phases when comparing only the cells labeled as G1, S, or G2/M in the U5-hNSCs (accuracy = 90%; Fig 1B).

Closer examination of cyclin expression across the U5-hNSC and ccSeurat cell cycle phases reveals that the subdivision of the G1 ccSeurat phase into Neural G0, G1, Late G1, and M/Early G1 phases is meaningful (Fig 1J and K). The novel Late G1 phase has the highest peak expression of *CCND1* (Fig 1J), which is consistent with prior studies that showed *CCND1* protein peaks just prior to entry into S phase (Matsushime *et al*, 1994). The Neural G0 subpopulation has the lowest peak *CCND1* gene expression (Fig 1J), a hallmark of quiescence (Sherr, 1995). ccSeurat does not capture that information as it lumps together high, medium, and low *CCND1* expressing cells (Fig 1K). In addition, the U5-hNSC cell cycle phases better stratify *CCNA2* and *CCNB1* expression into more discrete expressing subpopulations (high, medium, and low) across S, S/G2, G2/M, and M/Early G1, further demonstrating that these phases are distinct (Fig 1J). The U5-hNSC cell cycle phases highly overlap with ccSeurat cell cycle phases, and the U5-hNSC cell cycle phases outperform ccSeurat by classifying cells into more specific cell cycle phases which better capture the real biology of the cell cycle as demonstrated through meaningful changes in cyclin expression between cell cycle phases.

## Resolving the flow of cells through the cell cycle using RNA velocity

We resolved the possible trajectories between the seven distinct cell cycle phases through statistical assessment of the similarity between gene expression mediods for each cell cycle phase. The mediods are gene expression vectors where each gene holds the average expression across all cells from a cell cycle phase. First, the vectors were compared using correlation. Then, a distance matrix was constructed using Canberra distance. Finally, a cutoff was used to determine the edges of the resulting network. The resulting pattern from this network fits well with cell cycle progression and predicted transit through G0/G1 (Appendix Fig S1C). Cells from the candidate G0 population were linked solely to the G1 cluster, which is consistent with G0 as a cell cycle exit from G1. The linkages between clusters are not directed, and therefore, the cells may pass in either direction.

We added directionality to the edges using RNA velocity, which computes the ratio of unspliced to spliced transcripts and infers the likely trajectory of cells through a two-dimensional single-cell embedding, e.g., tSNE. The RNA velocity trajectories delineate the cell cycle in the expected orientation (Fig 1D), i.e., G1 → Late G1 → S → S/G2 → G2/M → M/Early G1 → G1. Both the similarity analysis and RNA velocity predict that cells enter Neural G0 through G1 (G1 → Neural G0). A reverse trajectory from Neural G0 to G1 was not apparent, although we demonstrate later that Neural G0 cells re-enter the cell cycle after transient stays of variable length. This model of cell cycle progression was further validated by total mRNA expression levels for the cell subpopulations (total UMI counts per cell). The total mRNA expression level was low in Neural G0 and peaks in G2/M (Appendix Fig S1D), which is consistent with prior

observations that total mRNA levels peak with the expression of cyclin B (*CCNB1/2*) and other mitotic genes (Shapiro, 1981). Through unbiased means, we reconstructed the cell cycle progression and identified gene expression signatures that can be used to track a cell's progress through the cell cycle.

**Constructing a cell cycle classifier from actively dividing hNSCs**

The ability to accurately assign a cell cycle phase based on a transcriptome profile has many potential uses in single-cell studies and beyond. We used the hNSC scRNA-seq data to build a cell cycle classifier. We tested four different methods which were previously found to be useful for building classifiers from scRNA-seq profiles (Abdelaal *et al*, 2019): (i) support vector machine with rejection (SVMrej), (ii) random forest (RF), (iii) scRNA-seq optimized K-nearest neighbors (KNN) (Wolf *et al*, 2018), and (iv) scRNA-seq optimized Neural Network (NN) method ACTINN (Ma & Pellegrini, 2020). We selected the 1,536 most highly variable genes in the U5-hNSC scRNA-seq profiles as the training dataset for the classifier. We applied 100-fold cross-validation (CV) for each classifier method and determined that the NN method ACTINN was statistically similar or slightly better at predicting each cell cycle phase than the next best classifier and had a significantly higher F1 score for Late G1 (*P*-value $\leq 4.3 \times 10^{-64}$, Fig 2A). The ACTINN classifier had an overall error rate of 18.4% in the CV studies, which was the best of all the methods tested. The ACTINN classifier was named ccAF for cell cycle ASU/Fred Hutch.

A significant issue in scRNA-seq studies is that the number of genes detected depends on sequencing depth, and missing gene information is commonplace. Therefore, we conducted a sensitivity analysis to determine the effect of randomly removing an increasing percentage of genes. We show that removing 40% of the classifier genes causes an increase of only ~10% in the error rate (Appendix Fig S3). The classifier is quite robust to even a large percentage of missing genes in query datasets, which provides a useful sensitivity analysis that informs future users of the ccAF classifier.

**Validating ccAF S and M phase cell cycle classifications**

Next, we validated S and M phase classifications by applying the ccAF classifier to a gold standard cell cycle synchronized time-series dataset from HeLa cells (Whitfield *et al*, 2002). The synchronized

HeLa cell study simultaneously characterized transcriptome profiles and experimentally determined whether the cells were in S or M phase at each time point (Whitfield *et al*, 2002). The ccAF classifier had an error rate of 13.7% when applied to the gold standard Whitfield *et al*, 2002 dataset (Appendix Fig S4), which demonstrated that ccAF could accurately predict the S and M phases for each query transcriptome profile (single-cell or bulk RNA-seq/microarray).

**Validating G0/G1 cell cycle classifications**

We validated the G0/G1 phase classifications by experimentally determining which cells from the U5-hNSCs belonged to the G0/G1 subpopulations. We used the well-established fluorescent ubiquitination-based cell cycle indicator (FUCCI) (Sakaue-Sawano *et al*, 2008) coupled with flow cytometry to enrich the CDT$^+$ G0/G1 cell subpopulations. The enriched G0/G1 subpopulations were then quantified using scRNA-seq, and the cell cycle phase of each cell was classified using ccAF. The U5-hNSC Neural G0 and G1 subpopulations were enriched in the CDT$^+$ subpopulation (log2(FC) > 0; Fig 2B), whereas the U5-NSC Late G1, S/G2, and G2/M subpopulations were all significantly depleted (log2(FC) $\leq -1$; Fig 2B). We experimentally demonstrated that we have correctly defined the G0/G1 subpopulations using the well-established FUCCI system. Importantly, the Neural G0 population is enriched when sorting for CDT$^+$ cells, which validates that this subpopulation is a part of the G0/G1 pool of cells.

Next, we evaluated whether the Neural G0 subpopulation from *in vitro* U5-hNSCs was similar to the quiescent NSCs (qNSCs) from two independent *in vivo* scRNA-seq profiling studies of NSCs from adult rodent neurogenesis in the subventricular zone (Llorens-Bobadilla *et al*, 2015; Dulken *et al*, 2017). In both studies, a majority of the qNSC cells were classified as Neural G0 by ccAF. One hundred percent of the dormant state qNSC1 from Llorens-Bobadilla *et al*, 2015 classified as Neural G0, and 96% of the primed-quiescent state qNSC2 classified as Neural G0 (Fig 2C). The non-mitotic activated NSCs (aNSC1) state cells were primarily classified as Neural G0, G1, Late G1, and M/Early G1, whereas the mitotic aNSC2 state cells classified as S, S/G2, and G2M (Fig 2C). The enrichment of Neural G0 in quiescent neural stem cells was validated in a second independent cohort from Dulken *et al*, 2017, where 64% of the qNSC state classified as Neural G0 and 88% classified as Neural G0,

**Figure 2. Application of the ccAF classifier to neuroepithelial-derived cell populations.**

A  Comparison of four different classifier methods (SVMrej, RF, KNN, and ACTINN) by F1 score, which is a metric that integrates precision and recall and reaches its maximum value at 1. An F1 score is computed for each cell cycle phase to be predicted, and the boxplots represent the distribution of F1 scores from the 100-fold cross-validation with a hold-out of 1,000 cells. Each boxplot shows the median (middle band), interquartile range (box), and the whiskers denote 1.5 times the interquartile range of the 100 F1 scores from the cross-validation.

B  Top, percent of cells found in each cluster for the U5-hNSCs. Middle, mapping of FUCCI reporter system to cell cycle phases. Bottom, fold-change between U5-hNSCs sorted for CDT$^+$ compared to unsorted U5-hNSC cells on the log base 2 scale. Positive values indicate an increase in a given cell cycle subpopulation in CDT$^+$ sorted relative to unsorted, and negative values indicate reduced cell subpopulations. The expected CDT$^+$ and CDT$^-$ cell subpopulations are found below the red bar and green bar, respectively. The others are expected to be transition subpopulations.

C  Percent of cells assigned to each ccAF cell cycle phase for scRNA-seq data from GLAST and PROM1 flow-sorted cells from the subventricular zone (SVZ) of mice (Llorens-Bobadilla *et al*, 2015). qNSC1 = dormant quiescent neural stem cell; qNSC2 = primed-quiescent neural stem cell; aNSC1 = active neural stem cell; aNSC2 = actively dividing neural stem cell.

D  Percent of cells assigned to each ccAF cell cycle phase for scRNA-seq data from EGFR, GFAP, and PROM1 flow-sorted cells from the subventricular zone (SVZ) of adult mice (Dulken *et al*, 2017). qNSC = quiescent neural stem cell; aNSC = active neural stem cell.

E  Percent of cells assigned to each ccAF cell cycle phase for scRNA-seq data from the developing human telencephalon (Nowakowski *et al*, 2017). Cell types were grouped by glial, neuronal, and vascular developmental cell lineages and ordered in each group by most to least differentiated (left to right). RG = radial glia, div = dividing. All cell type abbreviations are available in Dataset EV4.

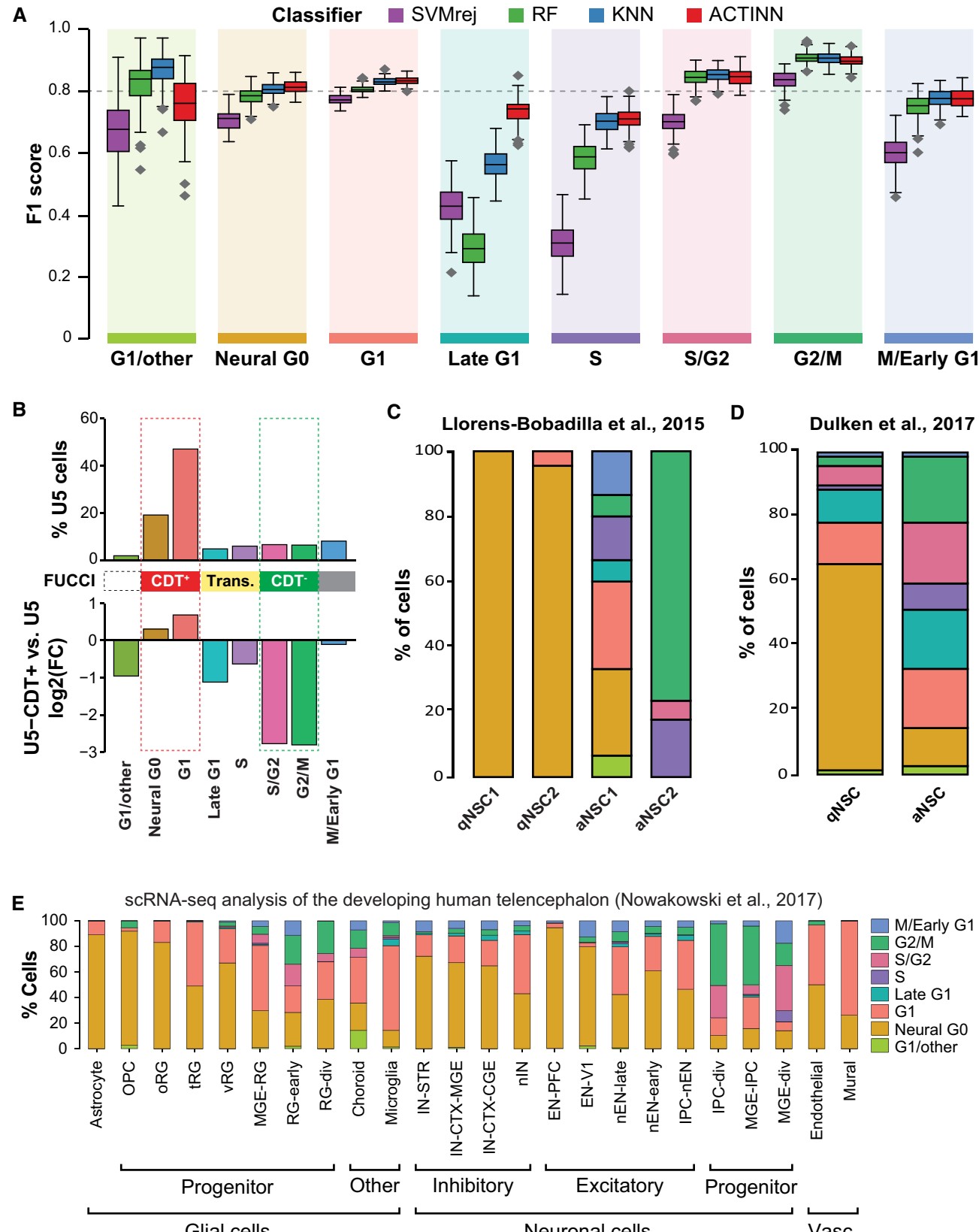

**Figure 2.**

G1, Late G1, or M/Early G1 (Fig 2D). These results validate that the Neural G0 subpopulation from *in vitro* U5-hNSCs is similar to the quiescent NSC subpopulation *in vivo*. Additionally, this validates that the ccAF classifier can accurately identify quiescent NSCs as Neural G0, and is robust enough to be applied across species using gene homology.

## The Neural G0 state is enriched in neuroepithelial-derived stem and progenitor cell populations

We investigated how Neural G0 might arise during mammalian development by applying the ccAF classifier to data from the developing human telencephalon (Nowakowski *et al*, 2017). We analyzed scRNA-seq data from microdissected developing human cerebral cortex samples (PCW 5.85-19), which capture the spatial and temporal developmental trajectories for 24 cell types, including astrocytes, oligodendrocyte precursor cells (OPC), microglia, radial glia (RG), intermediate progenitor cells, excitatory cortical neurons, ventral medial ganglionic eminence progenitors, inhibitory cortical interneurons, choroid plexus cells, mural cells, and endothelial cells. We classified the cell cycle phase of every single cell using the ccAF classifier and cross-tabulated with the 24 cell types from Nowakowski *et al*, 2017 (Fig 2E; Dataset EV4). We found that the Neural G0 category was significantly enriched in excitatory neurons of the pre-frontal cortex (EN-PFCs), non-dividing astrocytes, OPCs, and RGs (ventral, outer, and truncated), which had a Neural G0 population ranging from 10 to 94% (Fig 2E; Dataset EV4). Populations characterized as dividing (i.e., "div", "div1", or "div2") were highly enriched with S/G2 and/or G2/M classified cells, and Neural G0 and G1 were absent or greatly diminished. Further, microglia had a tiny Neural G0 population and the G0/G1 pool of cells were instead classified as G1 and Late G1, which is interesting because they arise from the embryonic mesoderm rather than neuroectoderm (Ginhoux & Garel, 2018). It is likely that the terminally differentiated EN-PFC cell types were classified as Neural G0 rather than G1 due to their expression of the Neural G0 markers BEX1, BEX4, GPM6A, NOVA1, SCD5, and TGLN3. However, EN-PFCs were negative or low for key Neural G0 stem/progenitor markers, e.g., CLU, SOX2, SOX9, and S100B (Appendix Figs S5 and S6). These results suggest that the ccAF classifier identifies quiescent populations of adult and fetal neural stem cells and astrocyte subpopulations as Neural G0.

## Applying ccAF to non-neuroepithelial cells

We next tested whether it was appropriate to apply the ccAF classifier to non-neuroepithelial cell lines by applying it to 3,468 actively dividing human embryonic kidney (HEK293T) cells. ccAF primarily classifies HEK293T cells as S/G2 (39%), G2/M (19%), and M/Early G1 (39%), with a negligible number of quiescent Neural G0 cells (0.49%). The UMAP embedding has the characteristic cyclical pattern of the cell cycle (Appendix Fig S7A). We were surprised by the lack of a G1 population by ccAF, which ccSeurat predicts (29%). However, we realized that this is because the cells that would otherwise be classified as G1 retain residual G2/M gene expression (e.g., CCNB1, CDK1) (Appendix Fig S7). Thus, ccAF correctly calls them as M/Early G1, rather than G1. This difference is likely due to the transforming activity of SV40 Large T antigen, which is expressed in

these cells (Manfredi & Prives, 1994). We further observed that ccSeurat misclassifies cells situated between S and G2/M as G1 (Appendix Fig S7B). On the other hand, ccAF classifies these cells as S/G2, consistent with their placement in the cyclic embedding and expression of cyclins in these cells (Appendix Fig S7D). These results suggest that the ccAF classifier can resolve the cell cycle phases in a non-neuronal cell type even in the presence of a transforming factor that partially skews cell cycle gene expression.

## Neural G0 is a prominent subpopulation in human glioma cells

Gliomas are tumors of the central nervous system which originate from neuroepithelial cells (Chen *et al*, 2012; Zong *et al*, 2015). They contain subpopulations of cells that express genes associated with NSCs, OPCs, and astrocytes, which may contribute to progression, therapy resistance, and tumor recurrence (Dirks, 2008; Zong *et al*, 2015). Recently, scRNA-seq has been applied to human gliomas of different grades and subtypes to reveal intratumoral cellular heterogeneity (Patel *et al*, 2014; Tirosh *et al*, 2016b; Darmanis *et al*, 2017; Venteicher *et al*, 2017; Filbin *et al*, 2018; Neftel *et al*, 2019; Bhaduri *et al*, 2020; Wang *et al*, 2020). To address whether Neural G0 exists in gliomas, we used the ccAF classifier to analyze the scRNA-seq data available for 68 gliomas from these studies (Table 1; Fig 3 showing Neftel *et al*, 2019; Dataset EV4).

First, we filtered the datasets using a common filtering criterion to remove clusters of terminally differentiated oligodendrocytes (MBP and PLP1) (Valério-Gomes *et al*, 2018), astrocytes (ETNPPL) (Zhang *et al*, 2016c), neurons (RBFOX3) (Herculano-Houzel & Lent, 2005), and immune cells (AIF1, CD14, CX3CR1, PTPRC), which were distinct from tumor cell clusters. Each glioma dataset was loaded, normalized, and scaled if necessary. *De novo* clustering identified co-expressed cells, UMAPs were plotted with cell clusters (Appendix Fig S8A showing Wang *et al*, 2020), and expression of the genes above was overlaid onto the cells (Appendix Fig S8B). Clusters with high expression of these markers were excluded from further analysis as they were likely to be terminally differentiated cells. Further, the inferCNV algorithm (Patel *et al*, 2014) was applied to each dataset to confirm that cells from each tumor shared copy number alterations and therefore were likely to be neoplastic (Appendix Fig S9 showing Darmanis *et al*, 2017). This filtering and copy number analysis ensured that the cells used in further studies were neoplastic glioma cells.

The scRNA-seq profiled tumors represent a broad range of gliomas, including grades II, III, and IV, IDH1wt and mutant tumors, as well as glioma developmental subclasses (i.e., classical, mesenchymal, and proneural) and tumor types (i.e., astrocytoma, oligodendroglioma, GBM, and pediatric diffuse midline gliomas). Our analysis revealed that Neural G0 and G1 are the two most prominent tumor subpopulations regardless of stage (Table 1; Dataset EV4). Neural G0 and G1 represent 92.5 and 6.7%, respectively, of stage II oligodendrogliomas, 91.2 and 6.3% of stage III astrocytomas, 49.4–67.7% and 17.3–34.8% of stage IV GBMs, and 77.9 and 2% of diffuse midline gliomas (Table 1; Fig 3H). GBM subtype analysis (Wang *et al*, 2017) of each tumor cell further revealed that Neural G0 subpopulations were reduced in mesenchymal cell subpopulations in stage IV cancers (Table 1; Fig 3G and J). Overall the prevalence of the Neural G0 state diminished as stage increased regardless of subtype (Table 1; Dataset EV4).

**Table 1.  Percentages of cell cycle states classified by ccAF for primary neoplastic cells.**

| Data set | Tumor type | No. tumors | Other | Neural G0 | G1 | Late G1 | S | S/G2 | G2/M | M/Early G1 | No. cells |
|---|---|---|---|---|---|---|---|---|---|---|---|
| Tirosh 2016 (GSE70630) | II-O IDH1[mut] | 6 | 0 | 92.5 | 6.7 | 0 | 0.2 | 0.2 | 0.3 | 0 | 4,047 |
| | Classical | | 0 | 89.9 | 9.6 | 0 | 0.2 | 0.2 | 0.2 | 0 | 1,203 |
| | Mesenchymal | | 0 | 89.7 | 9.6 | 0 | 0.3 | 0 | 0.2 | 0.1 | 875 |
| | Proneural | | 0 | 95.4 | 3.8 | 0 | 0.2 | 0.4 | 0.3 | 0 | 1,969 |
| Venteicher 2017 (GSE89567) | III-A IDH1[mut] | 7 | 0 | 91.2 | 6.3 | 0 | 0.4 | 1.1 | 0.6 | 0.3 | 3,010 |
| | Classical | | 0 | 88.9 | 8 | 0 | 0.2 | 1.3 | 1.2 | 0.4 | 830 |
| | Mesenchymal | | 0 | 88 | 9.7 | 0 | 0.4 | 0 | 0 | 1.9 | 267 |
| | Proneural | | 0 | 92.6 | 5.2 | 0 | 0.5 | 1.2 | 0.4 | 0.1 | 1,913 |
| Darmanis 2017 (GSE84465) | IV-GBM IDH[wt] | 4 | 0.7 | 57.7 | 32 | 2.1 | 0.1 | 2.4 | 2.4 | 2.7 | 1,091 |
| | Classical | | 0.3 | 66.8 | 25.8 | 1.4 | 0.1 | 1.8 | 2.4 | 1.3 | 760 |
| | Mesenchymal | | 0.8 | 19.4 | 74.2 | 4.8 | 0 | 0.8 | 0 | 0 | 124 |
| | Proneural | | 2.4 | 46.9 | 29.5 | 2.9 | 0 | 5.3 | 3.9 | 9.2 | 207 |
| Neftel 2019 (GSE131928) | IV-GBM IDH[wt] | 22 | 1.1 | 67.7 | 17.3 | 0.3 | 1.5 | 3.7 | 4.5 | 4 | 11,376 |
| | Classical | | 1.4 | 67.7 | 18.1 | 0.1 | 1.2 | 4 | 5.2 | 2.3 | 5,630 |
| | Mesenchymal | | 0.7 | 36.2 | 55.1 | 1.4 | 2.3 | 1.3 | 1.3 | 1.7 | 1,036 |
| | Proneural | | 0.9 | 74.6 | 8 | 0.2 | 1.6 | 3.7 | 4.2 | 6.6 | 4,710 |
| Bhaduri 2020 | IV-GBM IDH[wt] | 11 | 0.2 | 55.8 | 26.8 | 0.2 | 1.2 | 5.5 | 8.3 | 1.9 | 21,177 |
| | Classical | | 0.1 | 66.4 | 27.7 | 0.1 | 0.3 | 1.6 | 3.4 | 0.3 | 7,446 |
| | Mesenchymal | | 0.5 | 26.3 | 66.7 | 0.4 | 1.1 | 1.2 | 3.5 | 0.4 | 4,093 |
| | Proneural | | 0.2 | 60.2 | 9.2 | 0.2 | 1.9 | 10.4 | 14.2 | 3.6 | 9,638 |
| Wang 2020 (GSE139448) | IV-GBM IDH[wt] | 3 | 5 | 49.4 | 34.8 | 0.7 | 0.1 | 2.7 | 3.2 | 4.1 | 13,525 |
| | Classical | | 5.5 | 57.1 | 26.1 | 0.3 | 0.1 | 4.3 | 3.4 | 3.2 | 3,842 |
| | Mesenchymal | | 5.8 | 16.9 | 72.1 | 1.6 | 0.2 | 0.5 | 1.4 | 1.5 | 4,947 |
| | Proneural | | 3.9 | 77.1 | 2.9 | 0 | 0 | 3.8 | 4.9 | 7.5 | 4,736 |
| Filbin 2018 (GSE102130) | DMG H3K27M | 6 | 1.1 | 77.9 | 2 | 0.1 | 0.6 | 2.1 | 7.7 | 8.5 | 2,775 |
| | Classical | | 0.9 | 86.9 | 2.6 | 0 | 0.3 | 0.9 | 5.5 | 2.9 | 344 |
| | Mesenchymal | | 2.3 | 74.7 | 3.4 | 0.7 | 0.9 | 2.3 | 8.1 | 7.7 | 443 |
| | Proneural | | 0.9 | 77.1 | 1.6 | 0.1 | 0.6 | 2.3 | 8 | 9.6 | 1,988 |
| Puram 2017 (GSE103322) | HNSCC | 21 | 0.1 | 11.2 | 48.1 | 4.8 | 10.2 | 2.1 | 11.2 | 12.3 | 2,215 |

II-O = grade 2 oligodendroglioma; III-A = grade 3 astrocytoma; IV-GBM = grade 4 GBM; DMG = diffuse midline glioma; HNSCC = head and neck squamous cell carcinoma; SKCM = skin cancer melanoma.

Examining non-tumor brain cell types associated with stromal tissue available from Darmanis et al (2017), showed that Neural G0 populations were found in neuroepithelial-derived cells such as astrocytes (98%), OPCs (79%), and oligodendrocytes (86%), in CD45[+] immune cells (77%), but were completely absent in vascular endothelial cells (0%). In scRNA-seq data from 21 primary and metastatic head and neck tumors (Puram et al, 2017), we observed that 48% of these neoplastic cells appeared in G1 and only 11.2% were classified as Neural G0 (Table 1). These results suggest that the Neural G0 state may be applied more broadly to other human cell types beyond neuroepithelial-derived cells.

We also compared the ccAF-predicted cell classifications for individual GBM tumors with de novo clustering, ccSeraut, and developmental subtype classifications (Fig 3A–D). This comparison reveals that ccAF further stratifies clusters that are not apparent in ccSeraut,

and better matches de novo clustering of tumor populations (Fig 3A–C). This observation was also true at the study level when considering all tumors profiled by Neftel et al, 2019 (Figure 3E-J). Comparing cell cycle classifications from ccAF to the Neftel et al, 2019, alternative GBM developmental classification scheme (e.g., astrocytic (AC), neural progenitor cell (NPC), oligodentrocyte progenitor cell (OPC), and mesenchymal (MES)) shows that the two methods of characterizing had some similarities and some differences (Fig 3K and L). Most cells classified as AC, NPC, or OPC were also classified as Neural G0, while MES populations had fewer Neural G0 and more G1 cells. The MES and NPC cells had a higher S, S/G2, and G2M fraction than AC and OPC cells (Fig 3L). Thus, the abundance of Neural G0 cells in AC and OPC cells is consistent with Neural G0 representing a quiescent state and/or a pre-mesenchymal state associated with the proneural-to-mesenchymal

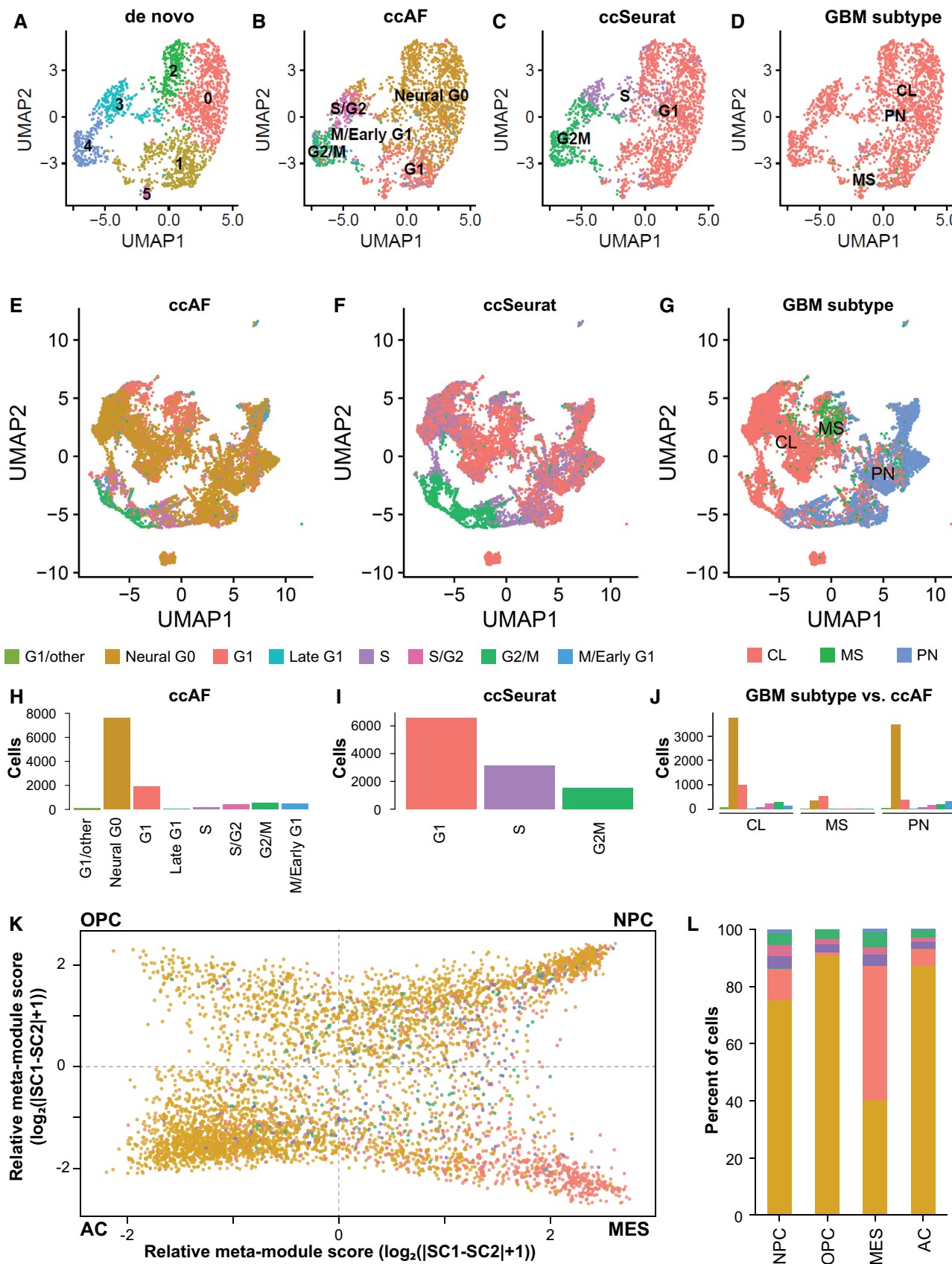

**Figure 3.**

**Figure 3.   Application of the ccAF classifier to glioblastoma tumors.**

A   *De novo* clustering of all cells from patient MGH143 (Neftel *et al*, 2019; *n* = 2,182 cells).
B   ccAF applied to all cells from patient MGH143 matches well to *de novo* clustering.
C   ccSeurat applied to all cells from patient MGH143 matches well to *de novo* clustering, but has S phase in between G2 M and G1, similar to what was observed in HEK293T (Appendix Fig S4).
D   GBM subtypes of all cells from patient MGH143 show that the vast majority of cells are from the classical subtype.
E   Application of the ccAF classifier to primary GBM neoplastic cells from Neftel *et al*, 2019 overlaid on a two-dimensional UMAP cell embedding (*n* = 11,376).
F   Application of the ccSeurat classifier to primary GBM neoplastic cells from Neftel *et al*, 2019 overlaid on a two-dimensional UMAP cell embedding (*n* = 11,376).
G   Application of the ssGSEA GBM subtype classifier to primary GBM neoplastic cells from Neftel *et al*, 2019 overlaid on a two-dimensional UMAP cell embedding (*n* = 11,376). CL = classical; MS = mesenchymal; PR = proneural.
H   Tabulation of cell counts for ccAF cell cycle phase classifications in E.
I   Tabulation of cell counts for ccSeurat cell cycle phase classifications in F.
J   Tabulation of cell counts for ccAF cell cycle phase classifications for each GBM subtype. CL = classical; MS = mesenchymal; PR = proneural.
K   Two-dimensional embedding representing cellular state from Neftel *et al*, 2019 (Fig 3F), where each quadrant represents a different cellular state oligodendrocyte progenitor cell-like (OPC), neural progenitor cell-like (NPC), mesenchymal-like (MES), and astrocyte-like (AC).
L   Tabulation of cell counts for ccAF cell cycle phase classifications in each quadrant of K (OPC-like, NPC-like, MES-like, and AC-like).

transition (Bhat *et al*, 2013; Halliday *et al*, 2014; Segerman *et al*, 2016).

## Putative stem-like cells are enriched in Neural G0 subpopulation of primary GBM tumors

To further investigate the cells within the Neural G0 population of GBM tumors, we examined the expression of markers associated with GBM stem-like cells (GSCs). The prevailing rationale for GSCs is that a small portion of GBM tumors have evolved stem-like characteristics and generate tumor cell heterogeneity (Lathia *et al*, 2015). The tumor stem-like cells may be slow dividing cells that are missed by surgery and which are resistant to standard of care treatments (Lathia *et al*, 2015). This concept was elegantly demonstrated in a mouse model of glioma where a quiescent subset of endogenous glioma cells were shown to be responsible for tumor regrowth after temozolomide treatment (Chen *et al*, 2012). However, unlike other non-transformed stem cell types (e.g., neural), there are no pre-existing, universal markers that can neatly resolve GSC subpopulations into quiescent, primed, or activated states (Lathia *et al*, 2015).

Therefore, to assess the prevalence of putative stem-like cells in GBM datasets, we adopted the method of Bhaduri *et al*, 2020. They defined a logic for discovering putative stem cells from scRNA-seq profiles: any cell expressing FUT4 (SSEA1) or L1CAM or PROM1 (CD133) in conjunction with SOX2 and not expressing TLR4 (Bhaduri *et al*, 2020). We applied this logic to discover 4,563 putative stem-like cells in 47,405 neoplastic cells from four GBM scRNA-seq studies (9.6% of neoplastic cells were putative stem-like cells; Fig 4A) (Darmanis *et al*, 2017; Neftel *et al*, 2019; Bhaduri *et al*, 2020; Wang *et al*, 2020). Putative stem-like cells were significantly enriched in Neural G0 classified cells (70% of putative stem-like cells are in Neural G0; hypergeometric enrichment *P*-value = $3.4 \times 10^{-60}$; Fig 4B; Dataset EV6). We then projected the putative stem-like cell marker genes onto the Bhaduri *et al*, 2020 (Fig 4C) and Neftel *et al*, 2019 embedding (Fig 4D). We discovered that L1CAM is expressed at 1.2–3.6% of OPC and NPC cells and is expressed in only 0.1–0.3% of AC and MES cells. PROM1 also shows a bias toward NPC cells (21.7%) and away from AC cells (9.8%), with OPC and MES in the middle (15.6 and 13.1%) (Fig 4D). These results suggest that Neural G0 populations harbor

putative stem-like cell subpopulations and that Neural G0 captures multiple subpopulations of non-dividing cells.

## Higher Neural G0 expression is associated with better patient prognosis in gliomas

Because the ccAF classifier was developed on NSCs, we wanted to characterize and tailor the Neural G0 for GBM, thereby ensuring that the Neural G0 signature was more relevant to disease. Thus, we identified GBM neoplastic cell-specific Neural G0 marker genes by applying ccAF to four GBM scRNA-seq studies (Table 1) (Darmanis *et al*, 2017; Neftel *et al*, 2019; Bhaduri *et al*, 2020; Wang *et al*, 2020) and discovered 22 Neural G0 marker genes in common across all four studies (Fig 5A and B; Dataset EV5). Eight of the 22 common GBM neoplastic cell-specific Neural G0 marker genes were originally identified as Neural G0 marker genes for hNSCs (*GPM6A*, *HOPX*, *MARCKSL1*, *PLP1*, *S100B*, *SCD5*, *SCRG1*, and *TTYH1*; Fig 5B; Dataset EV1). The remaining 14 genes were unique to GBM neoplastic cells (*AQP4*, *BCAN*, *BCHE*, *GATM*, *GFAP*, *ITM2C*, *NDRG2*, *PLEKHB1*, *PMP2*, RAMP1, *RTN3*, *SLC22A17*, *TSC22D4*, and *TSPAN7*; Fig 5B; Dataset EV5). Significantly, 13 of the 22 genes were previously known to be associated with GBM in the DisGeNET database (*AQP4*, *BCAN*, *BCHE*, *GFAP*, *HOPX*, *MARCKSL1*, *NDRG2*, *PLEKHB1*, *PLP1*, *S100B*, *SLC22A17*, *TSPAN7*, and *TTYH1*; hypergeometric over-enrichment *P*-value = $1.2 \times 10^{-6}$; Fig 5B) (Piñero *et al*, 2015). Of these, *AQP4* has previously been shown to be differentially expressed in quiescent astrocytes (Yoneda *et al*, 2001); *HOPX* is a marker of quiescent radial glial neural progenitors (Berg *et al*, 2019); *NDRG2* is up-regulated in G0/G1 arrested glioma cells (Li *et al*, 2012, 2); *S100B* is a chemoattractant for tumor-associated macrophages (Wang *et al*, 2013); and *TTYH1* is required to maintain NSC stemness via its role in activating the Notch signaling pathway (Kim *et al*, 2018, 1). The genes *AQP4*, *BCAN*, *GFAP*, *PLP1*, and *S100B* are part of the astrocytic, oligodendrocytic, or proneural glioma signatures. Interestingly, high levels of PLP1 expression are a marker for terminally differentiated oligodendrocytes, whereas moderate expression is a marker for neoplastic Neural G0 cells (Fig 5C). GFAP is more heavily expressed in neoplastic Neural G0 cells relative to other cell types, with astrocytes and OPCs coming in a close second (Fig 5D), and the AQP4 gene is equivalently expressed by astrocytes and neoplastic Neural G0 cells (Fig 5E).

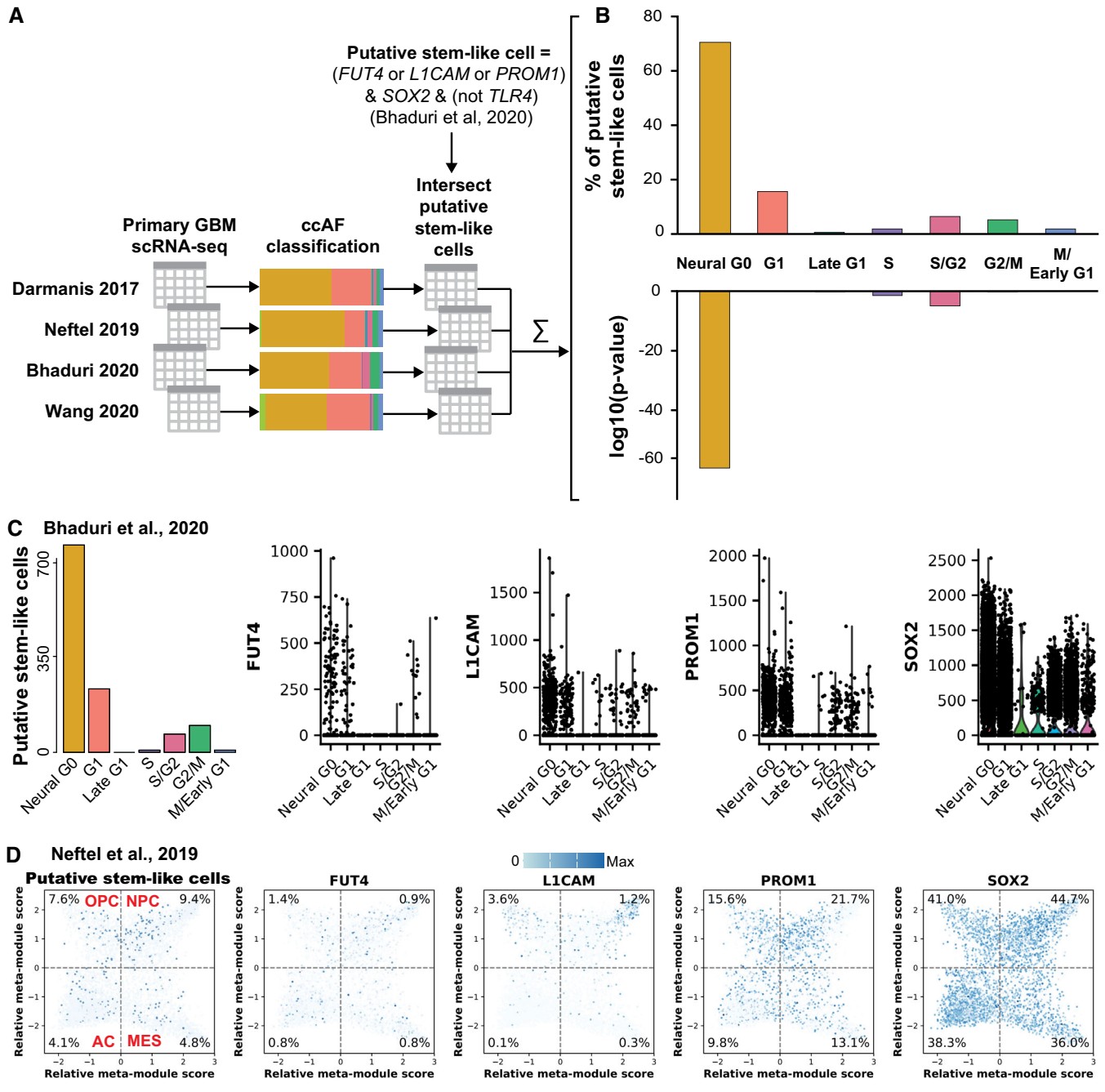

**Figure 4. Putative stem-like cells are significantly enriched in primary Neural G0 tumor subpopulations.**

A The ccAF classifier was applied to four GBM scRNA-seq datasets with a total of 47,169 cells (Darmanis *et al*, 2017; Neftel *et al*, 2019; Bhaduri *et al*, 2020; Wang *et al*, 2020). We then applied the logic for discovering putative stem-like cells from Bhaduri *et al*, 2020 to all 47,169 cells: (*FUT4* > 0 or *L1CAM* > 0 or *PROM1* > 0) & *SOX2* > 0 & (*TLR4* = 0). The subpopulation of cells from each ccAF cell cycle phase was intersected with the putative stem-like cell subpopulation. The resulting intersections for each dataset were summed and a hypergeometric enrichment *P*-value was computed for each ccAF cell cycle phase comparison.

B Top, Percentage of putative stem-like cells that map to each ccAF cell cycle phase. Bottom, Log 10 of hypergeometric enrichment *P*-values for each ccAF cell cycle phase. More negative values indicate increased significance.

C Number of putative stem-like cells in each ccAF cell cycle phase, and distribution of absolute gene expression across ccAF cell cycle phases for the genes used to identify putative stem-like cells in Bhaduri *et al*, 2020.

D Putative stem-like cells and expression of the genes used to identify the putative stem-like cells projected onto Neftel *et al*, 2019. Expression of genes is overlaid as the color of each cell (white = 0, dark blue = max expression). Quadrants are broken up by dashed gray lines: OPC = upper left; NPC = upper right; AC = lower left; and MES = lower right. Percentages of gene expression greater than 0 are shown in the corners of the plot.

While the 22 common GBM neoplastic cell-specific Neural G0 marker genes make poor markers alone as several of them show expression at the same level in at least one other cell type, combining expression of all of them together allows the signature to achieve discriminative power and specificity (Appendix Fig S10).

We further examined the roles for these genes and the Neural G0 gene signature in gliomas by investigating the relationship between Neural G0 marker gene expression and glioma patient survival. We interrogated data from 681 gliomas (grade II, III, and IV) from The Cancer Genome Atlas (TCGA). In addition to transcriptome profiles, the TCGA includes phenotypic and genetic information, which allowed us to include tumor grade and IDH1/2 mutational status as previously associated covariates in our model. We computed eigengenes for both the 22 GBM neoplastic cell-specific Neural G0 marker genes and the 54 cell cycle genes from the Seurat G2M classifier gene list. An eigengene represents the common variation across each patient tumor, i.e., the first principal component corrected for direction if necessary. The Neural G0 eigengene is significantly associated with tumor grade (Fig 5F). This is consistent with the observation from scRNA-seq that lower grade gliomas are predicted to have more quiescent Neural G0 cells than GBMs tumors (Table 1). The Neural G0 eigengene is significantly anti-correlated with the cell cycle eigengene ($R = -0.48$, $P$-value $< 2.2 \times 10^{-16}$; Fig 5G). Moreover, the Neural G0 versus cell cycle eigengene plot displays a characteristic L-shaped distribution which strongly suggests mutual independence of the two eigengenes (Fig 5G). This mutual independence is likely due to the fact that the cell cycle and quiescence are regulated by different means.

Next, we assessed the relationship between the Neural G0 eigengene and patient survival using a Cox proportional hazards regression model that included the covariates tumor grade and IDH1/2 mutation status. The Neural G0 eigengene was significantly associated with patient survival even with the inclusion of tumor grade and IDH1/2 mutation status (Cox PH coef. $= -8.0 \pm 3.4$; $P$-value $= 1.5 \times 10^{-2}$). This strongly suggests that the Neural G0 cell state is associated with patient survival variance independently from the common glioma survival-associated covariates (tumor grade, IDH1/2). Additionally, the Neural G0 eigengene was significantly associated with patient survival when the cell cycle eigengene is included in the model (Cox PH coef. $= -0.14$; and $P$-value $= 9.8 \times 10^{-9}$). Thus, Neural G0 has an independent effect beyond the cell cycle effects, and therefore, the Neural G0 state is not simply the opposite of an actively cycling cell state. We also computed a Kaplan–Meier plot with the Neural G0 eigengene top 25% versus the bottom 25% ($n = 175$ for each) Neural G0 gene expression (Fig 5H and I). The top 25% in Neural G0 eigengene expression had a median survival of 1,826 days, whereas the bottom 25% had a median survival of 342 days, which is a difference of 4.1 years in patient survival (Fig 5H). Subsetting the TCGA to only grade III gliomas can address tumor grade as a potential confounding factor. As before, increased Neural G0 eigengene in grade III gliomas is a significant predictor of better prognosis ($P$-value $= 3.0 \times 10^{-5}$). The top 25% in Neural G0 eigengene expression in grade III gliomas had a median survival of 1,674 days and the bottom 25% had a median survival of 292 days, which is a difference of 3.8 years in patient survival. This strongly suggests that the Neural G0 GBM-specific marker genes describe a cell state that is independently predictive of patient survival.

Taken together, these results demonstrate that Neural G0 cells represent significant subpopulations in gliomas which diminish by grade and are associated with better clinical outcomes. Thus, the results are consistent with a model whereby higher steady-state Neural G0 populations remove cells from the pool of cycling cells, leading to slower tumor growth.

## CRISPR-Cas9 gene knockout screens identify regulators of Neural G0 *in vitro*

Given the association of Neural G0 populations with more indolent glioma tumor growth, we wished to investigate whether the Neural G0 state indeed causes slower cell cycles. An alternative hypothesis, for example, is that Neural G0 in our cultured NSCs or in gliomas could represent a "terminal" exit from the cell cycle, rather than a transient G0-like state. For cultured hNSCs, we reasoned that if Neural G0 ingress/egress is rate-limiting for NSC cell cycles, diminishing Neural G0 would cause NSCs to cycle faster. If true, a simple

**Figure 5.  Identifying GBM-specific Neural G0 marker genes and application to 641 human gliomas.**

A   A GBM-specific Neural G0 expression signature was developed using four GBM scRNA-seq datasets with a total of 47,169 cells (Darmanis *et al*, 2017; Neftel *et al*, 2019; Bhaduri *et al*, 2020; Wang *et al*, 2020). The ccAF classifier was applied to each dataset, marker genes were discovered for the Neural G0 phase in each dataset, and 22 common marker genes were identified between all four datasets.

B   A network showing the inter-relatedness of the 22 common GBM-specific Neural G0 marker genes. Genes with red circles were previously known to be associated with GBM in the DisGeNET database.

C   Violin plot showing PLP1 expression across all cell types from Darmanis *et al*, 2017. Data are log 10-transformed before plotting. Expression of PLP1 in single cells is much higher in oligodendrocytes (Oligo) than in Neural G0 or oligodendrocyte progenitor cells (OPC).

D   Violin plot showing GFAP expression across all cell types from Darmanis *et al*, 2017. Data are log 10-transformed before plotting. Expression of GFAP in single cells is more highly expressed in Neural G0 relative to all other cell types.

E   Violin plot showing AQP4 expression across all cell types from Darmanis *et al*, 2017. Data are log 10-transformed before plotting. Neural G0 expression of AQP4 in single cells is similar to astrocytes but has a lower median expression and lower upper limit.

F   Relative neural G0 eigengene expression between grade II ($n = 226$), III ($n = 244$), and IV ($n = 150$) tumors (TCGA; LGG and GBM). An eigengene represents the common variation across each patient tumor for the Neural G0 genes, i.e., first principal component corrected for direction if necessary. All pairwise Student's *t*-tests comparisons had $P$-values $< 0.003$.

G   Comparison of cell cycle (54 genes annotated to the Seurat G2 M classifier gene list) and Neural G0 eigengene relative expression in each glioma. Each tumor is colored by its grade (green = II, red = III, and purple = IV).

H   Kaplan–Meier survival plot of tumors with top 25% ($n = 171$) and bottom 25% ($n = 171$) of Neural G0 eigengene expression of Neural G0 genes. A Fleming–Harrington survival $P$-value was used to determine significance. Shaded region is the 95% confidence interval for the survival curve.

I   Distribution of tumor grade between tumors with top 25% and bottom 25% of Neural G0 eigengene expression.

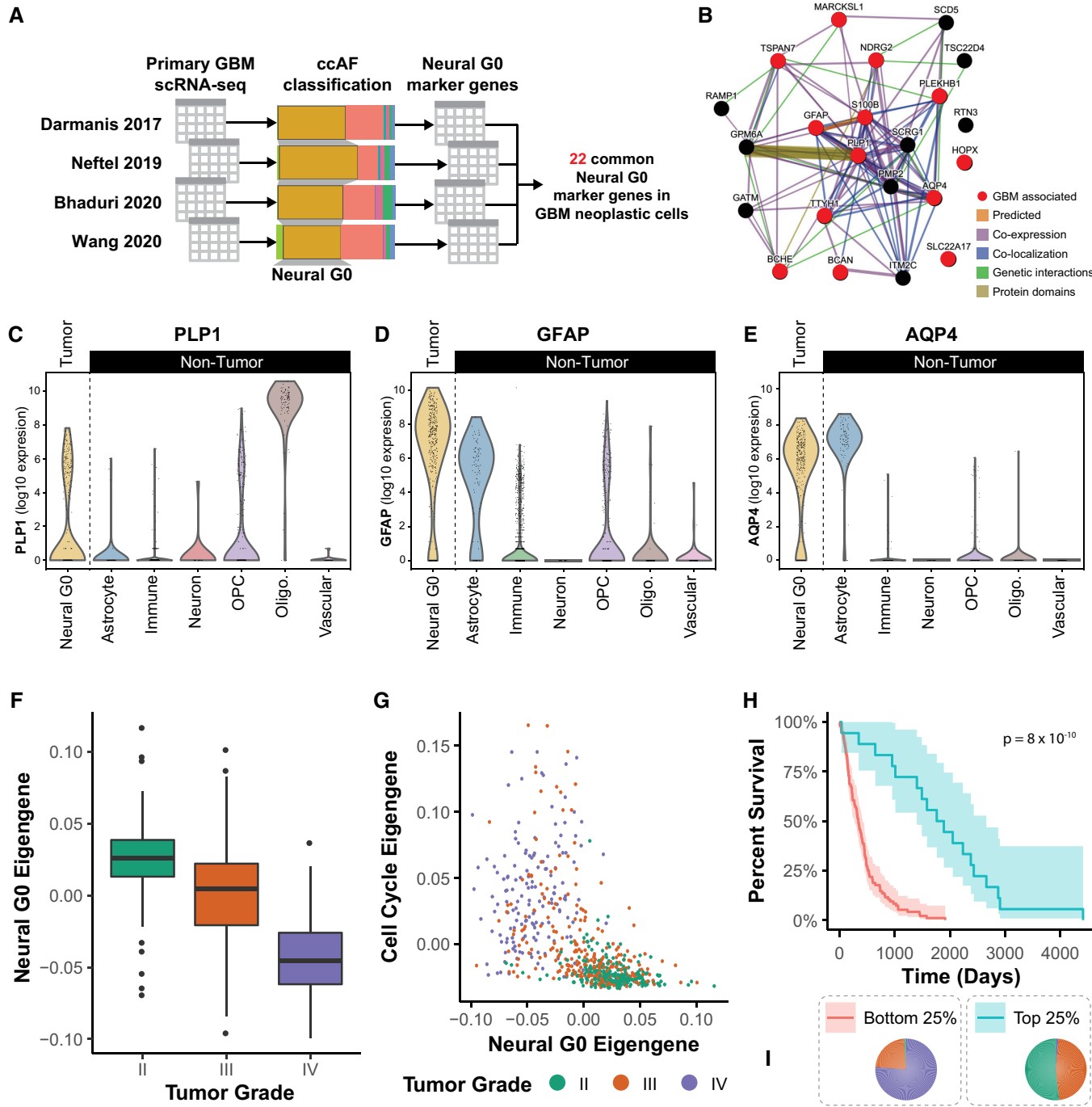

**Figure 5.**

pooled LV-CRISPR-Cas9 sgRNA library outgrowth screen in normal culture conditions should reveal overrepresented sgRNAs that cause diminished Neural G0 (Appendix Fig S11A).

To this end, we performed four separate CRISPR-Cas9 outgrowth screens, using three separate libraries, two different time points (10 days versus ~3 weeks), and two different human NSC isolates, CB660 and U5 (Appendix Figs S11 and S12; Dataset EV7) (Pollard *et al*, 2006; Bressan *et al*, 2017). These screens revealed dozens of candidate screen hits significantly enriched at the end of outgrowth period (Appendix Fig S12A). The sgRNAs targeted genes found mutated across 35 different tumor types (Appendix Fig S12C) and

validated tumor suppressor genes (Futreal *et al*, 2004) (Appendix Fig S12D). Examining the intersection of all of the screen data revealed five reproducible and robust proliferation-enhancing screen hits: *CREBBP*, *NF2*, *PTPN14*, *TAOK1*, and *TP53* (Appendix Figs S11 and S12A and B). These genes have previously suggested roles in the p53 pathway (for *TP53* and *CREBBP*) (Ito *et al*, 2001; Fischer, 2017) or the Hippo-YAP signaling pathway (for *NF2*, *PTPN14*, and *TAOK1*) (Zhang *et al*, 2010; Lin *et al*, 2013; Wilson *et al*, 2014; Plouffe *et al*, 2016). The p53 pathway has well-documented roles as a tumor and growth suppressor in brain tumor cells (Mercer *et al*, 1990; Van Meir *et al*, 1995; Brennan *et al*, 2013). The Hippo-Yap signaling pathway has been

implicated in the GBM proneural-to-mesenchymal transition to promote mesenchymal tumor cell expansion and tumor regrowth after standard of care (Minata *et al*, 2019). We chose to characterize the impact of these genes on Neural G0 further.

Knockout (KO) of *CREBBP, NF2, PTPN14, TAOK1,* and *TP53* in hNSCs caused a significant proliferative advantage over control cells in a 23-day outgrowth competition assay. In contrast, KO of the essential gene *KIF11* showed the opposite result (Appendix Fig S12A). However, the competitive advantage did not appear to be based on differences in survival since no changes in Annexin-V staining were observed following normal culturing or in co-cultures. Apoptosis remained < 2% regardless of the experimental condition.

Using cell proliferation assays (Appendix Fig S13B–D), we found that each KO significantly increased cell accumulation in 48- to 96-h outgrowth assays. Importantly, this effect was independent of cell density, as KO cells showed increased proliferation at both low and high densities (Appendix Fig S13B). Further, the doubling time significantly decreased for each KO, shortening from ~50 h to 30–40 h (Appendix Fig S13E), similar to two GSC isolates used in the same assay.

### Neural G0 is a transient state of variable length which determines hNSC cell cycle length and is reduced after KO of *CREBBP*, *NF2*, *PTPN14*, *TAOK1*, or *TP53*

We utilized the fluorescent ubiquitination cell cycle indicator (FUCCI) system to further investigate changes in cell cycle dynamics (Sakaue-Sawano *et al*, 2008). In normal culture conditions, ~63% of U5-NSCs cells were in G0/G1, ~15% were in S/G2/M, and the remainder were transitioning between these phases (Fig 6A). KO of *CREBBP*, *NF2*, *PTPN14*, *TAOK1*, or *TP53*, however, caused a dramatic loss of the G0/G1 populations (reducing the frequency to 47-38%) and significantly lowered the ratio of G0/G1 to S/G2/M cells (~2- to 4-fold lower) (Fig 6B and C).

We also measured transit time through G0/G1 and S/G2/M in individual NSCs using time-lapse microscopy (Fig 6D; Appendix Fig S14). We found that our control hNSCs exhibit variable G1 transit times and a wide distribution of G0/G1 transit times, from fast (4.3 h), medium, and extremely slow (95 h; averaging 32.5 h; Fig 6D). By contrast, S/G2/M transit times were much more uniform (~12.4 h; Fig 6D). KO of *CREBBP*, *NF2*, *PTPN14*, *TAOK1*, or *TP53* dramatically collapsed the distributed G0/G1 transit times leading to a highly significant, faster transit of < 11.7 h in KOs (*P*-value<0.0001; Fig 6D and Appendix Fig S14). S/G2/M transit times were not significantly affected. GSCs also exhibit collapsed and faster G0/G1 transit times similar to the KO hNSCs (Fig 6D).

To further examine possible changes in G0/G1 dynamics, we examined molecular features associated with G0, G1, and Late G1 (Appendix Fig S15A), including Rb phosphorylation, CDK2 activity, and p27 accumulation. In mammals, cell cycle ingress is governed by progressive phosphorylation of Rb by CDK4/6 and CDK2 as cells pass through the restriction point in Late G1, causing de-repression of E2F transcription factors (Weinberg, 1995; Zetterberg *et al*, 1995; Sherr & McCormick, 2002; Yao *et al*, 2008). We observed that KO of *CREBBP*, *NF2*, *PTPN14*, *TAOK1*, or *TP53* in U5-NSCs results in a pronounced increase in the intensity of phosphorylated Rb during G1, consistent with enrichment of the Late G1 state (Appendix Fig S15B).

CDK2 activity correlates with cell cycle progression. If CDK2 activity levels are low during G1, cells enter G0 (Spencer *et al*, 2013); if CDK2 activity is intermediate (relative to its peak during G2/M), they progress past the restriction point and into S phase (Spencer *et al*, 2013). Using the steady-state cytoplasmic to nuclear ratios of a DNA helicase B (DHB)-mVenus reporter as a readout of CDK2 activity (Hahn *et al*, 2009; Spencer *et al*, 2013), we observed significant increases in CDK2 activity in each KO in G0/G1 cells (Appendix Fig S15C and D). This was true either by total intensity or the proportion of cells with a reporter ratio greater than 1, a ratio which corresponds with the entrance to S phase observed in mammary epithelium (Spencer *et al*, 2013). Control cells averaged ~8% of G1 cells with > 1 cytoplasmic:nuclear reporter ratios CDK2 activity, while KOs were 20-27% (Appendix Fig S15D).

Another hallmark of G0/quiescence is the stabilization of p27, a G1 cyclin-dependent kinase (CDK) inhibitor required for maintaining G0 (Coats *et al*, 1996; Susaki *et al*, 2007). Consistent with loss of transient G0 cells, we observed that KO of *CREBBP*, *NF2*, *PTPN14*, *TAOK1*, or *TP53* resulted in a significant reduction of p27 levels in proliferating NSCs (Appendix Fig S15E and F).

Collectively, the above data demonstrate that KO of proliferation-limiting genes in U5-NSCs causes a cell autonomous decrease in cell cycle length with less distributed and faster G0/G1 transit times, an increase in the molecular features associated with Late G1, and a reduction in the molecular features associated with G0 (Appendix Fig S15G). These data are consistent with KOs either blocking the entry of cells into a transient G0 state or causing failure to maintain cells in G0. Therefore, we call these G0-skip genes.

### G0-skip mutants reprogram G0/G1, diminishing Neural G0 gene expression

To further characterize G0-skip genes, we performed a gene expression analysis of KO cells specifically in G0/G1 phase. To this end, RNA-seq was performed on mCherry-CDT1[+] sorted NSCs after KO,

---

**Figure 6.  Reduction of G0/G1 transit time in NSCs after KO of *CREBBP*, *NF2*, *PTPN14*, *TAOK1*, or *TP53*.**

A   Representative contour plot of flow cytometry for FUCCI (Sakaue-Sawano *et al*, 2008) in U5-NSCs after targeting of a non-growth limiting (NGL) control gene, *GNAS1*. Values are similar to wild type and non-targeting control (NTC) U5-NSCs under similar culture conditions. The system relies on cell cycle-dependent degradation of fluorophores using the degrons from CDT1 (amino acids (aa) 30-120) (present in G0 and G1; mCherry) and geminin (aa1–110) (present in S, G2, and M; monomeric Azami-Green (mAG)).

B   Representative contour maps of flow cytometry for FUCCI following the loss of *NF2*, *PTPN14*, *TAOK1*, *CREBBP*, and *TP53*.

C   Ratio of G0/G1 (mCherry-CDT1[+]) to S/G2/M (mAG-Geminin[+]) from (A) and (B). Values are mean from 4 individually tested LV guides per gene at 21 days post-selection.

D   G0/G1 and S/G2/M transit times using time-lapse microscopy and FUCCI. Differences in G0/G1 are statistically significant with *P* < 0.001 for targeted U5-NSCs and *P* = 0.0006 for GSC-131 compared to NTC.

Data information: The data are presented as the mean ± SD. Supporting information is provided in Appendix Fig S14. Significance was assessed using a two-tailed Student's *t*-test (C) or Mann–Whitney test (D).

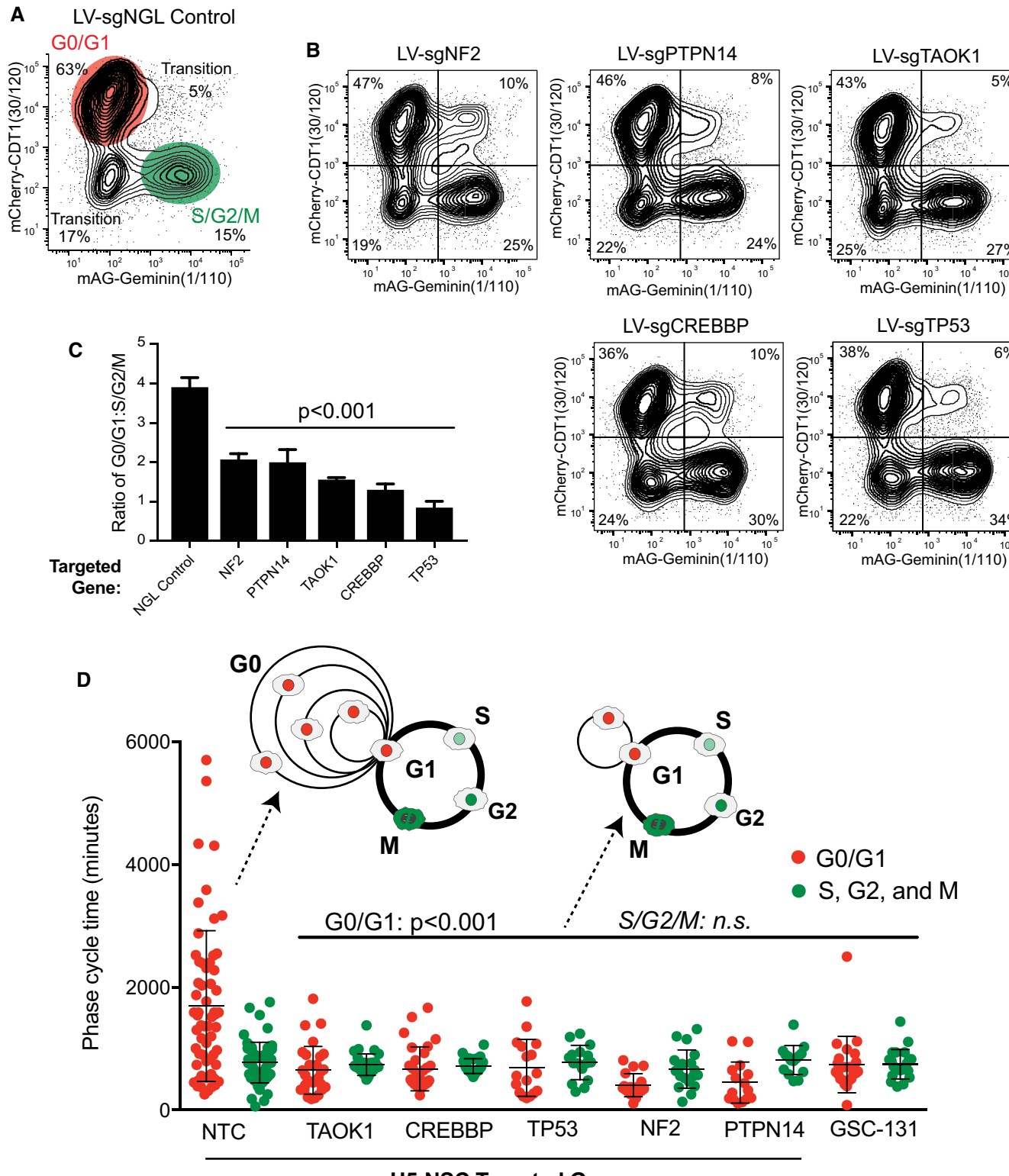

**Figure 6.**

which captures both G0 and G1 subpopulations (Fig 7A; Dataset EV8). In control NSCs, as expected, comparing G0/G1-sorted cells to unsorted populations revealed down-regulation of genes involved in cell cycle regulation, DNA replication, and mitosis (Fig 7A; Dataset EV9). Overall comparisons between the KOs and non-targeting control (NTC) U5-NSCs showed that KO of *NF2* and *PTPN14* were most similar by unsupervised clustering and the most overall gene changes, while *TAOK1* KO was most similar to the controls (Fig 7B).

However, a comparison of the overlapping up- or down-regulated genes showed that *TAOK1* KO up-regulated genes were more similar to *NF2* and *PTPN14* KO than the other KOs (Appendix Fig S16A).

We next evaluated whether KO of the G0-skip genes were consistent with previously published and suggested roles in the p53 pathway (for *TP53* and *CREBBP*) (Ito *et al*, 2001; Fischer, 2017) or the Hippo-YAP signaling pathway (for *NF2*, *PTPN14*, and *TAOK1*) (Zhang *et al*, 2010; Lin *et al*, 2013; Wilson *et al*, 2014; Plouffe *et al*, 2016). Evaluating p53 target genes, we found that only *TP53* KO significantly down-regulated the expression of high confidence p53 targets including *BAX*, *CDKN1A/p21*, *RRM2B,* and *ZMAT3* (Fig 7C; Appendix Fig S16B) (Fischer, 2017). However, none of the other KOs showed inhibition of p53 targets or p53 itself, strongly suggesting that the other G0-skip genes are not acting through p53-dependent transcriptional activity.

Evaluation of 55 conserved HIPPO-YAP pathway transcriptional targets (Cordenonsi *et al*, 2011) revealed that each KO, except for *CREBBP*, showed significant enrichment for YAP targets with *NF2* KO having increased expression of the largest subset (Fig 7C; Appendix Fig S16C–E). Interestingly, *NF2* KO activated one subset of YAP targets important in the biological process of extracellular matrix (ECM) organization, while *TAOK1* KO activated a different subset of YAP targets important in nuclear chromosome segregation, such as during mitosis (Appendix Fig S16C–E). *NF2* and *PTPN14* KO shared the most overlap in YAP target activation, including targets considered universal Hippo-YAP targets (e.g., *CTGF*, *CYR61*, and *SERPINE1*). Intriguingly, many of these Hippo-YAP target genes can be found in the mesenchymal GBM gene signature and are also up-regulated as the result of the GBM proneural-to-mesenchymal transition (e.g., ANGPTL4, COL1A1/2, CTGF, CYP1B1, ITGA1, LIF, and THBS1) (Minata *et al*, 2019).

We next used the ccAF classifier to determine whether genes associated with each phase change in G0/G1 populations after KO of *CREBBP*, *NF2*, *PTPN14*, *TAOK1*, or *TP53*. We observed that Neural G0 was significantly down-regulated in each KO (Fig 7D, Appendix Fig S17), which included those expressed in quiescent NSCs and others cited above with key roles in neural development (e.g., *CLU*, *HOPX*, *ID3*, *PTN*, *PTPRZ1*, *SOX2*, and *SOX4*; Appendix Fig S17B and C). By contrast, genes from the Late G1 cluster, including, for example, *CCND1* and *MYC*, were significantly up-regulated in each KO, with *TAOK1* KO cells additionally showing an increase in cell cycle phases as well (Fig 7E; Appendix Fig S18A–C). Examination of G0/G1-sorted populations from two GSC isolates

(0131-mesenchymal and 0827-proneural) showed similar trends, with suppression of Neural G0 and G1 signatures and higher expression of S and G2/M genes (Appendix Fig S19).

For NSC KOs, we also performed a more in-depth analysis of transcriptional changes of cell cycle genes and novel gene sets (Appendix Fig S20). These included cell cycle genes that could be causal for reprogramming G0/G1 dynamics, such as up-regulation of G1 cyclins, E2F1/2 or down-regulation of CDKN1A/p21 and CDKN1B/p27 (Appendix Fig S20A). We also noted that for both *NF2* and *PTPN14* KO, there was up-regulation of various Hippo-YAP pathway members, including *LATS2*, *TEAD1,* and *YAP1*, suggesting a possible feedback regulation of the pathway unique to *NF2* and *PTPN14* (Appendix Fig S20B). TAOK1 KO, in contrast to other KOs, strongly up-regulated > 40 key regulators of mitosis (e.g., *AURKA*, *BUB1*, *CCNB1/2*, *CDK1*, and *KIF11*), suggesting it may act to inhibit their precocious activation in G0/G1 or expression after completion of mitosis (Appendix Fig S20C).

*CREBBP* KO, unique among KOs, caused up-regulation of key nuclear-encoded mitochondrial genes, including members of the NADH dehydrogenase complex, the succinate dehydrogenase complex, and mitochondrial DNA polymerase (Appendix Fig S20D), which are direct transcriptional regulatory targets of nuclear respiratory factors 1 and 2 (NRF1 and NRF2) (Kelly & Scarpulla, 2004).

Finally, to more directly confirm reprogramming of G0/G1 population in a G0-skip mutant, we applied the ccAF classifier to scRNA-seq profiles from $CDT^+$ G0/G1-sorted hNSCs with KO of *TAOK1* and compared that to WT sorted scRNA-seq profiles (Fig 7F and G). The percentage of Neural G0 phase classified cells in $CDT^+$ WT was 16.3% and was significantly reduced to 4.1% in $CDT^+$ *TAOK1* KO ($P$-value $< 2.2 \times 10^{-16}$; Fig 7G). The S phase classified cells were also significantly decreased in $CDT^+$ WT versus $CDT^+$ *TAOK1* KO cells (from 2.7% to 0.67%; $P$-value $= 9.9 \times 10^{-7}$). On the other hand, the Late G1 classified cells were significantly increased (from 5.8 to 17.0%; $P$-value $< 2.2 \times 10^{-16}$), as were cells in the M/Early G1 (from 11.2 to 18.8%; $P$-value $= 8.3 \times 10^{-13}$) and G2/M (from 1.0 to 1.6%; $P$-value $= 0.036$). The expansion of the M/Early G1 in *TAOK1* KO cells could explain the increase in mitotic genes observed in the bulk G0/G1 RNA-seq data in *TAOK1* KO cells (Fig 7F), suggesting that TAOK1 helps attenuate the expression of mitotic genes from the previous cell cycle. The highly significant drop in Neural G0 cells and redistribution to the mitotic adjacent Late G1 and M/Early G1 phases supports the hypothesis that NSC G0-skip mutants lose a significant fraction of the Neural G0

**Figure 7. Transcriptional reprogramming of G0/G1 following loss of G0-skip genes.**

A Schematic of G0/G1 sorting for gene expression analysis: mCherry-CDT1$^+$ U5-NSCs (red box), heat maps of log2(fold-change) between G0/G1 NTC the significantly altered genes (FDR < 0.05) between WT unsorted U5-NSCs and non-targeting control (NTC) and WT G0/G1 U5-NSCs, and gene ontology analysis (Young *et al*, 2010) of some of the top biological processes down-regulated and reactome groups (Yu & He, 2016) up-regulated in the G0/G1-sorted cells. Full list in Datasets EV10 and EV11.

B Dendrogram of unbiased hierarchical clustering of gene expression from G0/G1-sorted U5-NSCs with the number genes up (green) and down (red) regulated (FDR < 0.05) in each KO compared to NTC. Complete results in Dataset EV10.

C Heat map of log$_2$FC compared to NTC for key genes changed in G0/G1 in following loss of *TP53*, *NF2/PTPN14*, *TAOK1*, and/or *CREBBP*, including genes from TP53 targets, YAP targets, the cell cycle, Hippo signaling, and electron transport genes. White dots indicate FDR < 0.05.

D, E Significance of overlap of the down (D)- and up (E)-regulated genes from bulk RNA sequencing of G0/G1-sorted cells with the single-cell cluster definitions (up-regulated genes). Significance assessed through hypergeometric enrichment test *P*-values. RF = representation factor.

F Fraction of cells classified by ccAF into the cell cycle phases for scRNA-seq profiles from CDT$^+$ sorted WT and sgTAOK1 U5-NSCs.

G Comparison of the proportions of each cell cycle phase between CDT$^+$ sorted WT and sgTAOK1 U5-NSCs. Significance was assessed using the proportion test. *≤ 0.05; **≤ $1 \times 10^{-6}$.

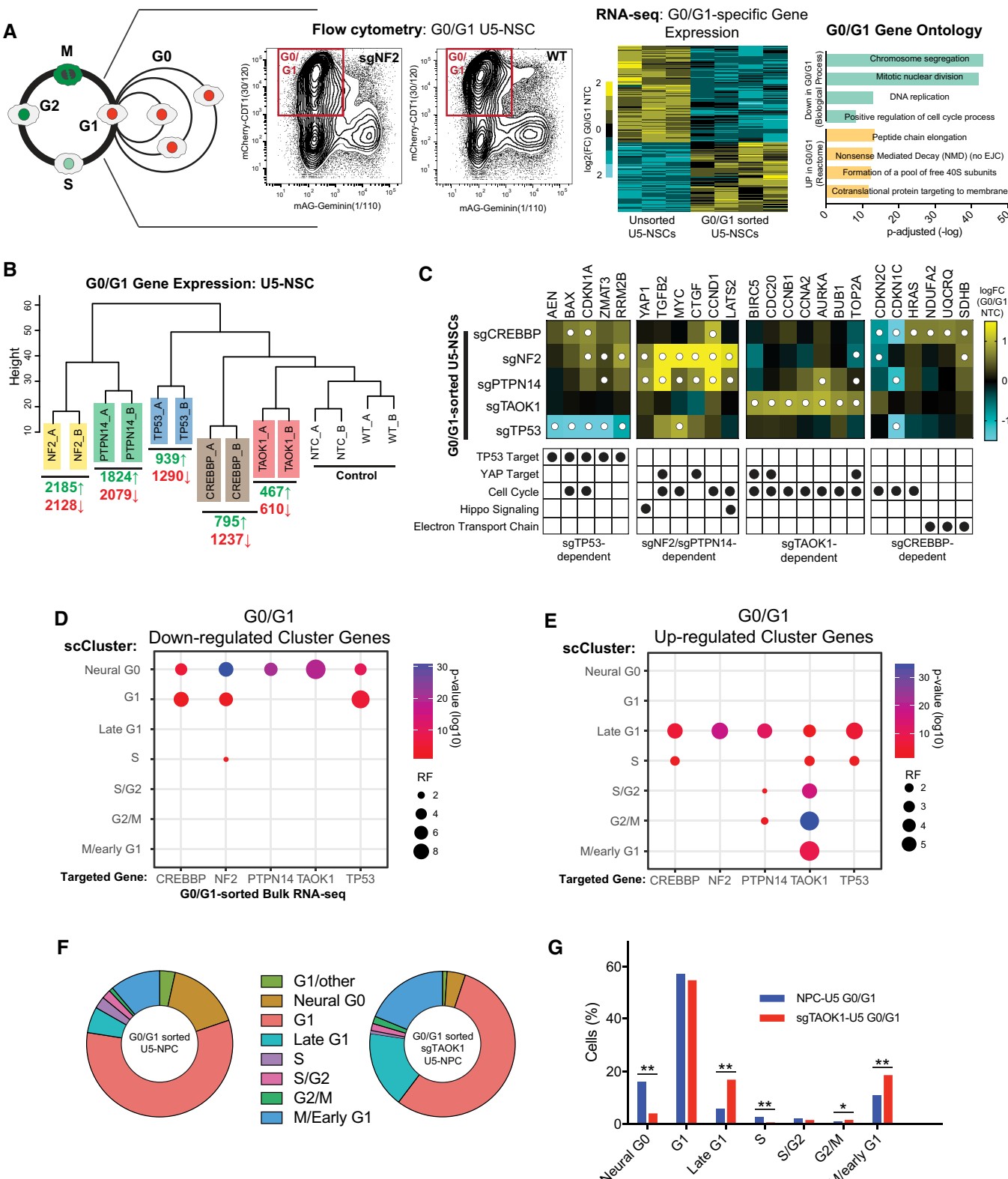

**Figure 7.**

subpopulation and reprogram G1 transcription networks to promote entry into G1-S.

## Discussion

We used scRNA-seq profiling and functional genomics screens to understand a fundamental difference between the *in vitro* self-renewal pattern of hNSCs and hGSCs. NSCs display a slower doubling rate due to a slower and variable length transit through G0/G1 even though hNSCs and hGSCs are isolated and grown in the same defined culture conditions. The rest of the cell cycle timing is uniform (as shown in the cell cycle phase time analysis of Fig 6B). By contrast, the GSCs have a uniform transit time through each phase of the cell cycle, including G0/G1, which results in a faster doubling rate. This result is perhaps not surprising given the known roles of oncogenic drivers to effect entry into the cell cycle (Hanahan & Weinberg, 2000, 2011). However, we probed this difference by transcriptionally resolving the NSC cell cycle into seven phases using scRNA-seq: G1, Late G1, S, S/G2, G2/M, M/Early G1, and a quiescence-like state Neural G0. We found that Neural G0 is highly enriched for markers of adult NSC quiescence. Through phenotypic assays and identification of fast growing "G0-skip" mutants, we determined that it is NSCs' ingress into and variable egress out of Neural G0 that determines the length of their cell cycle. Thus, Neural G0 is a transient quiescent state, which is diminished in GSCs *in vitro* (i.e., grade IV glioma isolates).

The scRNA-seq profiling of NSCs demonstrated that the current gold standard scRNA-seq cell cycle classifier (i.e., ccSeraut) did not adequately account for our *de novo* cell clusters including, Neural G0. Therefore, we created a new ccAF cell cycle classifier using a neural network-based approach. We validated the classifier by accurately classifying gold standard studies for Neural G0, S, and M phases of the cell cycle. The new classifier better accounts for our hNSCs cell cycle phases as judged by RNA velocity, gene expression vectors, and cyclin/CDK expression. It also better represents cell cycle phases in non-neuroepithelial-derived cell types, including HeLa and 293T cells, where Neural G0 subpopulations are absent. Moreover, ccAF accurately resolved populations of quiescent and activated adult NSCs from scRNA-seq data. The classifier also identified candidate Neural G0 populations among neural progenitors during fetal brain development, which generally diminish during differentiation. Finally, we have made the ccAF classifier available in a variety of useful forms (see Data Availability). Thus, ccAF is a useful tool for scRNA-seq classification of neuroepithelial- and non-neuroepithelial-derived cell types and for identifying novel subpopulations in a variety of biological contexts in actively dividing cell populations.

Application of ccAF to human glioma single-cell and bulk transcriptome profiles also revealed exciting insights into the structure of low- and high-grade glioma tumor populations. First, we again observed that ccAF does a better job at classifying cell cycle subpopulations for glioma than ccSeraut. The ccAF can classify G0/G1 populations into Neural G0, G1, and M/Early G1 across different developmental subtypes. Second, ccAF and Neural G0 expression patterns revealed a general trend that less aggressive grade II and III tumors have higher proportions of Neural G0 categorized cells than grade IV GBMs. Moreover, increased expression of Neural G0 genes

was associated with better patient prognosis, negatively correlated with the proliferative state in gliomas, and was independent of tumor grade and *IDH1/2* mutation status. Additionally, the Neural G0 state was shown to account for survival variance that is independent from active cell cycling, which means that the Neural G0 state is not simply the antithesis of active cell cycle states. Instead, the Neural G0 state has novel biological mechanisms regulating flow into and out of the G0 state that go beyond the biology of the active cell cycle. These results are consistent with Neural G0 acting as a barrier to progression in low-grade gliomas by promoting a longer pause between cell cycles, which is overcome in secondary gliomas.

In GBM tumors, the Neural G0 subpopulation contained putative glioma stem-like cells (as revealed by the scheme derived from Bhaduri *et al*, 2020), which represent 9.6% of the total tumor population. The mesenchymal subpopulation had the fewest Neural G0 classified cells (~40%), which is still a significant portion. These results are consistent with Neural G0 cells acting as a stem cell reservoir for non-mesenchymal subtypes, while mesenchymal/Neural G0 co-classified cells may capture cells that are in the process of undergoing proneural-to-mesenchymal transitions (Bhat *et al*, 2013; Halliday *et al*, 2014; Segerman *et al*, 2016). Future studies are warranted to determine whether the Neural G0 classified subpopulation contains terminally differentiated neoplastic cells, as it is difficult to assess given that tumor driver genes tend to interfere with lineage commitment.

The Neural G0 state is not exclusive to the neuroepithelial lineage (i.e., astrocytes, OPCs, RGs, and glioma cells). Instead, each Neural G0 cell is enriched for a portion, but not all, of the 158 genes present in the hNSCs' Neural G0, which helps distinguish it from G1 and other cell cycle phases. Thus, Neural G0 represents a mixed state that incorporates elements of qNSC and other neural progenitors, which likely results from the multipotency of fetal hNSCs combined with the effects of their *ex vivo* culture environment. G0-like states for non-neuroectoderm cells might be identified using an alternative set of developmental markers (e.g., Mesoderm G0).

With regard to the function, one possibility is that Neural G0 provides a compartment for the maintenance of neurodevelopmental potential. That is, it could allow time for reinforcing transcriptional and epigenetic programs associated with neurodevelopment gene expression. Consistent with this possibility, Neural G0 genes are up-regulated in quiescent NSCs *in vivo* and diminished during neural differentiation programs during corticogenesis or by KO of G0-skip genes in CDT[+] NSCs. Moreover, multiple Neural G0 genes significantly enriched in NSCs and glioma Neural G0 cells are known to help maintain "stemness". For example, *HEY1* and *TTYH1*, are both key players in the Notch signaling pathway in NSCs and help maintain the NSC identity *in vivo* (Kim *et al*, 2018; Than-Trong *et al*, 2018). *PTN* and its target *PTPRZ1* also may help promote stemness, signaling, and proliferation of neural progenitors and glioma tumor cells (Fujikawa *et al*, 2016, 2017; Zhang *et al*, 2016b). Moreover, *FABP7* expression and activity have been associated with lipid metabolism in slow-cycling GBM tumor cells (Hoang-Minh *et al*, 2018), consistent with Neural G0 state. Other functions for Neural G0 could include time for repair of DNA lesions that persist from the previous cell cycle (Arora *et al*, 2017; Barr *et al*, 2017), oxidative stress/mitochondrial maintenance (Mohrin & Chen, 2016), or regulation of structural RNAs (e.g., rRNAs, tRNAs)

(Roche *et al*, 2017). Future studies will be required to address these and other possibilities.

Lastly, we found that KO of five genes, *CREBBP, NF2, PTPN14, TAOK1,* or *TP53,* diminish Neural G0 *in vitro* in hNSCs. Gene expression changes in G0/G1 populations of KOs confirmed a reduction of Neural G0 genes and characteristic gene expression changes associated with the p53 transcriptional network, Hippo-YAP targets, cell cycle gene regulation, and many novel targets and pathways, including those downstream of *CREBBP* and *TAOK1.* Interestingly, in glioma, Hippo-Yap pathway activity has been shown to significantly increase with grade and is associated with shortened patient survival (Orr *et al*, 2011; Zhang *et al*, 2016a). Moreover, proneural tumors exhibit the lower Hippo-Yap pathway activity while mesenchymal tumors, the highest (Orr *et al*, 2011; Guichet *et al*, 2018). These data fit well with this pathway diminishing Neural G0 gene expression to promote a mesenchymal transition in more aggressive GBM cells (Bhat *et al*, 2013; Halliday *et al*, 2014; Segerman *et al*, 2016). However, it is less clear whether p53 would have a similar role in promoting G0-like states in tumors. *TP53* is among the most frequently altered genes in lower grade gliomas (26–74%) and in GBM (~30%) tumors (TCGA data; cbioportal). There are many examples of p53-independent pathways that regulate G0 ingress/egress in tumor contexts (e.g., Chen *et al*, 2012; Brown *et al*, 2017).

Consistent with this possibility, p27, but not p53-inducible p21, expression is significantly associated with longer-term survival in gliomas (Kirla *et al*, 2003). Thus, in *in vitro* hNSCs, low-level cellular stresses or DNA damage may trigger partial p53 activation and a transient p21-dependent G0-like state via *CDK2* inhibition, as has been reported for other cell types (Spencer *et al*, 2013). Regardless of whether p53 functions in this capacity *in vivo*, other pathways affecting G0 ingress/egress (e.g., microenvironmental signaling and transcriptional gene network pathways) will ultimately converge on the same set of regulatory events affecting cell cycle engine activity (e.g., raising or lowering *CyclinE/A/CDK2* activity). Thus, our results have relevance as a model of G0-like states and Neural G0 gene expression. Further, other G0-skip genes *CREBBP, NF2, PTPN14,* and *TAOK1* function independently of p53 (since they do not affect p53 target genes) and, thus, when mutated attenuate G0 through other mechanisms, including affecting transcription of key cell cycle targets (e.g., *CCNA2, CCND1, CDKN2C,* and *MYC*). Future studies will be required to address how these genes and pathways might affect G0-like states in NSCs and tumors.

Collectively, our data reveal Neural G0 is a cellular state shared by multiple neural epithelial-derived stem and progenitor cell types, which likely plays key roles in neurogenesis and glioma tumor development and recurrence.

# Materials and Methods

### Reagents and Tools table

| Reagent/Resource | Reference or source | Identifier or catalog number |
|---|---|---|
| **Experimental models** | | |
| U5-NSC human fetal neural stem cells | Jackson Lab | B6.129P2Gpr37tm1Dgen/J |
| 0131 GSC human adult glioma stem cells | Son et al (2009) Cell 4: 440–452 | 0131 |
| 0827 GSC human adult glioma stem cells | Son et al (2009) Cell 4: 440–452 | 0827 |
| **Recombinant DNA** | | |
| mCherry-CDT1(aa30–120) | Dr. Atsushi Miyawaki | FUCCI |
| mAG-Geminin(aa1–110) | Dr. Atsushi Miyawaki | FUCCI |
| **Antibodies** | | |
| Anti-CREBBP (WB, 1:500) | Cell Signaling | 7389 |
| Anti-NF2 (WB, 1:200) | Santa Cruz | SC-332 |
| Anti-Beta-Actin (WB,1:1,000) | Cell Signaling | 3700 |
| Anti-H4 (WB, 1:2,000) | Abcam | 17036-100 |
| Anti-phosphorylated RB (Ser807/811) (IF, 1:1,600) | Cell Signaling | 8516 |
| Anti-Rabbit AF647 (2°, IF, 1:200) | Fisher | A21245 |
| Anti-GFAP (IF, 1:1,500) | Millipore | AB5804 |
| Anti-β-tubulin III (TUJ1) (IF, 1:400) | Chemicon | MAB1637 |
| Anti-Nestin (IF, 1:250) | Santa Cruz | sc-23927 |
| Anti-Sox2 (IF, 1:200) | Cell Signaling | 3579S |
| Anti-Mouse AF488 (2° IF, 1:200) | Fisher | A11001 |
| Anti-Rabbit AF568 (2° IF, 1:200) | Fisher | A11011 |

**Reagents and Tools table** (continued)

| Reagent/Resource | Reference or source | Identifier or catalog number |
|---|---|---|
| **Oligonucleotides and other sequence-based reagents** | | |
| PCR primers | This study | Table EV9 |
| Guide sequences | This study | Table EV9 |
| **Chemicals, Enzymes and other reagents** | | |
| Chromium Next GEM Single Cell 3′ kit | 10X Genomics | CAT#1000269 |
| **Software** | | |
| CellRanger | https://support.10xgenomics.com/single-cell-gene-expression/software/pipelines/latest/what-is-cell-ranger | scRNA-seq Alignment and QC |
| Seurat v2.3.4 (R) | https://satijalab.org/seurat/install.html#previous | scRNA-seq analysis |
| Seurat v3.1.2 (R) | https://satijalab.org/seurat/install.html#cran | scRNA-seq analysis |
| scanpy v1.5.1 (Python) | https://scanpy.readthedocs.io/en/stable/ | scRNA-seq analysis |
| **Other** | | |
| Chromium Controller | 10× Genomics | scRNA-seq |
| TapeStation | Agilent | Library QC |
| Qubit 2.0 Fluorometer | Fisher | Library QC |
| HiSeq 2500 | Illumina | Sequencing |

## Methods and Protocols

### Cell culture

The U5 fetal human NSC line (Bressan *et al*, 2017) and adult 0131-mesenchymal and 0827-proneural human GSC lines (Son *et al*, 2009) were grown in NeuroCult NS-A basal medium (StemCell Technologies) supplemented with B27 (Thermo Fisher), N2 (2× stock in Advanced DMEM/F-12 (Fisher) with 25 μg/ml insulin (Sigma), 100 μg/ml apo-Transferrin (Sigma), 6 ng/ml progesterone (Sigma), 16 μg/ml putrescine (Sigma), 30 nM sodium selenite (Sigma), and 50 μg/ml bovine serum albumin (Sigma), and EGF and FGF-2 (20 ng/ml each) (Peprotech) on laminin (Sigma or Trevigen)-coated polystyrene plates and passaged according to previously published protocols (Pollard *et al*, 2009). Cells were detached from their plates using Accutase (Thermo Fisher). 293T (ATCC) cells were grown in 10% FBS/DMEM (Invitrogen).

### Flow cytometry

FUCCI constructs (RIKEN, gift from Dr. Atsushi Miyawaki) were transduced into wild-type U5-NSCs and sorted sequentially for the presence of mCherry-CDT1(aa30–120) and S/G2/M mAG-Geminin(aa1–110) on an FACSAria II (BD). Normal growth was verified post-sorting and then the FUCCI U5-NSCs were transduced with individual sgRNA-Cas9 (4 independent guides per gene) and selected with 1 μg/ml puromycin. Cells were grown out for 21 days with splitting every 3–4 days and maintaining equivalent densities. Cells were counted (Nucleocounter NC-100; Eppendorf) and plated 3 days before analysis on an LSR II (BD). Controls cultured in the same conditions included cells transduced with guides against 3 non-growth limiting genes, including *GNAS1*, and showed equivalent FUCCI ratios. Results were analyzed using FlowJo software.

### Single-cell RNA sequencing sample preparation

Single-cell RNA sequencing was performed using 10× Genomics' reagents, instruments, and protocols. scRNA-seq libraries were prepared using GemCode Single Cell 3′ Gel Bead and Library Kit. FUCCI U5-NSCs (both with and without lentiviral TAOK1 KO, > 14 days outgrowth) were harvested and half the cells were sorted using the FACSAria II (BD) for cells singly positive for mCherry-CDT1 FUCCI. Sorted cells were kept on ice before suspensions were loaded on a GemCode Single Cell Instrument to generate single-cell gel beads in emulsion (GEMs) (target recovery: 2,500 cells). GEM-reverse transcription (RT) was performed in a C1000 Touch Thermal cycler (Bio-Rad), and after RT, GEMs were broken and the single-strand cDNA was cleaned up with DynaBeads (Fisher) and SPRIselect Reagent Kit (Beckman Coulter). cDNA was amplified, cleaned up, and sheared to ~200 bp using a Covaris M220 system (Covaris). Indexed sequencing libraries were constructed using the reagents in the GemCode Single Cell 3′ Library Kit, following these steps: (i) end repair and A-tailing; (ii) adapter ligation; (iii) post-ligation cleanup with SPRIselect; and (iv) sample index PCR and cleanup. Library size distributions were validated for quality control using a 2200 TapeStation (Agilent). The barcoded sequencing libraries were quantified by a Qubit 2.0 Fluorometer (Fisher) and sequenced using HiSeq 2500 (Illumina) with the following read lengths: 98 bp Read1, 14 bp I7 Index, 8 bp I5 Index, and 10 bp Read2. Sequencing data can be accessed at the NCBI Gene Expression Omnibus (GSE117004).

### Discovery of cell cycle phases from U5-hNSC scRNA-seq profiles

CellRanger (10× Genomics) was used to align, quantify, and provide basic quality control metrics for the scRNA-seq data. Using Seurat version 2.3.0, the scRNA-seq data from wild-type U5 cells and sgTAOK1 knockout cells were merged and analyzed. Both

scRNA-seq data were loaded as counts, normalized, and then scaled while taking into account both percent of mitochondria and the number of UMIs per cell as covariates. The union of the top 1,000 most variant genes from each dataset was used in canonical correlation analysis (CCA) to merge the two datasets via alignment of their subspace. Clusters of cells were identified using a shared nearest neighbor (SNN) modularity optimization-based clustering algorithm. Marker genes for each cluster were identified as differentially expressed genes, and the determination of 8 clusters was based on the discovery of strong markers for 6 of the eight clusters (both the G1 and G1/other clusters did not have significantly up-regulated marker genes).

### Determining the identity of the cell cycle phases

The identity of clusters was determined primarily through the expression of cyclins and cyclin-dependent kinases and secondarily through the function of other marker genes. A tSNE visualization was generated with a perplexity setting of 26. Functional enrichment was calculated using the TopGO package in R for GO biological process terms, and significant associations were identified with BH-corrected $P$-values ≤ 0.05. CycleBase 3.0 genes with cell cycle arrest phenotypes when a gene is knocked down or knocked out for each arrested phase (G0/1 arrest = CMPO:0000173, S arrest = CMPO:0000204, G2 arrest = CMPO:0000203, and M arrest = CMPO:0000196) were intersected with the U5-hNSC cell cycle cluster marker gene lists. The cell cycle classifying CellCycleScoring function in Seurat (ccSeurat) was used to infer the cell cycle state of all U5-hNSCs for comparison. Significance was assessed using the hypergeometric distribution, and $P$-values were corrected for multiple hypothesis tests with the Benjamini–Hochberg FDR method. Significant enrichments were identified with intersected gene lists greater than 0 and with BH-corrected $P$-values ≤ 0.05.

### Resolving the flow of cells through the cell cycle

Network analyses were used to determine the connections between the phases of the cell cycle. First, the cluster medioids (mean expression for each gene across all the cells from a cluster) were used to compute the Canberra distance measure. In a cycle like a cell cycle, it is expected that on average there will be 2 edges between each cell cycle state. A distance cutoff of 240 led to 2.28 connections per cluster was used to turn the distance matrix into a network (Futreal *et al*, 2004).

RNA velocity was used to determine the trajectories of cells through the two-dimensional tSNE embedding using the Python packages scanpy and scVelo (Bergen *et al*, 2020). A loom file was exported from the U5 hNSC Seurat object and imported into Python using scanpy and scVelo. The RNA velocity was computed while grouping the cells based on the cell cycle phases. Then, a velocity graph was inferred, and the velocity was plotted as RNA velocity streamlines overlaid onto the tSNE embedding.

### Building the ccAF cell cycle classifier

The top 1,536 most variant genes (Ensembl gene identifiers) from the integrated U5 dataset were used to train each classifier. The ccAF classifier was evaluated using 100-fold cross-validation using a hold-out of 1,000 cells for each iteration. Comparisons were made between classification methods using F1 scores (where an F1 score is a metric that integrates precision and recall and reaches

its maximum value at 1), and error rates were computed using scikit-learn in Python after each iteration. The classification methods tested were all Python-based and included (i) support vector machine with reject option (SVMrej; classification cutoff ≥ 0.7), a general-purpose classifier from the scikit-learn library; (ii) random forest (RF), a general-purpose classifier from the scikit-learn library; (iii) k-nearest neighbor (KNN) from the scanpy ingest method (Wolf *et al*, 2018); and (iv) neural network (NN) ACTINN (Ma & Pellegrini, 2020). Sensitivity analyses were performed by randomly excluding a defined percentage of classifier genes, conducting 100-fold cross-validation, and recording the error rate after each iteration.

### Validating S and M phase ccAF classifications using a gold standard dataset

The ccAF classifier was evaluated using a gold standard dataset of 1,134 most cyclic genes from Whitfield *et al*, 2002 (Whitfield *et al*, 2002) (http://genome-www.stanford.edu/Human-CellCycle/HeLa/). The DNA microarray expression data were quantile normalized, the ccAF classifier was applied, and F1 scores and error rates were computed using the scikit-learn library in Python.

### Validating the G0 state using gold standard scRNA-seq profiling studies

Three different gold standard mouse studies (Llorens-Bobadilla *et al*, 2015, GSE67833; Dulken *et al*, 2017, PRJNA324289) that experimentally determined G0 state using flow-sorting on established G0 markers were classified using ccAF. First, each dataset was converted from mouse Ensembl gene IDs to human Ensembl gene IDs using homology. The classifier was applied and then confusion matrices and statistics were computed that allowed comparison of the experimental and predicted G0 states.

### Application of ccAF to neuroepithelial and other scRNA-seq profiling studies

The ccAF classifier was applied to neuroepithelial development scRNA-seq profiles from Nowakowski *et al*, 2017 (http://bit.ly/cortexSingleCell). It was also applied to HEK293T cells from a barnyard assay conducted by 10× (https://support.10xgenomics.com/single-cell-gene-expression/datasets/3.0.2/5k_hgmm_v3_nextgem) to assess the ability to apply the classifier to non-neuroepithelial scRNA-seq profiles. A total of 3,468 HEK293T cells were selected from the barnyard based on the expression of human transcripts and no expression of mouse transcripts. The classifier was applied and then confusion matrices and statistics were computed that allowed comparison of the experimental and predicted G0 states.

### Application of ccAF to tumors

The ccAF was applied to seven studies of glioma primary patient tumor scRNA-seq: (i) grade II oligodendrogliomas that are IDH1 mutant from Tirosh *et al*, 2016—GSE70630; (ii) grade III astrocytomas that are IDH1 mutant from Venteicher *et al*, 2017—GSE89567; (iii) grade IV glioblastomas that are IDH wild type from Darmanis *et al*, 2017—GSE84465; (iv) grade IV glioblastomas that are IDH wild type from Neftel *et al*, 2019—GSE131928; (v) grade IV glioblastomas that are IDH wild type from Bhaduri *et al*, 2020—PRJNA579593; (vi) grade IV glioblastomas that are IDH wild type from Wang *et al*, 2020—GSE139448; (vii) diffuse midline glioma

with H3K27M from Filbin *et al*, 2018—GSE102130. We also applied the classifier to head and neck squamous cell carcinoma (HNSCC) tumors from Puram *et al*, 2017—GSE103322. scRNA-seq profiles and patient meta-data were collected, normalized and scaled if necessary, filtered to only neoplastic cells, and saved as loom files to be loaded into scanpy for classification.

Each GBM dataset went through this analysis pipeline:

1 Raw count data were used when available otherwise processed data were used. Data were input into Seurat V3 and the standard scRNA-seq analysis pipeline was applied and, if necessary, normalization and scaling were performed. The quality of each cell was assessed for studies starting with count data by plotting the number of UMIs/counts per cell versus percentage of mitochondrial gene expression. Cutoffs were determined from these plots to remove damaged cells or barcodes mapping to more than one cell.

2 Meta-data were loaded into R and imported into the Seurat meta.data object for later comparisons.

3 A UMAP embedding and *de novo* clustering were applied to all cells from each dataset.

4 Neoplastic cells were enriched for by removing cells that belonged to a *de novo* cell cluster expressing genes from a terminally differentiated cell type: oligodendrocytes (MBP and PLP1) (Valério-Gomes *et al*, 2018); astrocytes (ETNPPL) (Zhang *et al*, 2016c); neurons (RBFOX) (Herculano-Houzel & Lent, 2005); or immune cells (AIF1, CD14, CX3CR1, PTPRC).

5 Neoplastic cell identify was validated using the inferCNV algorithm applied with the terminally differentiated cells defined in step 4 as the reference (Patel *et al*, 2014).

6 The GBM subtypes (CL = classical, MS = mesenchymal, and PN = proneural) were inferred using single sample GSEA for GBM (ssGSEA.GBM) from Wang *et al*, 2017 (Wang *et al*, 2017). The subtype calls for each cell were loaded up into Seurat object meta.data.

7 A loom file was written out for each glioma Seurat object so that the data could be loaded into Python where the ccAF classifier could be applied.

8 The loom file was loaded up into scanpy in Python. If necessary gene IDs were converted into Ensembl human IDs.

9 The ccAF classifier was applied to each glioma dataset and stored back into the scanpy object.

10 Downstream analysis integrating ccAF predictions with GBM subtype and other analyses were then conducted.

### Discovering GBM neoplastic cell-specific Neural G0 marker genes

The ccAF-predicted cell cycle states were imported back into the Seurat objects for each of the four GBM scRNA-seq profiling studies. The T-statistic-based differentially expressed marker gene discovery function in Seurat was used to identify genes upregulated in Neural G0 relative to all other ccAF cell cycle states (average log fold-change $\geq 0.3$; FDR adjusted *P*-value $\leq 0.05$; Dataset EV5). The marker gene lists for each GBM scRNA-seq profiling study were compared and the marker genes common across all four studies are reported. Gene–gene association networks were generated and visualized using the GeneMANIA webtool (Warde-Farley *et al*, 2010). Previously, GBM-associated genes from the DisGeNET database (Piñero *et al*, 2015) were

selected based on the same criteria as (Plaisier *et al*, 2016). The previously GBM-associated genes were intersected with GBM neoplastic cell-specific Neural G0 marker genes, and the significance of the enrichment of the overlap was assessed using a hypergeometric *P*-value.

### Classifying putative GBM stem-like cells in GBM tumors

Putative GBM stem-like cells were classified using the logic provide by Bhaduri *et al*, 2020: expressing PROM1, FUT4, or L1CAM, in conjunction with SOX2, and not expressing TLR4. This was formulated into a logical expression for application to scRNA-seq count data: (FUT4 > 0 or L1CAM > 0 or PROM1 > 0) and SOX > 0 and TLR4 == 0. This logic was applied to every cell in the GBM studies, and cells passing this logic were classified as putative GBM stem-like cells. Significance of overlap between the putative GBM stem-like cells and the ccAF classified cell cycle states was computed using the hypergeometric test.

### Deriving eigengenes for Neural G0 and the cell cycle and association with patient survival

Gene expression matrices of Neural G0 marker genes and cell cycle genes (G2M genes from ccSeurat) were summarized into a single vector using the first principal component corrected for sign, which is referred to as the eigengene. Eigengenes for Neural G0 marker genes and cell cycle genes were calculated across all 688 low-grade glioma (LGG) and GBM patient tumors from The Cancer Genome Atlas (Brennan *et al*, 2013). A Pearson correlation in R was used to compare the Neural G0 and cell cycle eigengenes. Patient survival and the commonly mutated IDH1 and IDH2 gene mutations were collected into a meta-data matrix for survival analyses. Survival analyses were conducted in R using the survival package with the following conditions:

1 The Cox proportional hazards regression method was used to determine whether Neural G0 eigengene was still a significant predictor of patient survival with the covariates tumor grade and IDH1/2 status.

2 The top 25% and bottom 25% of patient expression for all patients based on Neural G0 eigengene were compared against patient survival using Kaplan–Meier plot and a G-ρ Harrington–Fleming *P*-value. Increased survival time was calculated based on the difference in observed time to 50% survival and converted to years.

3 The top 25% and bottom 25% of patient expression for only patients with grade III tumors based on Neural G0 eigengene were compared against patient survival using Kaplan–Meier plot and a G-ρ Harrington–Fleming *P*-value. Increased survival time was calculated based on the difference in observed time to 50% survival and converted to years.

### CRISPR-Cas9 screening

For large-scale transduction, NSC cells were plated into T225 flasks at an appropriate density such that each replicate had 250–500-fold representation, using the two previously published CRISPR-Cas9 libraries (Shalem *et al*, 2014; Doench *et al*, 2016) (Addgene) or a custom-synthesized sgRNA library (Twist Biosciences) targeting 1,377 genes derived from (Toledo *et al*, 2015). NSCs and GSCs were infected at MOI < 1 for all cell lines. Cells were infected for

48 h, followed by selection with 1–2 µg/ml (depending on the target cell type) of puromycin for 3 days. Post-selection, a portion of cells were harvested at Day 0 time point. The remaining cells were then passaged in T225 flasks maintaining 250–500-fold representation and cultured for an additional 21–23 days (~10–15 cell doublings) or 10 days. Genomic DNA was extracted using QiaAmp Blood Purification Mini or Midi kit (Qiagen). A two-step PCR procedure was performed to amplify sgRNA sequence. For the first PCR, DNA was extracted from the number of cells equivalent to 250–500-fold representation (screen-dependent) for each replicate (2–4 replicates) and the entire sample was amplified for the guide region. For each sample, ~100 separate PCRs (library and representation dependent) were performed with 1 µg genomic DNA in each reaction using Herculase II Fusion DNA Polymerase (Agilent) or Phusion High-Fidelity DNA Polymerase (Thermo Fisher). Afterward, a set of second PCRs was performed to add on Illumina adaptors and to barcode samples, using 10–20 µl of the product from the first PCR. Primer sequences are in Dataset EV10. A primer set was used to include both a variable 1–6 bp sequence to increase library complexity and 6 bp Illumina barcodes for multiplexing of different biological samples. The whole amplification was carried out with 12 cycles for the first PCR and 18 cycles for the second PCR to maintain linear amplification. The resulting amplicons from the second PCR were column-purified using Monarch PCR & DNA Cleanup Kit (New England Biolabs; NEB) to remove genomic DNA and first-round PCR product. Purified products were quantified (Qubit 2.0 Fluorometer; Fisher), mixed, and sequenced using HiSeq 2500 (Illumina). Bowtie was used to align the sequenced reads to the guides (Langmead *et al*, 2009). The R/ Bioconductor package edgeR was used to assess changes across various groups (Robinson *et al*, 2010). For the tiling library, only guides that mapped once to the genome and are within the gene's coding region were considered for further analysis.

Raw and mapped data files are available at the Gene Expression Omnibus database (GSE117004).

### Individual lentiviral-sgRNA assembly for validation

For retests, individual or pooled sgRNA were cloned into lentiCRISPR v2 plasmid. Briefly, DNA oligonucleotides were synthesized with sgRNA sequence flanked by the following:

5′: tatatcttGTGGAAAGGACGAAACACCg
3′: gttttagagctaGAAAtagcaagttaa

PCR was then performed with the ArrayF and ArrayR primers (Dataset EV10). The PCR product was gel-purified using the ZymoClean Gel DNA recovery kit (Zymo Research). Gibson Assembly Master Mix (NEB) was used to clone the PCR product into lentiCRISPR v2 plasmid (Sanjana *et al*, 2014). The ligated plasmid was then transformed into Stellar Competent cells (Clontech) and streaked onto LB agar plates. The resulting clones were grown up and sequence verified (GeneWiz).

### Lentiviral production

For virus production, lentiCRISPR v2 plasmids (Sanjana *et al*, 2014) were transfected using polyethylenimine (Polysciences) into 293T cells along with psPAX and pMD2.G packaging plasmids (Addgene) to produce lentivirus. Lentivirus particles for the

whole-genome CRISPR-Cas9 libraries were produced in 25 × 150 mm plates of 293T cells seeded at ~15 million cells per plate. Fresh media was added 24 h later and viral supernatant was harvested 24 and 48 h after that. For screening, virus was concentrated 1,000× following ultracentrifugation at 6,800× *g* for 20 h. For validation, lentivirus was used unconcentrated at an MOI < 1.

### Viability and proliferation assays

Cells were infected with lentiviral gene pools containing 3–4 sgRNAs per gene or with lentivirus containing a single sgRNA to the respective gene (Dataset EV10). Initial cell density was carefully controlled for each experiment by counting cells using a Nucleocounter NC-100 (Eppendorf) and cells were always grown in subconfluent conditions. For viability assays, following selection, cells were outgrown for 7–10 days, harvested, counted, and plated in triplicate onto 96-well plates coated with laminin in dilution format starting at 1,000 cells to 3,750 cells per well (cell density depended on cell isolate and duration of assay). Cells were fed with fresh medium every 3–4 days. After 7–12 days under standard growth conditions, cell proliferation rates were measured using Alamar blue reagent according to the manufacturer's instructions (Invitrogen). For analysis, sgRNA-containing samples were normalized to their respective non-targeting control (NTC) samples. For doubling time assays, cells infected with individual sgRNAs or NTC were routinely cultured (split every 3–5 days) and counted at each split (Nucleocounter NC-100; Eppendorf). The overall growth of each well containing an individual sgRNA was calculated and compared to the NTC well. Comparisons between multiple experiments were normalized.

### Competition experiment

NSCs were infected with lentiviral gene pools containing 3–4 sgRNAs per gene, puromycin-selected, and mixed with NSCs infected with lentiviruses containing turboGFP at an approximate 1:9 ratio, respectively. Cultures were outgrown for 23–31 days, and flow analysis (FACS Canto; Becton Dickinson) was conducted every 7–8 days for GFP expression. Flow analysis data were analyzed using FlowJo software. For each sample, the GFP- population for each time point was normalized to its respective Day 0 GFP- population and the NTC (competition index).

### Time-lapse microscopy

U5-hNSCs were infected with lentiviral gene pools containing 3–4 sgRNAs per gene or with individual sgRNAs, puromycin selected, outgrown for > 13 days, and plated onto 96-well plates or 24-well plates. Plates were then inserted into the IncuCyte ZOOM (Essen BioScience), which was in an incubator set to normal culture conditions (37° and 5% $CO_2$), and analyzed with its software. For the cell confluency experiment, phase images were taken every hour for 72 h. For the FUCCI cell cycle experiment, images were taken every 10–15 min for 72–120 h. Cell cycle transit time for G0/G1 (mCherry-CDT1(aa30–120)[+]) and S/G2/M (mAG-Geminin(aa1–110)[+]) was manually scored by three different observers in actively dividing cells (those that could be followed from mitosis to mitosis). Each KO was scored by at least 2 independent observers and consistency between scorers was checked through shared analysis of a standard.

### Western blotting

Cells were harvested, washed with PBS, and immediately either lysed or snap-frozen and stored at −80°C until lysis. Cells were lysed with modified RIPA buffer (150 mM NaCl, 50 mM Tris, pH 7.5, 2 mM MgCl$_2$, 0.1% SDS, 2 mM DDT, 0.4% deoxycholate, 0.4% Triton X-100, 1X complete protease inhibitor cocktail (complete Mini EDTA-free, Roche) and 1 U/μl benzonase nuclease (Novagen) at room temperature for 15 min. Cell lysates were quantified using Pierce 660 nm protein assay reagent and proteins were loaded onto SDS-PAGE for Western blot. The Trans-Blot Turbo transfer system (Bio-Rad) was used according to the manufacturer's instructions. See Dataset EV10 for antibodies and dilutions. An Odyssey infrared imaging system was used to visualize blots (LI-COR) following the manufacturer's instructions.

### Immunofluorescence and CDK2 activity

U5-NSCs were plated on acid-washed glass coverslips (phosphory-lated Rb and CDK2 activity) or 96-well imaging plates (differentia-tion; Corning). They were fixed overnight in 2% paraformaldehyde (USB) at 4°C, washed with Dulbecco's phosphate-buffered saline (DPBS) (with calcium and magnesium) (Fisher), and blocked and permeabilized with 5% goat serum (Millipore), 1% bovine serum albumin (Sigma), and 0.1% Triton X-100 (Fisher) in DPBS for 45 min at room temperature. Samples were stained with primary antibody diluted in 5% goat serum in DPBS overnight at 4°C, washed with DPBS, and stained with secondary antibody (diluted 1:200 in 5% goat serum in DPBS) at 37°C for 45 min. See Dataset EV10 for antibodies and dilutions. Samples were washed with DPBS, dyed with 100 ng/ml 4′,6-diamidino-2-phenylindole (DAPI) diluted in DPBS for 20 min at room temperature, and washed with DPBS. Coverslips were preserved using ProLong Gold Antifade Mountant (Fisher) and inverted on glass slides. For differentiation, images were acquired on Nikon Eclipse Ti using NIS-Elements software (Nikon).

### Phosphorylated Rb and CDK2 activity image analysis

Cells were transduced with mVenus-DNA helicase B (DHB) (amino acids 994–1,087) (Hahn et al, 2009) (gift from Dr. Sabrina Spencer) and the mCherry-CDT1 FUCCI and sorted on a FACSAria II flow cytometer (BD). Cells were outgrown to ensure normal growth and then transduced with individual sgRNA-Cas9. After > 10 days outgrowth, cells were counted and plated, grown for 2 days, and stained for phosphorylated Rb and imaged on a TISSUEFAXS microscope (TissueGnostics), 54 fields per KO or NTC. Cells were analyzed using CellProfiler (Kamentsky et al, 2011). G0/G1 nuclei were identified by the presence of the CDT1 FUCCI reporter (25–120 pixel diameter, Global/Otsu thresholding, and distinguishing clumped objects by shape). CDK2 activity was defined by the cytoplasmic to the nuclear ratio of the mVenus-DHB reporter, with the cytoplasmic intensity of the DHB reporter defined as the upper quartile intensity of a 2-pixel ring around the CDT1-defined nucleus due to the irregular shape of the U5-NSCs.

### p27 reporter

The p27 reporter was constructed after Oki et al (2014), using a p27 allele that harbors two amino acid substitutions (F62A and F64A) that block binding to cyclin/CDK complexes but do not interfere with its cell cycle-dependent proteolysis. This p27K$^-$ allele was fused to mVenus to create p27K$^-$-mVenus. To this end, the p27 allele and mVenus were synthesized as gBlocks (IDT) and cloned via Gibson assembly (NEB) into a modified pGIPz lentiviral expression vector (Open Biosystems). Lentivirally transduced cells were puromycin-selected and validated using mCherry-CDT1 FUCCI and HDAC inhibitor treatment (48 h of 5 μM apicidin (Cayman)) to induce G0/G1 arrest using FACS (LSR II from Becton Dickinson and FlowJo software).

### Bulk RNA sequencing expression analysis

For G0/G1 NSC, cells singly positive for mCherry-CDT1 FUCCI were sorted on a FACSAria II (BD) directly into TRIzol reagent (Life Technologies). For differentiating cells, cells were sparsely plated and cultured with growth medium without EGF or FGF-2 for 7 days before being lysed with TRIzol reagent. For both, 2 replicates per condition were harvested. RNA was extracted using Direct-zol RNA MiniPrep Plus (Zymo Research). Total RNA integrity was checked and quantified using a 2200 TapeStation (Agilent). RNA-seq libraries were prepared using the KAPA Stranded mRNA-seq Kit with mRNA capture beads (KAPA Biosystems) according to the manufacturer's guidelines. Library size distributions were validated using a 2200 TapeStation (Agilent). Additional library QC, blending of pooled indexed libraries, and cluster optimization were performed using the Qubit 2.0 Fluorometer (Fisher). RNA-seq libraries were pooled and sequencing was performed using an Illumina HiSeq 2500 in Rapid Run mode employing a paired-end, 50 base read length (PE50) sequencing strategy.

### Bulk RNA sequencing data analysis

RNA-seq reads were aligned to the UCSC mm10 assembly using Tophat2 (Trapnell et al, 2012) and counted for gene associations against the UCSC gene database with HTSeq (Anders et al, 2015). Differential expression analysis was performed using R/Bioconductor package edgeR (Robinson et al, 2010). Samples for G0/G1 bulk RNA-seq were collected in two batches, so batch-dependent genes were removed before analysis (inter-batch P-value < 0.01 by Wilcoxon–Mann–Whitney). To ensure that no genes were eliminated that may be regulated specific to a particular knockout, genes with a CPM variability > 2-fold compared to the internal batch control and an expression greater than 1 CPM in at least one sample were retained. Differentially expressed genes (DEG) at the transcription level were found using a statistical cutoff of FDR < 0.05 and visualized using R/Bioconductor package pheatmap. Kolmogorov–Smirnov tests were conducted in R using the function ks.test from the stats package. Raw sequencing data and read count per gene data can be accessed at the NCBI Gene Expression Omnibus (GSE117004).

### Gene ontology analysis

Gene Ontology (GO)-based enrichment tests were implemented using GOseq (v 1.23.0) (Young et al, 2010), which corrects for gene length bias. Gene lists were also analyzed for pathways using the R/Bioconductor package ReactomePA (v 1.15.4) (Yu & He, 2016). The analysis used all genes either up or down-regulated with a FDR < 0.05 compared to NTC. GO terms with adjusted P-values < 0.05 were considered significantly enriched. Venn diagrams were generated on http://bioinformatics.psb.ugent.be/webtools/Venn/.

***Analyses of CDT[+] sorted scRNA-seq profiles from WT and sgTAOK1 KO***

The ccAF classifier was applied to CDT[+] sorted scRNA-seq profiles from WT and TAOK1 knockout (sgTAOK1). The scRNA-seq data were preprocessed and normalized as was done for the non-sorted scRNA-seq profiles, saved as loom files, and loaded into scanpy for classification by ccAF. The significance of differences in proportions was tested using the 2 population proportion test (prop.test) in R.

***Statistics and reproducibility***

Data are presented as the mean or median $\pm$ standard deviation (SD) or standard error of the mean (SEM), as specified in the Figure legends. Statistics were performed using GraphPad Prism 7.0 or analysis-specific functions in R and Python. All statistical tests are specified in Figure legends. The number of independent experiments is indicated in the Figures, Figure legends, or Methods. The significance of enrichment was assessed using the hypergeometric distribution, and $P$-values were corrected for multiple hypothesis tests with the Benjamini–Hochberg FDR method. Significant enrichments were identified with intersected gene lists greater than 0 and with BH-corrected $P$-values $\leq 0.05$.

# Data availability

The datasets and computer code produced in this study are available in the following databases:

- Overarching super series of all data in Gene Expression Omnibus—GSE117004 (https://www.ncbi.nlm.nih.gov/geo/query/acc.cgi?acc=GSE117004), which contains:

  a. U5 and U5-sgTAOK1 scRNA-seq data: Gene Expression Omnibus—GSE117003 (https://www.ncbi.nlm.nih.gov/geo/query/acc.cgi?acc=GSE117003)
  b. CRISPR-Cas9 knockout outgrowth screen in U5 sgRNA data: Gene Expression Omnibus—GSE117002 (https://www.ncbi.nlm.nih.gov/geo/query/acc.cgi?acc=GSE117002)
  c. U5 and knockout bulk RNA-seq data: Gene Expression Omnibus - GSE116970 (https://www.ncbi.nlm.nih.gov/geo/query/acc.cgi?acc=GSE116970)

- Data used to conduct analyses: figshare.com—Neural G0: a quiescent-like state found in neuroepithelial-derived cells and glioma (https://figshare.com/projects/Neural_G0_a_quiescent-like_state_found_in_neuroepithelial-derived_cells_and_glioma/86939)
- Code used to conduct analyses: github.com - U5_hNSC_Neural_G0 (https://github.com/plaisier-lab/U5_hNSC_Neural_G0)
- Walkthrough of code and analyses: pages.github.io U5_hNSC_Neural_G0 (https://plaisier-lab.github.io/U5_hNSC_Neural_G0/)

  Additionally, this work generated the ccAF classifier which is available in multiple forms:

- ccAF code: github.com – ccAF (https://github.com/plaisier-lab/ccAF)
- ccAF pypi.org package – ccAF (https://pypi.org/project/ccAF/1.0.1/)
- ccAF installed as a Docker image – cplaisier/ccaf (https://hub.docker.com/r/cplaisier/ccaf)

Expanded View for this article is available online.

## Acknowledgements

We thank Jon Cooper, Eric Holland, and members of the Paddison laboratory for helpful discussions and Atsushi Miyawaki and Sabrina Spencer for providing reagents. This work was supported by the following grants: Interdisciplinary Training in Cancer Fellowship NCI T32CA080416 (P.H.); NCI/NIH (R01CA190957; R21CA170722; P30CA15704) (P.P.); (5R21CA232244) (C.L.P.); NINDS/NIH (R01NS119650) (A.P., P.P., C.L.P.); DoD Translational New Investigator Award CA100735 (P.P.); and the Pew Biomedical Scholars Program (P.P.).

## Author contributions

Project conception and design were carried out by PJP, CLP, HMF, SAO, and CMT CRISPR-Cas9 screening was performed by HMF, CMT, and PH; screen analysis was performed by RB and JD; hit validation was performed by HMF, CMT, PH, and MK; critical reagents were generated by PC and LC; screen and RNA-seq data analysis and statistics were performed by SA with input from AP; scRNA-seq was performed by HMF under supervision of JLM-F and CT and analyzed by SAO and CLP; HB performed cancer mutation analysis; JM designed the tiling library; SMP provided and validated the hNSCs; and PJP, HMF, SAO, AP, and CLP wrote the manuscript with input from all authors.

## Conflict of interest

The authors declare that they have no conflict of interest.

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
