## [Review Process File · Molecular Systems Biology]

Neural G0: a quiescent-like state found in neuroepithelial-derived cells and glioma

Samantha O'Connor, Heather Feldman, Chad Toledo, Sonali Arora, Pia Hoellerbauer, Philip Corrin, Lucas Carter, Megan Kufeld, Hamid Bolouri, Ryan Basom, Jeffrey Delrow, José McFaline-Figueroa, Cole Trapnell, Steven Pollard, Anoop Patel, PATRICK PADDISON, and Christopher Plaisier
DOI: 10.15252/msb.20209522

Corresponding author(s): Christopher Plaisier (plaisier@asu.edu) , Christopher Plaisier (plaisier@asu.edu), PATRICK PADDISON (paddison@fredhutch.org)

Review Timeline:

Submission Date:	14th Feb 20
Editorial Decision:	15th Apr 20
Revision Received:	11th Aug 20
Editorial Decision:	28th Sep 20
Revision Received:	8th Feb 21
Editorial Decision:	9th Mar 21
Appeal Received:	31st Mar 21
Editorial Decision:	19th Apr 21
Revision Received:	30th Apr 21
Accepted:	14th May 21

Editor: Maria Polychronidou

Transaction Report:

Thank you again for submitting your work to Molecular Systems Biology. First of all, I would like to apologise for the delay in sending you a decision on your work. Unfortunately, after a series of reminders and promises that they would send us their comments on your work, reviewer #1 has just informed us that they will not be able to send a report after all. As such, in the interest of time, we have decided to proceed with making a decision based on the two available reports. Overall, both reviewers think that the presented findings seem interesting. They raise however a series of concerns, which we would ask you to address in a major revision.

Without repeating all the points listed below, some of the more fundamental issues are the following:

- Reviewer #2 points out that the relevance of the presented signature in the context of glioma needs to be more convincingly demonstrated.
- Reviewer #3 thinks that the ability of ccAF to capture G0 and the performance of the ccAF classifier need to be better supported.
- Reviewer #2 mentions that providing some further support, ideally experimental, for the reported stem-cell-like phenotype would significantly enhance the impact of the study.

Both reviewers provide constructive suggestions on how to address the points above and improve the study. Please let me know in case you would like to discuss any of the issues raised.

On a more editorial level, we would ask you to address the following.

REFEREE REPORTS

Reviewer #2:

Feldman et al. derive a gene signature by clustering scRNA-seq data from human neural stem cell cultures, which they term "neural G0". The neural-G0 signature correlates with DNA-replication factor expression, and hence G1 phase, in hNSC cultures. The neural-G0 signature is expressed by

>80% of astrocytes or OPCs in the developing brain, and in >50% of radial glia. Notably, the neural-G0 signature is expressed in >50% of GBM cells in a cohort of 22 GBMs. Neuro-G0 expression negatively correlates with glioma grade. They use CRISPR screening to identify hippo/yap and p53

pathway genes as positively regulating cell-cycle progression in hNSCs.

The idea of identifying the signature of a quiescent glioma stem cell in scRNA-seq data is compelling. Quiescent glioma stem cells are a putative mechanism of treatment resistance, and the authors' study has the potential to provide clinically relevant insights. The authors' methodology is thoughtfully designed. There are several moderate concerns with the approach that diminish enthusiasm for the manuscript in its current form. I would ask the authors to consider the following points:

1. It is not clear that the signature the authors have identified is relevant for glioma. Their signature is expressed in >50% of GBM cells (Figure 4a,d), and yet they claim neuro-G0 represents a glioma stem cell. GBM stem cells are very rare (~1% of cells) and are certainly not half of the tumor. It is more likely that the neuro-G0 signature is picking up differentiated astrocytic and oligodendrocytic cell types. Indeed, the authors show the neuro-G0 signature is enriched in astrocytes and oligodendrocyte-lineage genes (Figure 2e). This would explain why the neuro-G0 signature is enriched in classical and proneural samples, but depleted in mesenchymal samples (which are actually the most enriched for radial-glia-like cell types).

I would ask the authors to consider some approach to "subtracting" the component of the neuro-G0 signature which correlates with terminally differentiated glia. The original neuro-G0 signature may be relevant for NSC quiescence, but it is not identifying glioma stem cells in practice. I would challenge the authors to derive a signature that is more relevant to quiescent GBM stem cells. The first part of the manuscript could remain unchanged, but the glioma component would first "refine" the neuro-G0 signature so that the result is more relevant to glioma.

2. Along these lines, have the authors considered applying their quiescent-signature inference approach directly to GBM patient-derived lines, as they did for hNSCs? Is single-cell necessary, or could they just RNA-seq CDT1+ cells?

3. Figure 4 is problematic. In the text, it says that 60 GBMs were used from sources in Table 1. However, the figure's legend says the data is 22 tumors from Wang 2019. However, Wang 2019 is not in Table 1 and is not cited. Moreover, it is not clear if neoplastic cells have been separated from non-neoplastic glia in these images.

4. If the authors are able to derive a more GBM-relevant quiescent stem-cell signature, then I would ask them to compare it to the data of Neftel Cell 2019 and Wang Cancer Discovery 2019. In particular, how does the neuro-G0 signature distribute across the cell types of Neftel 2019? Likewise, Wang 2019 postulate a mesenchymal to proneural-cell hierarchy using in silico lineage tracing. One weakness of Wang 2019 is that only putative cycling cells were used in the analysis, which would miss or greatly under-represent the clinically relevant phenotype the authors are interested in. What is the lineage relationship between the neuro-G0 cells and the cell types of Neftel and the hierarchy of Wang? Which of Neftel or Wang is correct is an open, clinically relevant problem that the authors' approach can shed light on. Moreover, the authors clearly have the expertise to perform the large-scale meta-analysis of single-cell GBM data that is required to assess these differences, and is lacking in the literature. I would recommend that the authors place their revised glioma analysis in the context of addressing this question, and recent reviews on the subject: Fine Cancer Discovery 2019, Platten Neuro-Oncology 2020.

5. As the authors know, the only true demonstration of a glioma stem-cell is tumor propagation in vivo. It may be somewhat beyond the scope of this study, but it would greatly enhance the impact of the study if a novel marker of quiescent glioma stem cells could be identified and used to isolate cells from patient specimens and some of the more standard assays performed to demonstrate a stem-like phenotype.

6. The novelty of the results of the CRISPR screen are unclear. It seems expected that attenuation of cell-cycle checkpoints would enhance cell-cycle progression. The novelty is not clear.

7. There are many small typos. The manuscript would benefit from proofreading.

Reviewer #3:

Summary

In this manuscript, Feldman and colleagues profile neuroepithelia-derived cells and gliomas to characterize a dynamic quiescent-like state which they call Neural G0. Using single-cell RNA-seq of NSCs, the authors derive a cell cycle classifier (ccAF) which is able to distinguish cells in Neural G0 as well as transitional states, thus reaching a level of granularity not currently achieved by the gold-standard Seurat classifier. The authors apply this classifier to glioblastoma datasets, concluding that Neural G0 is enriched in lower stage tumours and correlates with patient survival. The authors furthermore knocked out genes identified in a proliferation screen, which resulted in decreased cell cycle lengths and time spent in the G1/G0 phase.

General remarks

The cell cycle classifier ccAF represents the major contribution of this work which the remainder of the text builds upon. I am, however, not convinced that ccAF accurately captures the G0 state of the cell cycle. My main concern is that the predictions made by ccAF have not been validated to an extent sufficient to affirm their efficacy in eg. the reported glioma datasets. Validation of ccAF in terms of overlap with CTD1+ cells in a FUCCI model is insufficient as there is no ability to distinguish G1 and G0 cells in this system. Furthermore, post-hoc validation of this nature does not sufficiently constitute a ground truth dataset which can be used to train a supervised classifier.

To remedy this, I recommend the authors create a ground truth dataset with accurate labels for the various cell cycle stages. Distinguishing G0 from G1 can be accomplished eg. by employing a p27 reporter (Velthoven C.T.J. & Rando T.A. Cell Stem Cell 2019) in combination with the existing FUCCI system. Having such a dataset would allow the authors to assess the accuracy of their classifier and benchmark it against current state of the art cell-cycle classifiers in a systematic manner. Though the remainder of the author's conclusions appear intuitively correct, deficiencies in reporting on the accuracy of the classifier lead me to question their results in context.

Major points

- As the classifier was only trained on data from cultured U5-hNSCs, its output is potentially very specific and not proved to be usable for other cell types. This is demonstrated in Figure 2E where neuronal cell types are predicted to be in G1 while they are expected to be in G0.
- The classifier's performance is further put into question by the gene-set overlap depicted in Figure 2C: both qNSC2 and aNSC1 should also show G1. This discrepancy is indicative of the classifier having actually learned neural stem cell quiescence as a proxy for cell cycle, as these quantities are likely correlated. This is further demonstrated in Figure 4D&E (cf. Figure 3B, the in-vitro system) where the number of cells in S-phase appears to be quite different between ccSeurat and ccAF, possibly hinting an inability to classify on in-vivo data.
- It is not clear why the authors chose to use a random forest model for classification. They should try other state of the art classifiers like a simple neural network, or support vector machine and compare their accuracy to predict the G0 state with a "ground-truth" data set.
- The CRISPR screen and KOs are not analysed using the classifier and thus, do not provide further evidence for the accuracy of the classifier.
- Methods describing the analysis presented in Figures 4 & 5 are missing.

Minor points

- No citation is offered for the expression patterns depicted in Figure 3C. To improve readability I recommend the authors format Figures 3D & E such that their axis are as those in Figure 3C.
- The authors might consider moving Table 1 to the supplement and replacing it with a summary plot.
- The authors could add cell count information to Figure 4A-C and include tumour subtype annotations in a separate UMAP. It might furthermore be insightful to group figures 4D-F by tumour stage.
- Using a broad GO category like "Mitotic Cell Cycle" for eigengene analysis is likely to return many unspecific or tangentially-related genes. The authors should consider using a curated list of cell cycle genes, such as those recommended by Seurat.
- Missing information on replicates (technical and biological), as well as batches. Several figures are missing error bars.
- I recommend the authors undertake additional rounds of proofreading, as the manuscript is wrought with spelling and grammatical errors. Furthermore, several within-text references are broken (Figure 4 legend lists subfigures from ABCDCD, while the text at several points references Figure 3 instead of Figure 5).
- If further review is considered, I would appreciate if the authors could make the source code available to the reviewers as part of the review process.

RE: MSB-20-9522**Neural G0: a quiescent-like state found in neuroepithelial-derived cells and glioma**

Samantha A. O'Connor^{1*}, Heather M. Feldman^{2*}, Sonali Arora², Pia Hoellerbauer^{2,3}, Chad M. Toledo^{2,3}, Philip Corrin², Lucas Carter², Megan Kufeld², Hamid Bolouri², Ryan Basom⁴, Jeffrey Delrow⁴, José L. McFaline-Figueroa⁵, Cole Trapnell⁵, Steven M. Pollard⁶, Anoop Patel^{2,7}, Patrick J. Paddison^{2,3*} and Christopher L. Plaisier^{1*}

Points raised by the editor:

Without repeating all the points listed below, some of the more fundamental issues are the following:

- Reviewer #2 points out that the relevance of the presented signature in the context of glioma needs to be more convincingly demonstrated.

We have addressed the relevance of the Neural G0 signature and the ccAF classifier in the context of glioma with five major points. **First**, we demonstrated in Figures 3A-D that when *de novo* clustering is applied the cells from a single GBM tumor, the resulting clusters match well to the ccAF classifications. This demonstrates that the cell cycle is a major source of variation in cells from GBM tumors, and that the ccAF classifier is capable of capturing this variation. **Second**, application of the ccAF classifier to scRNA-seq profiling of 40 GBM patient tumors showed that the Neural G0 cell cycle phase is the most prevalent phase (49.4-67.7% of neoplastic cells). This enhances our interest in Neural G0 cells as they make up a significant fraction of a tumor. **Third**, we demonstrated that the Neural G0 subpopulation was enriched with putative GBM stem-like cells based on the criteria established for scRNA-seq from Bhaduri et al., 2020 in Figure 4. This provides a novel biological function for a subset of the Neural G0 subpopulation, and suggests that the Neural G0 may be further subdivided in future studies to identify a quiescent stem-like cell subpopulation. This is especially important because a quiescent stem-like subpopulation has been shown to be responsible for tumor regrowth after Temozolomide treatment (Chen *et al*, 2012). **Fourth**, we used bulk glioma patient tumor transcriptome profiles and clinical data to demonstrate that increased expression of Neural G0 marker genes is associated improved patient prognosis. The association was independent of the common covariates tumor grade and *IDH1/2* mutation. This demonstrates that the Neural G0 signatures is clinically relevant for gliomas. **Fifth**, plotting Neural G0 signature versus a cell cycle signature showed a robust cell cycle signal along the vertical y-axis, Neural G0 signal along the lateral x-axis, and a distinct lack of cells along the diagonal (Figure 5D). The shape of this plot demonstrates that Neural G0 is independent of the cell cycle. This demonstrates that Neural G0 state is defined not just by a lack of cell cycle markers, but explicit expression of Neural G0 markers.

Taken together these five points provide strong evidence that the Neural G0 signature derived from hNSCs is relevant for GBM and glioma. Comparison of Neural G0 to the developmental

subtypes of Neftel et al., 2019 further demonstrates that the two approaches are not redundant. Neftel et al., 2019 focus on developmental subtypes whereas the ccAF focuses on the cell cycle phases, and both shed light on difference aspects of tumor biology.

- Reviewer #3 thinks that the ability of ccAF to capture G0 and the performance of the ccAF classifier need to be better supported.

We have completed new analyses that directly address the issue of overall ccAF classifier performance and specifically classification of Neural G0. **First**, we compared four top performing state-of-the-art classification methods (Abdelaal et al, 2019) by conducting 100-fold cross-validation with the training study to determine which method was superior (Figure 2A). We found that the best method was the neural network-based ACTINN method (error rate 18.4%; F1 rate ≥ 0.8 for 4 of the 7 cell cycle states), which we then developed into the revised ccAF classifier. **Second**, we validated the S and M phase ccAF predictions using a gold-standard study from Whitfield et al., 2002 (Appendix Figure S3), and G0 predictions using two studies of *in vivo* sorted quiescent neural stem cells (qNSCs) from Llorens-Bobadilla et al., 2015 and Dulken et al., 2017 (Figure 2C-D). The ccAF classifier was able to accurately classify the S and M phases, and accurately classified qNSCs as Neural G0. **Third**, we compared the ccAF classifier against the state-of-the-art ccSeurat scRNA-seq cell cycle classifier in U5-hNSCs (Figure 1A-C), HEK293T (Appendix Figure 6A-C), and a GBM tumor (Figure 3B-C). These comparisons to ccSeurat helped us to determine that ccAF cell cycle predictions were valid in non-neuroepithelial cells (HEK293T) and in patient tumors.

Thus we completely revamped the ccAF classifier construction, and evaluated its performance using gold-standard datasets and comparison to a state-of-the art scRNA-seq cell cycle classifier. These improvements greatly strengthen the confidence in the performance of the ccAF classifier. We have improved the ccAF classifier usability by making it into an easy to install and run Python package (<https://pypi.org/project/ccAF/>) that takes as input the AnnData objects used by the Python scRNA-seq analysis package scanpy. We have also made it available as an easy to run Docker image that includes all dependencies (<https://hub.docker.com/repository/docker/cplaisier/ccaf>).

- Reviewer #2 mentions that providing some further support, ideally experimental, for the reported stem-cell-like phenotype would significantly enhance the impact of the study.

We feel as well that further experimental studies are warranted, but they go beyond the scope of this manuscript in terms of time to complete and the goal of the research. We have addressed this issue computationally by adopting the scRNA-seq criteria for putative GBM stem-like cells from Bhaduri et al., 2020, and assessing the prevalence of these putative stem-like cells in four independent scRNA-seq studies of primary GBM tumors. The criteria encoded a logic for discovering putative stem cells from scRNA-seq profiles: any cell expressing *FUT4* (*SSEA1*) or *L1CAM* or *PROM1* (CD133) in conjunction with *SOX2* and not expressing *TLR4* (Bhaduri et al, 2020). This approach takes into consideration the heterogeneity observed in GBM stem cells (Lathia et al., 2015; Bradshaw et al., 2016) and provides a useful proxy for putative stem-like cells in GBM tumors. The application of this logic discovered 4,563 putative stem cells in 47,405 neoplastic cells from the four GBM scRNA-seq studies (Figure 4A) (Darmanis et al, 2017; Neftel

et al, 2019; Bhaduri et al, 2020; Wang et al, 2020). Thus, putative GBM stem-like cells account for on average 9.6% of neoplastic cells in GBM tumors. We then tested whether there was any relation between these putative stem-like cells and the ccAF defined cell cycle phases. Putative stem-like cells were significantly enriched in Neural G0 classified cells (70% of putative stem cells are in Neural G0; hypergeometric enrichment p-value = 3.4×10^{-60} ; Figure 4B; Dataset EV6). These results suggest that the Neural G0 subpopulation is where stem-like cells primarily reside, and suggests that the Neural G0 may be further subdivided in future studies to identify a quiescent stem-like cell subpopulation. We believe that these results illuminate the relationship between Neural G0 and stem-like cells, and suggest specific future studies that can further support and expound upon this important new relationship.

Reviewer #2:

Feldman et al. derive a gene signature by clustering scRNA-seq data from human neural stem cell cultures, which they term "neural G0". The neural-G0 signature correlates with DNA-replication factor expression, and hence G1 phase, in hNSC cultures. The neural-G0 signature is expressed by >80% of astrocytes or OPCs in the developing brain, and in >50% of radial glia. Notably, the neural-G0 signature is expressed in >50% of GBM cells in a cohort of 22 GBMs. Neuro-G0 expression negatively correlates with glioma grade. They use CRISPR screening to identify hippo/yap and p53 pathway genes as positively regulating cell-cycle progression in hNSCs.

The idea of identifying the signature of a quiescent glioma stem cell in scRNA-seq data is compelling. Quiescent glioma stem cells are a putative mechanism of treatment resistance, and the authors' study has the potential to provide clinically relevant insights. The authors' methodology is thoughtfully designed. There are several moderate concerns with the approach that diminish enthusiasm for the manuscript in its current form. I would ask the authors to consider the following points:

1a. *It is not clear that the signature the authors have identified is relevant for glioma. Their signature is expressed in >50% of GBM cells (Figure 4a,d), and yet they claim neuro-G0 represents a glioma stem cell. GBM stem cells are very rare (~1% of cells) and are certainly not half of the tumor. The original neuro-G0 signature may be relevant for NSC quiescence, but it is not identifying glioma stem cells in practice. I would challenge the authors to derive a signature that is more relevant to quiescent GBM stem cells. The first part of the manuscript could remain unchanged, but the glioma component would first "refine" the neuro-G0 signatures so that the result is more relevant to glioma.*

The reviewer's point that not all Neural G0 cells are stem cells was not clearly stated in the manuscript. We agree with the reviewer as the Neural G0 subpopulation is too large for all of them to be considered stem cells. We have addressed this issue computationally by adopting the scRNA-seq criteria for putative GBM stem-like cells from Bhaduri et al., 2020 which states that putative stem-like cells can be identified in scRNA-seq profiles by applying the logical filter: stem cell = ($FUT4 > 0$ or $L1CAM > 0$ or $PROM1 > 0$) and $SOX2 > 0$ and $TRL4 = 0$. This approach takes into consideration the heterogeneity observed in GBM stem cells (Lathia et al., 2015; Bradshaw et al., 2016) and provides a useful proxy for putative stem-like cells in primary GBM tumors. We applied this criteria to identify putative stem-like cell subpopulations in all four GBM tumor studies and found that the average frequency of putative GBM stem-like cells in tumors was 9.6%. We provide a new Figure 4 that directly addresses the prevalence of putative GBM stem-like cells in each of the ccAF cell cycle phases, and have plotted the expression of the GBM stem-like cell marker genes: *FUT4*, *L1CAM*, *PROM1*, and *SOX2*.

In addition, we tested whether any of the ccAF cell cycle phases were enriched with putative GBM stem-like cells. The Neural G0 subpopulation was highly significantly enriched (p-value = 3.4×10^{-60}) with the putative GBM stem-like cells and contained 70% of the total putative stem-like cells. These results suggest that the Neural G0 subpopulation is where stem-like cells primarily reside, and suggests that the Neural G0 may be further subdivided in future studies to identify a quiescent stem-like cell subpopulation. We believe that these results illuminate the

relationship between Neural G0 and stem-like cells, and suggest specific future studies that can further support and expound upon this important new relationship.

We have described and clarified the role of putative GBM stem-like cells by adding: a new section to the main text (Pages 16-17), a new Figure 4, and new methods (Pages 38). We hope this new analysis addresses the reviewers critique on the relevance of Neural G0 to GBM stem cells.

1b. It is more likely that the neural-G0 signature is picking up differentiated astrocytic and oligodendrocytic cell types. Indeed, the authors show the neuro-G0 signature is enriched in astrocytes and oligodendrocyte-lineage genes (Figure 2e). This would explain why the neuro-G0 signature is enriched in classical and proneural samples, but depleted in mesenchymal samples (which are actually the most enriched for radial-glia-like cell types). I would ask the authors to consider some approach to "subtracting" the component of the neuro-G0 signature which correlates with terminally differentiated glia.

As a preprocessing step before classification with ccAF, each GBM patient tumor scRNA-seq dataset was clustered *de novo* and then filtered to exclude clusters of oligodendrocytes (*MBP* and *PLP1*) (Valério-Gomes *et al*, 2018), astrocytes (*ETNPPL*) (Zhang *et al*, 2016c), neurons (*RBFOX3*) (Herculano-Houzel & Lent, 2005), and immune cells (*AIF1*, *CD14*, *CX3CR1*, *PTPRC*), which were distinct from tumor cell clusters. We validated these markers in the Darmanis *et al.*, 2017 scRNA-seq study where they used flow-sorting with cell-surface markers to sort out immune, oligodendrocyte, and astrocyte subpopulations from GBM patient tumors (<http://www.gbmseq.org/>). In Appendix Figure S7, we demonstrate the process we undertook to remove the terminally differentiated clusters and show that there were two immune cell clusters (9 and 16) and one putative terminally differentiated oligodendrocyte cluster (15) that were excluded for Wang *et al.*, 2020. The goal of this approach was to reduce the possibility of including terminally differentiated cells in the GBM analyses. Thus, we excluded the terminally differentiated cells from the GBM scRNA-seq studies prior to applying the ccAF classifier. Which has the same effect as what the reviewer has suggested in “subtracting the terminally differentiated astrocytic and oligodendrocytic cell types”. Therefore, it is unlikely that terminally differentiated cells are the reason behind the prevalence of Neural G0 cells we observed in the GBM tumors.

In addition, we refined the Neural G0 marker genes prior to conducting the association and survival analyses with bulk TCGA GBM patient tumor data. We identified GBM neoplastic cell specific Neural G0 marker genes by applying ccAF to classifier to the four GBM scRNA-seq studies (Table 1) (Darmanis *et al*, 2017; Bhaduri *et al*, 2020; Neftel *et al*, 2019; Wang *et al*, 2020), computed Neural G0 marker genes for each study, and discovered 22 Neural G0 marker genes in common across all four studies (Figure 5A & 5B; Dataset EV5). Eight of the 22 common GBM neoplastic cell specific Neural G0 marker genes were originally identified as Neural G0 marker genes for hNSCs (*GPM6A*, *HOPX*, *MARCKSL1*, *PLP1*, *S100B*, *SCD5*, *SCRG1*, and *TTYH1*; Figure 5B; Dataset EV1). The remaining 14 genes were unique to GBM neoplastic cells (*AQP4*, *BCAN*, *BCHE*, *GATM*, *GFAP*, *ITM2C*, *NDRG2*, *PLEKHB1*, *PMP2*, *RAMP1*, *RTN3*, *SLC22A17*, *TSC22D4*, and *TSPAN7*; Figure 5B; Dataset EV5). Significantly, 13 of the 22 genes were previously known to be associated with GBM in the DisGeNET database (*AQP4*, *BCAN*, *BCHE*, *GFAP*, *HOPX*, *MARCKSL1*, *NDRG2*, *PLEKHB1*, *PLP1*, *S100B*,

SLC22A17, *TSPAN7*, and *TTYH1*; hypergeometric over-enrichment p-value = 1.2×10^{-6} ; Figure 5B) (Piñero *et al*, 2015). Of these, *AQP4* has previously been shown to be differentially expressed in quiescent astrocytes (Yoneda *et al*, 2001); *HOPX* is a marker of quiescent radial glial neural progenitors (Berg *et al*, 2019); *NDRG2* is up-regulated in G0/G1 arrested glioma cells (Li *et al*, 2012, 2); *S100B* is a chemoattractant for tumor-associated macrophages (Wang *et al*, 2013); and *TTYH1* is required to maintain NSC stemness via its role in activating the Notch signaling pathway (Kim *et al*, 2018, 1). The genes *AQP4*, *BCAN*, *GFAP*, *PLP1*, and *S100B* are part of the astrocytic, oligodendrocytic, or proneural glioma signatures.

In summary, we addressed this critique in two ways: 1) better description of the removal of putatively terminally differentiated oligodendrocytes, astrocytes, neurons, and immune cells; and 2) better description of method used to refine the Neural G0 marker genes into GBM neoplastic cell specific Neural G0 marker genes. The removal of putatively differentiated cells was addressed in the results (Page 14), Appendix Figure 7, and methods (Page 36). The refinement of marker genes was addressed in the results (Pages 17-18), improved Figure 5 with an analysis walkthrough in Figure 5A, and methods (Page 37-38).

2. Along these lines, have the authors considered applying their quiescent-signature inference approach directly to GBM patient-derived lines, as they did for hNSCs? Is single-cell necessary, or could they just RNA-seq CDT1+ cells?

Yes, we have considered this point. The major is that *in vitro* GSCs isolates do not have as prominent a G0 subpopulation as glioma cells in patient tumors or NSCs. We experimentally define this in Figure 6D, where GSC-131 cells have a faster and more uniform G0/G1 than cultured NSCs. However, we are pursuing this in a follow up manuscript, where we performed a functional genomic screen for genes that when inhibited, "trap" GSCs in G0 using a p27 reporter. We succeeded in identifying genes that "trap" GSCs in G0 and are currently assembling a follow up manuscript that focuses on this very point. The results agree very nicely with the ccAF/Neural G0 results from the current manuscript; however, these experiments are well beyond the scope of the current paper.

In answer to the second part of this question whether single-cell characterization is required, the answer is that yes, it is necessary to use scRNA-seq to identify the different states. As demonstrated by Figure 2B, the CDT+ cells enrich for both Neural G0 and G1 subpopulations. Without single-cell characterization and using CDT+ cells we would have lumped together Neural G0 and G1, which would have confounded the discovery of Neural G0. In addition, the use of a single-cell approach allowed us to define other cell cycle states that are masked by standard FUCCI linked to flow-cytometry based approaches. However, once the cell subpopulation is defined we can use marker genes to sort out a specific cell cycle subpopulation using flow-cytometry.

3. Figure 4 is problematic. In the text, it says that 60 GBMs were used from sources in Table 1. However, the figure's legend says the data is 22 tumors from Wang 2019. However, Wang 2019 is not in Table 1 and is not cited. Moreover, it is not clear if neoplastic cells have been separated from non-neoplastic glia in these images.

We apologize for the lack of methods and incomplete legend that made the source data and study sample information unclear for Figure 4. The Wang *et al.*, 2017 reference was to the

method used to classify the GBM subtypes for Figure 4C, and did not reference the source scRNA-seq study. The source of the scRNA-seq study was in fact Neftel et al., 2019 10X genomics profiled samples and we have made corrections to the legend and the main text to reflect this. Please note that Figure 4 has now become Figure 3. The GBM study in Figure 3 is from the 22 GBM tumors from Neftel et al., 2019 that passed our preprocessing pipeline (including removal of non-neoplastic cell clusters as described for critique 1b, and removal of samples with missing clinical information). We chose this set of GBM tumors because there were many tumors and four tumors were deeply characterized with >1000 single cells, and it gave us the opportunity make comparisons against the Neftel et al., 2019 developmental subtypes. We have completely reanalyzed the Neftel et al., 2019 scRNA-seq study based on the updated cCAF classifier and have updated the figure and legend to specify the correct source and number of tumors for the scRNA-seq data.

This critique was addressed by adding information to the results (Pages 14-16), Figure 3, and the Figure 3 legend.

4. If the authors are able to derive a more GBM-relevant quiescent stem-cell signature, then I would ask them to compare it to the data of Neftel Cell 2019 and Wang Cancer Discovery 2019. In particular, how does the neuro-G0 signature distribute across the cell types of Neftel 2019? Likewise, Wang 2019 postulate a mesenchymal to proneural-cell hierarchy using in silico lineage tracing. One weakness of Wang 2019 is that only putative cycling cells were used in the analysis, which would miss or greatly under-represent the clinically relevant phenotype the authors are interested in. What is the lineage relationship between the neuro-G0 cells and the cell types of Neftel and the hierarchy of Wang? Which of Neftel or Wang is correct is an open, clinically relevant problem that the authors' approach can shed light on. Moreover, the authors clearly have the expertise to perform the large-scale meta-analysis of single-cell GBM data that is required to assess these differences, and is lacking in the literature. I would recommend that the authors place their revised glioma analysis in the context of addressing this question, and recent reviews on the subject: Fine Cancer Discovery 2019, Platten Neuro-Oncology 2020.

Using the embedding from Neftel et al., 2019 we were able to observe how the Neural G0 and other cell cycle states from the cCAF classifier are distributed across the Neftel et al., 2019 cell types (AC, MES, NPC, and OPC). The percentage of cells in the Neural G0 cell cycle state was highest in OPC and AC (91 and 87%, respectively) and lowest in MES (40%). In MES the difference in Neural G0 cells was primarily made up with cells in the G1 cell cycle state (47%). Comparing cell cycle classifications from cCAF to the Neftel et al., 2019 alternative GBM developmental classification scheme (e.g., Astrocytic (AC), Neural Progenitor Cell (NPC), Oligodendrocyte Progenitor Cell (OPC) and Mesenchymal (MES)) shows that the two methods of characterizing had some similarities and some differences (Figure 3K-L). Most cells classified as AC, NPC, or OPC were also classified as Neural G0, while MES populations had fewer Neural G0 and more G1 cells. The MES and NPC cells had a higher S, S/G2, and G2M fraction than AC and OPC cells (Figure 3L). Thus, the abundance of Neural G0 cells in AC and OPC cells is consistent with Neural G0 representing a quiescent state and/or a pre-mesenchymal state associated with the proneural to mesenchymal transition (Halliday *et al*, 2014; Bhat *et al*, 2013; Segerman *et al*, 2016).

We also projected the stem cell marker genes onto the Neftel et al., 2019 embedding and discovered that *L1CAM* is expressed at 20-23% of OPC and NPC cells, and is expressed in only 2-4% of AC and MES cells. *PROM1* also shows a bias towards NPC cells (48%) and away

from AC cells (19%), with OPC and MES in the middle (30% and 25%). The proportion of *FUT4* and *SOX2* expressing cells were fairly even across the developmental subtypes of Neftel et al., 2019. Interestingly, the putative GBM stem-cells were nearly double in prevalence in the NPC and OPC cells (7.6% and 9.4%) relative to the AC and MES (4.1% and 4.8%). Thus, the progenitor cell subpopulations have more putative GBM stem-like cells due to increased *L1CAM* and *PROM1* expression.

Unfortunately, we were unable to get access to the Wang et al., 2019 dataset and were therefore unable to analyze the mesenchymal to proneural-cell hierarchy using *in silico* lineage tracing. The raw data was available but without the necessary phenotypic information to be able to make any inferences about the hierarchies generated through *in silico* lineage tracing.

We have addressed this critique by adding to the results (Pages 15-17), Figures 3K-L and 4D, and methods (Pages 35-38).

5. As the authors know, the only true demonstration of a glioma stem-cell is tumor propagation in vivo. It may be somewhat beyond the scope of this study, but it would greatly enhance the impact of the study if a novel marker of quiescent glioma stem cells could be identified and used to isolate cells from patient specimens and some of the more standard assays performed to demonstrate a stem-like phenotype.

We agree and are in the process of developing the required constructs and cell lines in order to conduct these studies. Given the current status with the COVID-19 pandemic this process has been slower than anticipated. And, as the reviewer suggests, this is beyond the scope of this current work. We hope that future work in this vein may address better establish the functional potential of the GBM stem-like cells discovered in the Neural G0 subpopulation.

6. The novelty of the results of the CRISPR screen are unclear. It seems expected that attenuation of cell-cycle checkpoints would enhance cell-cycle progression. The novelty is not clear.

We pursued CRISPR-Cas9 screens with the notion that we could find genes which when knocked out would attenuate G0-like states in cultured U5-hNSCs. We felt this would serve two purposes. **First**, it would allow us to demonstrate "phenotypically" that genetic manipulation of certain genes would attenuate "Neural G0". Because we know from phenotypic data presented in Figure 6 that cells going into G0 in U5-hNSCs do so transiently, identifying modifiers would show that it is a dynamic state and not simply a "dead end" in cultured cells. This was indeed the case as shown in Figure 6D using time lapse microscopy of cell cycle reporter-containing cells, as well the supporting supplemental data, including loss of molecular markers associated G0 and scRNA-seq data for sgTAOK1.

Second, we hypothesized that identifying specific modifiers of G0 might help understand how Neural G0 is regulated, e.g., in glioma, and also whether attenuation of G0 would lead to down-regulation of Neural G0 in G0/G1 populations. We found three primary modifiers of G0 in U5-hNSCs that fall into different transcriptional "epistasis" groups: *p53*, *CREBBP*, and negative regulators of Hippo-Yap signaling. In the revision, we have improved the highlighting of recent findings implicating the Hippo-Yap signaling during the proneural to mesenchymal transition in

glioma, where Hippo-Yap signaling becomes higher in mesenchymal cells. This is consistent with our observation that Hippo-Yap signaling is driving down Neural G0 gene expression in our G0-skip mutant cells, and that the Neural G0 state is reduced in mesenchymal cell subpopulations in stage IV cancers (Table1; Figure 3G & J). While *CREBBP* has previously been shown to regulate p53 function, based on transcriptional responses this does not appear to be the case in U5-hNSCs. This is because the differentially expressed genes do not include p53 target genes nor Hippo-Yap target genes. Instead, *CREBBP* KO showed up-regulation of *Cyclin D1*, down-regulation of CDK inhibitors, and up-regulation of targets of *NRF1/2* transcription factors. Future experiments will be required to address how loss of *CREBBP* and negative regulators of Hippo-Yap signaling specifically attenuate G0, through indirect action (e.g., up-regulation of *CDK1* activity in previous cell cycle) or direct effects (e.g., regulation of key G1 substrates in G0/G1 phase). For *p53*, it is less clear. We explore if its role in promoting G0 is merely due to the presence of low-level DNA damage due to *in vitro* culture in the discussion. This would fit with the "checkpoint" idea. We did attempt to look for gamma *H2AX* foci in G0/G1 cells but were unable to detect them with certainty over background, compared to DNA damaging agent treatment.

We feel the screens and results are novel because we were able to identify factors that regulate G0/G1 in "self-renewing" U5-hNSCs in culture, specifically tying changes in cell cycle rate with loss of Neural G0/developmental gene expression. We feel that this is an interesting and important biological phenotype.

We have improved descriptions and discussion of the CRISPR-Cas9 screen by adding to the results (Pages 19-26), and discussion (Pages 26-30).

7. There are many small typos. The manuscript would benefit from proofreading.

We thank the reviewer for the comment and have made sure to proofread the manuscript multiple times before resubmitting.

Reviewer #3:

Summary

In this manuscript, Feldman and colleagues profile neuroepithelia-derived cells and gliomas to characterize a dynamic quiescent-like state which they call Neural G0. Using single-cell RNA-seq of NSCs, the authors derive a cell cycle classifier (ccAF) which is able to distinguish cells in Neural G0 as well as transitional states, thus reaching a level of granularity not currently achieved by the gold-standard Seurat classifier. The authors apply this classifier to glioblastoma datasets, concluding that Neural G0 is enriched in lower stage tumours and correlates with patient survival. The authors furthermore knocked out genes identified in a proliferation screen, which resulted in decreased cell cycle lengths and time spent in the G1/G0 phase.

General remarks

The cell cycle classifier ccAF represents the major contribution of this work which the remainder of the text builds upon. I am, however, not convinced that ccAF accurately captures the G0 state of the cell cycle. My main concern is that the predictions made by ccAF have not been validated to an extent sufficient to affirm their efficacy in eg. the reported glioma datasets. Validation of ccAF in terms of overlap with CTD1+ cells in a Fucci model is insufficient as there is no ability to distinguish G1 and G0 cells in this system. Furthermore, post-hoc validation of this nature does not sufficiently constitute a ground truth dataset which can be used to train a supervised classifier.

To remedy this, I recommend the authors create a ground truth dataset with accurate labels for the various cell cycle stages. Distinguishing G0 from G1 can be accomplished eg. by employing a p27 reporter (Velthoven C.T.J. & Rando T.A. Cell Stem Cell 2019) in combination with the existing Fucci system. Having such a dataset would allow the authors to assess the accuracy of their classifier and benchmark it against current state of the art cell-cycle classifiers in a systematic manner.

Though the remainder of the author's conclusions appear intuitively correct, deficiencies in reporting on the accuracy of the classifier lead me to question their results in context.

Major points

1. As the classifier was only trained on data from cultured U5-hNSCs, its output is potentially very specific and not proved to be usable for other cell types. This is demonstrated in Figure 2E where neuronal cell types are predicted to be in G1 while they are expected to be in G0.

This is an important point raised by the reviewer, and with the previous random forest classifier we may have had issues with performance for some cell cycle phases. However, the new neural network based ccAF classifier that we have developed is more accurate and better able to be applied to non-neuronal cell types. The new neural network based classifier significantly improves classification of the Neural G0, G1, Late G1 and S phases for NSCs determined by cross-validation analyses (Figure 2A). Additionally, we tested the improved ccAF classifier's ability to classify across species (quiescent NSC in rodents) and in other cell types (neuroepithelial derived cells and human embryonic kidney cells).

We tested whether the Neural G0 subpopulation from *in vitro* U5-hNSCs is similar to the quiescent NSCs (qNSCs) from two independent *in vivo* scRNA-seq profiling studies of NSCs from adult rodent neurogenesis in the subventricular zone (Dulken *et al*, 2017; Llorens-Bobadilla

et al, 2015). In both studies a majority of the qNSC cells were classified as Neural G0 by ccAF. One hundred percent of the dormant state qNSC1 from Llorens-Bobadilla *et al.*, 2015 classified as Neural G0, and 96% of the primed-quiescent state qNSC2 classified as Neural G0. The non-mitotic activated NSCs (aNSC1) state cells primarily classified as Neural G0, G1, Late G1, and M/Early G1. Whereas the mitotic aNSC2 state cells classified as S, S/G2, and G2M. These results are validated in a second independent cohort from Dulken *et al.*, 2017 where 64% of the qNSC state were classified as Neural G0, and 88% were classified as Neural G0, G1, Late G1, or M/Early G1. These results validate that the Neural G0 subpopulation from *in vitro* U5-hNSCs is similar to the quiescent NSC subpopulation *in vivo*. Additionally, this validates that the ccAF can accurately identify quiescent NSCs as Neural G0, and that the ccAF is robust enough to be applied across species using gene homology.

We also investigated where Neural G0 might be found during mammalian development by applying the ccAF to data from the developing human telencephalon (Nowakowski *et al*, 2017). We analyzed scRNA-seq data from micro-dissected developing human cerebral cortex samples (PCW 5.85-19), which was previously used to analyze the spatial and temporal developmental trajectories for 24 cell types: astrocytes, oligodendrocyte precursor cells (OPC), microglia, radial glia (RG), intermediate progenitor cells, excitatory cortical neurons, ventral medial ganglionic eminence progenitors, inhibitory cortical interneurons, choroid plexus cells, mural cells, and endothelial cells. We classified the cell cycle phase of each single cell using the ccAF classifier and cross tabulated with the 24 cell types from Nowakowski *et al.*, 2017 (Figure 2E; Dataset EV4). We found that the Neural G0 category was significantly enriched in excitatory neurons of the pre-frontal cortex (EN-PFCs), non-dividing astrocytes, OPCs, and RGs (ventral, outer, and truncated), which had a Neural G0 population ranging from 10-94% (Figure 2E; Dataset EV4). Populations characterized as dividing (i.e., "div", "div1", or "div2") are highly enriched with S/G2 and/or G2/M classified cells, and Neural G0 and G1 are absent or greatly diminished. Further, microglia have a very small Neural G0 population and the G0/G1 pool of cells are instead classified as G1 and Late G1, which is interesting because they arise from the embryonic mesoderm rather than neuroectoderm (Ginhoux & Garel, 2018). It is likely that the terminally differentiated EN-PFC cell types were classified as Neural G0 rather than G1 due to their expression of the Neural G0 markers *BEX1*, *BEX4*, *GPM6A*, *NOVA1*, *SCD5*, and *TGLN3*. However, EN-PFCs were negative or low for key Neural G0 stem/progenitor markers, e.g., *CLU*, *SOX2*, *SOX9*, and *S100B* (Appendix Figure S4-5). These results suggest that the ccAF classifier identifies quiescent populations of adult and fetal neural stem cells and astrocyte subpopulations as Neural G0.

We next tested whether the ccAF could be applied to non-neuroepithelial cell lines by applying it to 3,468 actively dividing human embryonic kidney (HEK293T) cells. The ccAF primarily classifies HEK293T cells as S/G2 (39%), G2/M (19%), and M/Early G1 (39%), with a negligible number of quiescent Neural G0 cells in the HEK293 kidney cells (0.49%). The UMAP embedding had the characteristic cyclical pattern of the cell cycle (Appendix Figure S6). We were surprised by the lack of a G1 population by the ccAF, which ccSeurat predicts (29%). However, we realized that the reason is because the cells that would otherwise be classified as G1, retain residual G2/M gene expression (e.g., *CCNB1*, *CDK1*) (Appendix Figure S6). Thus, ccAF correctly calls them as M/Early G1, rather than G1. This difference is likely due to the transforming activity of SV40 Large T antigen, which is expressed in these cells (Manfredi & Prives, 1994). We further observed that ccSeurat misclassifies cells situated between S and G2/M as G1, while ccAF classifies these cells as S/G2, which is consistent with their placement

in the cyclic embedding (Appendix Figure S6A-D) and expression of cyclins in these cells (Appendix Figure S6D). This suggests that the ccAF can resolve the cell cycle phases in a non-neuronal cell type even where they are partially skewed by changes in cell cycle gene expression

We have updated the results (Pages 9-13), figures (Figure 2, Appendix Figure S2-6), legends, and methods (Pages 34-35) to describe the construction of the ccAF classifier by testing four different state-of-the-art methods. We include the comparison to Llorens-Bobadilla et al., 2015 and Dulken et al., 2017 to demonstrate the ability to classify across species and validate the quiescent G0/G1 subpopulation. We have also updated the classification and analyses of neuroepithelial and HEK293T cells.

2. The classifier's performance is further put into question by the gene-set overlap depicted in Figure 2C: both qNSC2 and aNSC1 should also show G1. This discrepancy is indicative of the classifier having actually learned neural stem cell quiescence as a proxy for cell cycle, as these quantities are likely correlated. This is further demonstrated in Figure 4D&E (cf. Figure 3B, the in-vitro system) where the number of cells in S-phase appears to be quite different between ccSeurat and ccAF, possibly hinting an inability to classify on in-vivo data.

The previous Figure 2C was only an analysis of Neural G0 and no comparisons were made with G1, which is why there was no G1 gene overlap. This issue made us reassess this figure and decided that applying the ccAF classifier would be a more relevant and holistic approach to determine how the hNSC cell cycle phases overlap with the qNSC1, qNSC2, aNSC1, and aNSC2 subpopulations. Therefore, we applied the ccAF classifier to two independent studies of quiescent neural stem cells (qNSCs) in the adult rodent brain (Llorens-Bobadilla et al., 2015 and Dulken et al., 2017). As a side note, we have swapped out the Artegiani et al., 2017 study for the Dulken et al., 2017 study because the Artegiani et al., 2017 study scRNA-seq profiles were not publically available in a form that was amenable to classification. Additionally, the Dulken et al., 2017 study was more directly relevant to determining the association of Neural G0 with quiescence. This analysis found that 100% of qNSC1 cells classified as Neural G0 and qNSC2 cells classified as 96% Neural G0 and 4% G1. The non-mitotic aNSC1 had Neural G0 and G1 in equal parts, and to a lesser extent Late G1, S, G2/M, and M/Early G1. The mitotic aNSC2 were primarily G2/M (76%) with S and S/G2 taking up the remainder. This demonstrates that the qNSC2 has a minimal but existing G1 subpopulation, and aNSC1 has a reasonably large G1 subpopulation. This matches what the reviewer suggests for aNSC1, and is perhaps a bit less than expected for qNSC2. Future studies are required to better resolve the G0 and G1 states, and these will require a broader input training dataset and more direct gold standard comparison studies.

The difference of S phase numbers between ccAF and ccSeurat is because ccAF breaks down the cell cycle into more segments. The three ccSeurat cell cycle phases become seven in ccAF, and this necessitates that the ccSeurat clusters subdivide among the ccAF cell phases. To address whether these subdivision are meaningful beyond the ccSeurat phases, we used cyclin expression patterns. Closer examination of cyclin expression across the U5-hNSC and ccSeurat cell cycle phases reveals that the subdivision of the G1 ccSeurat phase into Neural G0, G1, Late G1, and M/Early G1 phases by ccAF is meaningful (Figure 1J-K). The novel Late G1 phase had the highest peak expression of *CCND1* (Figure 1J), which is consistent with prior studies that showed *CCND1* protein peaks just prior to entry into S phase (Matsushime *et al*, 1994). In

addition, the Neural G0 subpopulation has the lowest peak *CCND1* gene expression (Figure 1J), a hallmark of quiescence (Sherr, 1995), whereas ccSeurat lumps together high, medium, and low *CCND1* expressing cells (Figure 1K). In addition, the U5-hNSC cell cycle phases better stratifies *CCNA2* and *CCNB1* expression into more discrete expressing subpopulations (high, medium, and low) across S, S/G2, G2/M, and M/Early G1, further demonstrating that these phases are distinct (Figure 1J). The U5-hNSC cell cycle phases highly overlap with the state-of-the-art ccSeurat, and the U5-hNSC cell cycle phases outperform ccSeurat by classifying cells into more specific cell cycle phases which better capture the real biology of the cell cycle as demonstrated through meaningful changes in cyclin expression between cell cycle phases. These meaningful changes in cyclin expression between the ccAF cell cycle phases were validated in HEK293T cells (Appendix Figure S6).

In addition, we have added to the ccAF and ccSeurat analysis a comparison from the tumor MGH143 from Neftel et al., 2019. This nicely shows that from *in vivo* data we have good concordance between ccAF and ccSeurat (Figure 3B-C). In addition, the full analysis of all tumors from Neftel et al., 2019 shows good concordance between ccAF and ccSeurat (Figure 3E-F).

To address this critique we have updated the results (Pages 11-12, 13, 15-16), figures (Figure 2-4, Appendix Figure S6), legends, and methods (Pages 35-37) to describe ccAF classification of two independent qNSC studies, and provided a more thorough comparison of the ccAF and ccSeurat across HEK293T and GBM studies.

3. It is not clear why the authors chose to use a random forest model for classification. They should try other state of the art classifiers like a simple neural network, or support vector machine and compare their accuracy to predict the G0 state with a "ground-truth" data set.

As the reviewer suggested, we tested four different state of the art classifier methods which were previously found to be useful for building classifiers in scRNA-seq (Abdelaal et al., 2019; PMID = 31500660): 1) Support Vector Machine with rejection (SVMrej), 2) Random Forest (RF), 3) scRNA-seq optimized K-Nearest Neighbors (KNN) (Wolf *et al*, 2018), and 4) scRNA-seq optimized Neural Network (NN) method ACTINN (Ma & Pellegrini, 2020). We selected the 1,536 most highly variable genes in the U5-hNSC scRNA-seq profiles as the training dataset for the classifier. We applied 100-fold cross-validation (CV) for each classifier method and determined that the NN method ACTINN was statistically similar or slightly better at predicting each cell cycle phase than the next best classifier, and had a significantly higher F1 score for Late G1 (p -value $\leq 4.3 \times 10^{-64}$, Fig. 2A). The ACTINN classifier had the best overall error rate of 18.4% in the CV studies, which was the best of all the methods tested. The ACTINN based classifier was labeled ccAF for cell cycle ASU/Fred Hutch.

Next, as the reviewer suggested, we applied the ccAF classifier to a gold standard dataset. We chose a cell-cycle synchronized time-series dataset from HeLa cells that simultaneously characterized transcriptome profiles and experimentally determined whether the cells were in S or M phase at each time point (Whitfield et al., 2002). The ccAF classifier had an error rate of 13.7% when applied to the gold standard Whitfield et al., 2002 dataset (Fig. S2). This demonstrates that the ccAF classifier accurately predicts S and M cell cycle phase for each query transcriptome profile (single cell or standard RNA-seq/microarray).

We validated the G0/G1 phase classifications by experimentally determining which cells from the U5-hNSCs belonged to the G0/G1 subpopulations using the well-established fluorescent ubiquitination-based cell cycle indicator (FUCCI) (Sakaue-Sawano *et al*, 2008), coupled with flow-cytometry to enrich for the CDT⁺ G0/G1 cell subpopulations. The enriched G0/G1 subpopulations were then quantified using scRNA-seq and the cell cycle phase of each cell was classified using ccAF. The U5-hNSC Neural G0 and G1 subpopulations were enriched in the Cdt⁺ subpopulation ($\log_2(\text{FC}) > 0$; Figure 1B), whereas the U5-NSC Late G1, S/G2, and G2/M subpopulations were all significantly depleted ($\log_2(\text{FC}) \leq -1$; Figure 1B). This experimentally validates that we have correctly defined the G0/G1 subpopulations using the well-established FUCCI system for detecting cell cycle states. Importantly, the Neural G0 population is enriched when sorting for CDT⁺ cells, which validates that this subpopulation is a part of the G0/G1 pool of cells.

Next, we tested whether the Neural G0 subpopulation from *in vitro* U5-hNSCs is similar to the quiescent NSCs (qNSCs) from two independent *in vivo* scRNA-seq profiling studies of NSCs from adult rodent neurogenesis in the subventricular zone (Dulken *et al*, 2017; Llorens-Bobadilla *et al*, 2015). In both studies a majority of the qNSC cells were classified as Neural G0 by ccAF. One hundred percent of the dormant state qNSC1 from Llorens-Bobadilla *et al.*, 2015 classified as Neural G0, and 96% of the primed-quiescent state qNSC2 classified as Neural G0. The non-mitotic activated NSCs (aNSC1) state cells primarily classified as Neural G0, G1, Late G1, and M/Early G1. Whereas the mitotic aNSC2 state cells classified as S, S/G2, and G2M. These results are validated in a second independent cohort from Dulken *et al.*, 2017 where 64% of the qNSC state were classified as Neural G0, and 88% were classified as Neural G0, G1, Late G1, or M/Early G1. These results validate that the Neural G0 subpopulation from *in vitro* U5-hNSCs is similar to the quiescent NSC subpopulation *in vivo*. Additionally, this validates that the ccAF can accurately identify quiescent NSCs as Neural G0, and that the ccAF is robust enough to be applied across species using gene homology.

We have updated the results (Pages 9-11), figures (Figure 2, Appendix Figure S2-3), legends, and methods (Pages 34-35) to describe the construction of the ccAF classifier by testing four different state-of-the-art methods. We have also included the comparison to “ground truth” datasets from Whitfield *et al.*, 2002 to validate S and M phases, our own FUCCI CDT⁺ sorted NSCs to validate G0/G1-like phases (Neural G0, G1, Late G1, and M/Early G1), and Llorens-Bobadilla *et al.*, 2015 and Dulken *et al.*, 2017 to validate the quiescent G0/G1 subpopulation. We are also in the process of generating our own hNSC p27 sorted NSC cells. However, given the current status with the COVID-19 pandemic, this process has been slower than anticipated.

- The CRISPR screen and KOs are not analysed using the classifier and thus, do not provide further evidence for the accuracy of the classifier.

In the paper we do apply the ccAF classifier to CDT⁺ sorted scRNA-seq data from a KO of *TAOK1* in U5-hNSCs (Appendix Figure S18). Because this is an important point we have moved this up from the Appendix to main Figure 7F-G. This analysis allowed us to connect the CRISPR screen back to the ccAF at the end of the paper.

Finally, to more directly confirm reprogramming of G0/G1 population in a G0-skip mutant, we applied the ccAF classifier to scRNA-seq profiles from CDT⁺ G0/G1-sorted U5-hNSCs with KO of *TAOK1* and compared that to WT-sorted scRNA-seq profiles (Figure 7F-G). The percentage of Neural G0 phase-classified cells in CDT⁺ WT is 16.3% and is significantly reduced to 4.1% in CDT⁺ *TAOK1* KO (p-value < 2.2×10^{-16} ; Figure 7G). The S phase-classified cells are also significantly decreased in CDT⁺ WT vs. CDT⁺ *TAOK1* KO cells (from 2.7% to 0.67%; p-value =

9.9×10^{-7}). On the other hand, the Late G1 classified cells are significantly increased (from 5.8% to 17.0%; p-value $< 2.2 \times 10^{-16}$), as are cells in the M/Early G1 (from 11.2% to 18.8%; p-value = 8.3×10^{-13}) and G2/M (from 1.0% to 1.6%; p-value = 0.036). The expansion of the M/Early G1 in *TAOK1* KO cells could explain the increase in mitotic genes observed in the bulk G0/G1 RNA-seq data in *TAOK1* KO cells (Figure S7F), suggesting that *TAOK1* helps attenuate expression of mitotic genes from the previous cell cycle. The highly significant drop in Neural G0 cells and redistribution to the mitotic adjacent Late G1 and M/Early G1 phases supports the hypothesis that NSC G0-skip mutants lose a significant fraction of the Neural G0 subpopulation and reprogram G1 transcription networks to promote entry into G1-S.

To address this critique we have updated the results (Pages 26), Figure 7F-G, legends, and methods (Pages 46-47) to describe comparison CDT+ sorted scRNA-seq profiles from WT and sgTAOK1 KO.

- Methods describing the analysis presented in Figures 4 & 5 are missing.

The methods describing the analysis of the Neftel et al., 2019 and all other gliomas, including the TCGA bulk transcriptome analyses have been added to the methods on Pages 35-39.

Minor points

- No citation is offered for the expression patterns depicted in Figure 3C. To improve readability I recommend the authors format Figures 3D & E such that their axis are as those in Figure 3C.

The cyclin expression patterns in Fig 3C are based upon Fig 1 from Darzynkiewicz et al., 1996 who used flow-cytometry coupled with intracellular cyclin antibody staining. We have slightly modified the CCND1 expression based on Fig 4 from Matsushime et al., 1994, which Darzynkiewicz et al., 1996 show in Fig 6 is a known expression pattern. Taken together these two sources were used to construct Fig 3C. We have moved Fig 3C to the supplement as it no longer fits in the current figures.

Additionally, the reviewer makes a good point that Fig 3C and the ridge plots are fundamentally different plots. The ridge plots are useful for the single-cell use case because we do not have a time component, which is what is on the x-axis of Fig 3C. The ridge plots dichotomize the cells by cluster and show the histograms of cyclin expression across an ordered qualitative representation of the cell cycle on the y-axis. We feel this is a fairer representation of the data than deriving a time-component from the single-cell data. We also moved Fig 3C to the supplement to alleviate any confusion caused by these different representations of similar data.

Fig 1. Cellular levels of D type cyclins, cyclin E, cyclin A, and cyclin B1 at different phases of the cell cycle. In normal cells and in some tumor cell lines during their exponential growth, the expression of D type cyclins is transient during G₁ and the cells entering S phase are cyclin D negative (e.g., see Fig. 2). Also, G₀ cells are cyclin D negative (see Fig. 3). In many tumor lines, however, the level of D-type cyclins remains high and invariable through the whole cell cycle (see Fig. 6).

To address this critique we have moved this figure to Appendix Figure S1A and added the appropriate references to the Appendix Figure S1A legend.

- The authors might consider moving Table 1 to the supplement and replacing it with a summary plot.

We appreciate the reviewer's recommendation that a summary figure would help focus readers on key points. However, the sheer number of datasets we have in Table 1 make it difficult to conceptualize the key points, which are detailed in the results section of the manuscript. Additionally, if we add a figure to replace Table 1 we would have to cut out a figure we currently have in the manuscript. We feel that Table 1 accurately captures this information, and if the editor feels it is important and is willing to allow another figure, we would be happy to develop such a summary figure.

- The authors could add cell count information to Figure 4A-C and include tumour subtype annotations in a separate UMAP. It might furthermore be insightful to group figures 4D-F by tumour stage.

We have updated the legend information for Figure 4 (now Figure 3) to clarify the issues the reviewer describes. This figure uses the 11,376 cells from 22 grade IV GBM tumors from Neftel et al., 2019. Thus, grouping by tumor stage is unnecessary as all the tumors are grade IV GBMs. The UMAP in Figure 3G & 3J show the cells labeled by the Wang et al., 2017 GBM subtypes classical, mesenchymal, and proneural. We also show the break-down of those subtypes across the ccAF cell cycle phases. We have significantly improved the Figure 3 legend, and descriptions in the text to highlight and make these analyses more obvious.

- Using a broad GO category like "Mitotic Cell Cycle" for eigengene analysis is likely to return many unspecific or tangentially-related genes. The authors should consider using a curated list of cell cycle genes, such as those recommended by Seurat.

We found a significant similarity between the eigengenes for the 124 genes from the "mitotic cell cycle" GO term and the 54 Seurat G2M phase genes (Spearman correlation $R = 0.94$; $p\text{-value} \leq 2.2 \times 10^{-16}$). Meaning the results would change very little if we swapped over to the more selective and potentially better curated Seurat G2M cell cycle gene list. But, we have done as the reviewer suggested and swapped over the underlying cell cycle gene list to the 54 genes annotated to the Seurat G2M cell cycle. The results of Figure 5D did not change significantly, the magnitude of anti-correlation (R) between the cell-cycle and Neural G0 eigengene dropped slightly from -0.58 to -0.48 , and the survival results are very similar. We have updated Figure 5, the results (Pages 17-19), and methods (Pages 38-39) to reflect this change to the 54 genes annotated to Seurat G2M cell cycle.

- Missing information on replicates (technical and biological), as well as batches. Several figures are missing error bars.

We have scanned all figures for missing error bars and added sample numbers and replicate information where applicable.

- I recommend the authors undertake additional rounds of proofreading, as the manuscript is

wrought with spelling and grammatical errors. Furthermore, several within-text references are broken (Figure 4 legend lists subfigures from ABCDCD, while the text at several points references Figure 3 instead of Figure 5).

We thank the reviewer for the comment and have made sure to proofread the manuscript multiple times before resubmitting.

- If further review is considered, I would appreciate if the authors could make the source code available to the reviewers as part of the review process.

To address this comment we have released the code into GitHub, and the data into FigShare.

- Data used to conduct analyses: figshare.com - Neural G0: a quiescent-like state found in neuroepithelial-derived cells and glioma (https://figshare.com/projects/Neural_G0_a_quiescent-like_state_found_in_neuroepithelial-derived_cells_and_glioma/86939)
- Code used to conduct analyses: github.com - U5_hNSC_Neural_G0 (https://github.com/plaisier-lab/U5_hNSC_Neural_G0)

We have also put together a website that is a walk-through of the codes use: https://plaisier-lab.github.io/U5_hNSC_Neural_G0/

Additionally, this work generated the ccAF classifier which is available in multiple forms:

- ccAF code: github.com – (<https://github.com/plaisier-lab/ccAF>)
- ccAF pypi.org package – (<https://pypi.org/project/ccAF/1.0.1/>)
- ccAF installed as a Docker image – (<https://hub.docker.com/r/cplaisier/ccaf>)

Thank you for sending us your revised manuscript. We have now heard back from the two reviewers who were asked to evaluate your study. Overall, the reviewers think that the study has improved as a result of the performed revisions. However, reviewer #2 still raises significant concerns regarding the Neural GO signature, which preclude the publication of the study in its current form.

As you may already know our editorial policy in principle allows a single round or major revision. However, given the interest in the topic and the supportive comments by reviewer #3, we consulted with reviewer #2 and asked whether they think that you could be given a chance to perform an exceptional second round of revisions to address their concerns. Reviewer #2 mentioned: "The authors should be given another chance to revise their manuscript. The signature is flawed as it stands but it could be improved in a reasonable amount of time if the authors were willing to do so. The authors seemed somewhat resistant to change their signature in the first round and the signature they have derived would not, in my opinion, be useable. My main concern is that the existing signature overlaps with core signature genes for differentiated glia and as such would not be useable to distinguish quiescent glioma cells from differentiated glia. A true signature of quiescence would be very useful and I would like to see them have another chance to revise." As such, we would like to offer you a chance to address the remaining issues raised by reviewer #1 in an exceptional (and last) round of major revisions. It is very important that all remaining issues are convincingly addressed.

On a more editorial level, we would like to ask you to address the following issues.

REFEREE REPORTS

Reviewer #2:

The authors have improved their manuscript and addressed some of my concerns. I am an advocate for data-driven and machine-learning approaches in general, and I'd like to see machine-learning more widely adopted and accepted more readily by biologists with a more traditional molecular background/perspective. I say this by way of preamble, since I have significant remaining concerns regarding the actual signature which has been derived. This is partially due to confusion in the literature which has been propagated to the author's work, combined with a lack of direct expertise in glioma biology among the key authors. Once published, researchers will have a tendency to accept the signature on face value (as is the case for the Seurat signatures) and apply it without further scrutiny. I would ask the authors to consider the following points:

1. In their response, the authors say: "The reviewer's point that not all Neural G0 cells are stem cells was not clearly stated in the manuscript. We agree with the reviewer as the Neural G0 subpopulation is too large for all of them to be considered stem cells. We have addressed this issue computationally by adopting the scRNA-seq criteria for putative GBM stem-like cells from Bhaduri et al., 2020 which states that putative stem-like cells can be identified in scRNA-seq profiles by applying the logical filter: stem cell = (FUT4>0 or L1CAM>0 or PROM1>0) and SOX2>0 and TLR4==0."

However, the Bhaduri paper is of extremely poor quality. Their approach should not be mimicked or their filters adopted. Firstly, PROM1 (CD133) is not a stem cell marker at the RNA level. Only specific protein epitopes (AC133, AC141) enrich for glioma-propagating cells. Bhaduri et al do not seem to be aware of this, as no glioma specialists are among their key authors. Secondly, while TLR4 expression is reduced in CD133+ glioma stem cells, there is no reason to expect that it is not expressed in CD133- glioma stem-cells. Bhaduri et al need to set TLR4==0 in their filter since they were unable to separate neoplastic cells from non-neoplastic cells bioinformatically in their preprocessing (note they label many of their cells as "unclassified" and that they did not assess expressed point mutations). So, they use TLR4==0 to rule out macrophages. Lastly, this simple filter is surely biased as is demonstrated by the authors' own analysis, which showed that "The proportion of FUT4 and SOX2 expressing cells were fairly even across the developmental subtypes of Neftel et al.,".

I think the authors would be better served to identify quiescent populations from within several established glioma cell types, while not directly asserting if these are stem cells or not. The GBM cell of origin is unknown and there are multiple GBM cell types that will form colonies in soft agar or which can propagate lesions in vivo. What are the quiescent subsets of the four Neftel subtypes (which the authors address), or of cells classified by the Verhaak mesenchymal and proneural cell types? I think this is a more tractable application of the authors' technology and not require much reworking of the existing study.

Researchers have been looking for a marker for glioma propagating cells, and in particular quiescent glioma-propagating cells, for over a decade. It is unlikely that the authors will identify a single gene marker for this population or even a 22 gene signature which uniquely identifies these cells from scRNA-seq data. Rather, what I think is tractable is a hierarchical decision procedure to filter scRNA-seq data and enrich for quiescent (but not terminally differentiated) cells.

2. In their response to my concerns about the signature quality, the authors say: "As a preprocessing step before classification with ccAF, each GBM patient tumor scRNA-seq dataset was clustered de novo and then filtered to exclude clusters of oligodendrocytes (MBP and PLP1)...". Yet later they say, "Eight of the 22 common GBM neoplastic cell specific Neural G0 marker genes were originally identified as Neural G0 marker genes for hNSCs (GPM6A, HOPX, MARCKSL1, PLP1, S100B, SCD5, SCRG1, and TTYH1; Figure 5B; Dataset EV1). The remaining 14 genes were unique to GBM neoplastic cells (AQP4, BCAN, BCHE, GATM, GFAP, ITM2C, NDRG2, PLEKHB1, PMP2, RAMP1, RTN3, SLC22A17, TSC22D4, and TSPAN7; Figure 5B; Dataset EV5).".

PLP1 is myelin proteolipid protein, which is expressed by terminally differentiated oligodendrocytes. Nerve-fiber myelination is a core function of oligodendrocytes. So, it makes sense that the authors would use it to identify and remove differentiated oligodendrocytes. How then does it reappear in the authors' list for the Neural G0 signature?

Similarly, GFAP and AQP4 are broadly expressed in the astrocytic lineage and are certainly expressed by non-quiescent cells. GFAP is commonly used as a control for RT-PCR for example. The authors have not addressed my concern about the Neural G0 signature having significant overlap with very generic markers of differentiated glia. Hence, any classifier that uses this signature will mostly pick up differentiated glia.

3. Another example that the Neural G0 signature is flawed, which was raised in the original review, is that it is expressed in the majority of GBM cells regardless of cell type (Fig 3K-L). This is not surprising, as the core signature contains GFAP which is very broadly expressed in glia. Looking at Figure 5G, it is fairly clear that the Neural G0 signature anticorrelates with markers of cycling cells. And, it seems like many of the results are simply derived from that, for example, the fact that Neural G0 anticorrelates with tumor grade (Figure 5C) and survival (Figure 5E) is easily interpreted as higher grade samples simply having more cycling cells and fewer differentiated glia, and these more advanced tumors being associated with shorter survival.

4. Minor: It would be an advantage to the community to include an analysis of the 19 samples from Wang Cancer Discovery 2019, as this is the second largest cohort outside of Neftel 2019. I notice that Wang et al. have uploaded a list of phenotype labels to their GEO repo. At any rate it shouldn't be hard to classify them de novo based on PCA or simple markers of mesenchymal (CD44, CHI3L1) and proneural (OLIG2, DLL3) cells.

Reviewer #3:

The authors address all my previous concerns and the revised manuscript has significantly improved as compared to the original one.

Only a minor request to further improved clarity would be to include the various marker genes used to identify the clusters and assess the cell cycle phases of the cells in the tSNE plot. This would

give the reader an idea of how good the computed cluster boundaries are.
There are still some typos.

RE: MSB-20-9522R**Neural G0: a quiescent-like state found in neuroepithelial-derived cells and glioma**

Samantha A. O'Connor¹ [✦], Heather M. Feldman² [✦], Sonali Arora², Pia Hoellerbauer^{2,3}, Chad M. Toledo^{2,3}, Philip Corrin², Lucas Carter², Megan Kufeld², Hamid Bolouri², Ryan Basom⁴, Jeffrey Delrow⁴, José L. McFaline-Figueroa⁵, Cole Trapnell⁵, Steven M. Pollard⁶, Anoop Patel^{2,7}, Patrick J. Paddison^{2,3*} and Christopher L. Plaisier^{1*}

Points raised by the editor:

Reviewer #2 mentioned: "The authors should be given another chance to revise their manuscript. The signature is flawed as it stands but it could be improved in a reasonable amount of time if the authors were willing to do so. The authors seemed somewhat resistant to change their signature in the first round and the signature they have derived would not, in my opinion, be useable. My main concern is that the existing signature overlaps with core signature genes for differentiated glia and as such would not be useable to distinguish quiescent glioma cells from differentiated glia. A true signature of quiescence would be very useful and I would like to see them have another chance to revise." As such, we would like to offer you a chance to address the remaining issues raised by reviewer #1 in an exceptional (and last) round of major revisions. It is very important that all remaining issues are convincingly addressed.

We appreciate the opportunity to submit a second revision. It was also our concern that terminally differentiated glial cells could confound our analyses, which is why we excluded them by removing clusters of cells expressing well-established marker genes of terminally differentiated cells. We included this in our original submission, and the goal was to focus all downstream analyses on neoplastic cells. For this revision, we have further validated that the cells included from the glioma scRNA-seq studies are neoplastic by inferring copy number variations for each cell. We observed distinct patterns of chromosomal deletions or amplifications from the cells for each of the tumor samples. We did not observe any normal diploid cells after filtering out terminally differentiated cells. Therefore, our application of the ccAF classifier to the scRNA-seq glioma datasets was unlikely to be confounded by terminally differentiated cells.

The reviewer also suggested that terminally differentiated cells may be confounding our analyses of whole tumor biopsies from the TCGA. However, each tumor was scrutinized by multiple pathologists and was required to have a high percentage of neoplastic cells. And tumor purity measurements conducted by Aran et al., Nat. Comm. 2015 showed that LGG and GBM had the second and seventh greatest median purity out of the 21 tumors surveyed. These results demonstrate that the biopsy tissue profiled was primarily neoplastic and suggests that

the transcriptome profiles' information captures neoplastic cell expression. With this knowledge, we felt justified deriving a Neural G0 signature using the 22 neoplastic GBM cell-specific Neural G0 marker genes identified from the GBM scRNA-seq studies.

Thus, the likelihood of terminally differentiated cells confounding these studies is improbable, and we feel like we have taken adequate measures to avoid just such a situation. We appreciate the reviewer's concern but hope that the filtering strategy and the new copy number analyses have sufficiently addressed this concern. We also hope that the classification of the quiescent subpopulation of NSCs as the Neural G0 state and observations of patient-level associations with tumor grade and patient survival demonstrates that our signature is a proxy for quiescent cells in glioblastoma tumors.

We additionally addressed the remaining editorial comments. Please let us know if they do not meet your requirements, and we would be happy to revise as needed.

Reviewer #2:

The authors have improved their manuscript and addressed some of my concerns. I am an advocate for data-driven and machine-learning approaches in general, and I'd like to see machine-learning more widely adopted and accepted more readily by biologists with a more traditional molecular background/perspective. I say this by way of preamble, since I have significant remaining concerns regarding the actual signature which has been derived. This is partially due to confusion in the literature which has been propagated to the author's work, combined with a lack of direct expertise in glioma biology among the key authors. Once published, researchers will have a tendency to accept the signature on face value (as is the case for the Seurat signatures) and apply it without further scrutiny. I would ask the authors to consider the following points:

1. In their response, the authors say: "The reviewer's point that not all Neural G0 cells are stem cells was not clearly stated in the manuscript. We agree with the reviewer as the Neural G0 subpopulation is too large for all of them to be considered stem cells. We have addressed this issue computationally by adopting the scRNA-seq criteria for putative GBM stem-like cells from Bhaduri et al., 2020 which states that putative stem-like cells can be identified in scRNA-seq profiles by applying the logical filter: stem cell = (FUT4>0 or L1CAM>0 or PROM1>0) and SOX2>0 and TRL4==0."

However, the Bhaduri paper is of extremely poor quality. Their approach should not be mimicked or their filters adopted. Firstly, PROM1 (CD133) is not a stem cell marker at the RNA level. Only specific protein epitopes (AC133, AC141) enrich for glioma-propagating cells.

Bhaduri et al do not seem to be aware of this, as no glioma specialists are among their key authors. Secondly, while TLR4 expression is reduced in CD133+ glioma stem cells, there is no reason to expect that it is not expressed in CD133- glioma stem-cells. Bhaduri et al need to set TLR4==0 in their filter since they were unable to separate neoplastic cells from non-neoplastic cells bioinformatically in their preprocessing (note they label many of their cells as "unclassified" and that they did not assess expressed point mutations). So, they use TLR4==0 to rule out macrophages. Lastly, this simple filter is surely biased as is demonstrated by the authors' own analysis, which showed that "The proportion of FUT4 and SOX2 expressing cells were fairly even across the developmental subtypes of Neftel et al.,".

I think the authors would be better served to identify quiescent populations from within several established glioma cell types, while not directly asserting if these are stem cells or not. The GBM cell of origin is unknown and there are multiple GBM cell types that will form colonies in soft agar or which can propagate lesions in vivo. What are the quiescent subsets of the four Neftel subtypes (which the authors address), or of cells classified by the Verhaak mesenchymal and proneural cell types? I think this is a more tractable application of the authors' technology and not require much reworking of the existing study.

Researchers have been looking for a marker for glioma propagating cells, and in particular quiescent glioma-propagating cells, for over a decade. It is unlikely that the authors will identify a single gene marker for this population or even a 22 gene signature which uniquely identifies these cells from scRNA-seq data. Rather, what I think is tractable is a hierarchical decision procedure to filter scRNA-seq data and enrich for quiescent (but not terminally differentiated) cells.

Reviewer comments prompted the impetus for the investigation of stem cells, and the results were conclusive and novel enough that we thought it prudent to include them in the manuscript. As for refocusing our analysis, we feel that we have reached the extent of the assertions we can make for both the quiescent and stem cell subpopulations with the current scRNA-seq datasets. We understand the reviewer's point that the criteria for identifying putative stem-like cells do have caveats. However, in these studies we use scRNA-seq data which unfortunately cannot inform us about the glycosylation status of specific gene (e.g., CD133) epitopes, especially the glycosylations bound by the antibodies AC133 and AC141. Therefore, we used the scRNA-seq data at hand and the putative stem-like cell classification method from the peer-reviewed Bhaduri et al., 2020 paper published in the reputable journal Cell Stem Cell. We found this was the only approach to query the relationship between putative stem-like and the Neural G0 cells. If we had additional information, we would have used it. Gathering such information goes beyond the scope of these studies.

We agree with the reviewer that the Bhaduri et al., 2020 classification method has some caveats. Still, it is the best method published thus far for identifying putative stem-like cells from

scRNA-seq data. The reviewer's assertion about PROM1 (CD133) and TLR4 is interesting but developing an improved putative stem-like cell classifier goes beyond this manuscript's scope. Even with the caveats, we found that the Bhaduri et al., 2020 putative stem-like cell classification method was useful in determining that the Neural G0 subpopulation is significantly enriched with putative stem-like cells (**Figure 4A**). This result was replicated across four independent scRNA-seq datasets (Dataset EV6), strongly suggesting it was not spurious. We feel confident that this result supports our assertion in the main text that, "*These results suggest that Neural G0 populations harbor stem-like cell subpopulations and that Neural G0 captures multiple subpopulations of non-dividing cells.*" Additionally, we have been cautious that each use of "stem cell" in the manuscript is prefaced with "putative stem-like" to indicate that these are likely stem cells. Future studies of these subpopulations will require collecting new data that simultaneously quantify CD133 glycosylation and single cell transcriptomes through methods like Cellular Indexing of Transcriptomes and Epitopes by Sequencing (CITE-seq from 10X), but that goes beyond the scope of our current studies.

We have standardized how we refer to the putative stem-like cells in the main text on Pages 16-17, Figure 4, and the Figure 4 legend to indicate that these are likely stem cells.

2. In their response to my concerns about the signature quality, the authors say: "As a preprocessing step before classification with ccAF, each GBM patient tumor scRNA-seq dataset was clustered de novo and then filtered to exclude clusters of oligodendrocytes (MBP and PLP1)...". Yet later they say, "Eight of the 22 common GBM neoplastic cell specific Neural G0 marker genes were originally identified as Neural G0 marker genes for hNSCs (GPM6A, HOPX, MARCKSL1, PLP1, S100B, SCD5, SCRG1, and TTYH1; Figure 5B; Dataset EV1). The remaining 14 genes were unique to GBM neoplastic cells (AQP4, BCAN, BCHE, GATM, GFAP, ITM2C, NDRG2, PLEKHB1, PMP2, RAMP1, RTN3, SLC22A17, TSC22D4, and TSPAN7; Figure 5B; Dataset EV5).".

PLP1 is myelin proteolipid protein, which is expressed by terminally differentiated oligodendrocytes. Nerve-fiber myelination is a core function of oligodendrocytes. So, it makes sense that the authors would use it to identify and remove differentiated oligodendrocytes. How then does it reappear in the authors' list for the Neural GO signature?

Similarly, GFAP and AQP4 are broadly expressed in the astrocytic lineage and are certainly expressed by non-quiescent cells. GFAP is commonly used as a control for RT-PCR for example. The authors have not addressed my concern about the Neural GO signature having significant overlap with very generic markers of differentiated glia. Hence, any classifier that uses this signature will mostly pick up differentiated glia.

The reviewer brings up an interesting point that high expression of PLP1 is a marker for oligodendrocytes (**Figure 5C**), and medium expression of PLP1 is useful in the Neural G0

signature (**Figure 5C**). While the use of PLP1 both as an oligodendrocyte marker and in the Neural G0 signature may seem contradictory, it just points to the power of employing an unbiased data-driven approach to study human disease.

Interestingly, GFAP is more heavily expressed in neoplastic Neural G0 cells relative to other cell types, with astrocytes and OPCs coming in a close second (**Figure 5D**). The AQP4 gene is equivalently expressed by astrocytes and neoplastic Neural G0 cells (**Figure 5E**). The PLP1, GFAP, or AQP4 genes make poor markers alone as they also show expression at the same level in at least one other cell type. Only by combining all 22 common GBM neoplastic cell-specific Neural G0 marker genes can we achieve discriminative power.

The final question about whether this signature would pick out differentiated glia or not is definitely of interest for our future studies. Based upon our preliminary results looking at Nowakowski et al., 2017 from the developing telencephalon, it appears that partially differentiated radial glial cells (oRG, tRG, and vRG) and OPCs are classified primarily as Neural G0. Differentiated astrocytes are also mainly classified as Neural G0. An interesting future question will be whether it is possible to delineate the difference between the different G0 subpopulations (quiescence, senescence, and differentiated cell cycle exit). Additional datasets that characterize these states would be required to determine if it is possible to differentiate and what markers differentiate between these biological states.

We have addressed this comment by adding three new panels to Figure 5 (C-E) and the corresponding figure legend updates. The main text was modified on Page 18 Lines 18-26 to describe the essential points concerning PLP1, GFAP, and AQP4 expression across neuroepithelial cell types. Additional text has been added to the manuscript to describe how each of these genes contributes to the 22 gene signature. We have added an appendix figure (Appendix Figure S10) that shows the signature's discriminative power to the Supplemental PDF.

3. Another example that the Neural GO signature is flawed, which was raised in the original review, is that it is expressed in the majority of GBM cells regardless of cell type (Fig 3K-L). This is not surprising, as the core signature contains GFAP which is very broadly expressed in glia. Looking at Figure 5G, it is fairly clear that the Neural GO signature anticorrelates with markers of cycling cells. And, it seems like many of the results are simply derived from that, for example, the fact that Neural GO anticorrelates with tumor grade (Figure 5C) and survival (Figure 5E) is easily interpreted as higher grade samples simply having more cycling cells and fewer differentiated glia, and these more advanced tumors being associated with shorter survival.

As noted by the reviewer in major point #2, we have expressly excluded differentiated cells from our glioma analyses. We accomplished this by applying *de novo* clustering to the normalized scRNA-seq data and removing clusters with high expression of well known, validated gene markers for oligodendrocytes, astrocytes, neurons, and immune cells. Thus, we applied the

ccAF classification only to neoplastic cells, and therefore neither the classification nor the downstream analyses are likely to be confounded by differentiated glial cells.

In response to major point #2, we describe how GFAP alone will not be the sole determinant of whether a cell is classified as the Neural G0 state. Additionally, we show that GFAP is more highly expressed in Neural G0 cells relative to astrocytes and all other cell types (**Figure 5D**). Thus, GFAP in conjunction with the other 21 genes of the GBM specific Neural G0 expression signatures does have predictive value.

The G0 state is defined as being outside the cell cycle (non-proliferative), and thus the Neural G0 expression signature should be anticorrelated with the cell cycle expression signature. However, the cell cycle and Neural G0 eigengene plot's actual pattern tells a slightly different story. They have a more perpendicular pattern of expression in single cells with very few joint expressing cells (**Figure 5G**). This perpendicular pattern indicates that the cell cycle and Neural G0 states are mutually exclusive, as would be expected. This is likely because cells in the Neural G0 state are not in the cell cycle by definition.

Our work's novelty and power are that the ccAF classifier can identify this subpopulation of quiescent neoplastic cells. As we described above, our analyses are very unlikely to be confounded by differentiated glia because these cells have been expressly removed from the data before analysis. Thus, the reviewer's interpretation that differentiated glial cells drive the associations from patient tumors in the TCGA is improbable. We stand by our interpretation of the results as they make biological sense. In **Figure 5F**, as the tumor grade increases and the tumors become more aggressive, i.e., grows and spreads more quickly, they have fewer cells in the Neural G0 state and more in the cell cycle. In **Figure 5H**, tumors with more cells in the Neural G0 state in a patient's tumor were associated with a better prognosis. Notably, the survival association was still significant when including tumor grade and IDH1/2 mutation status. This strongly suggests that the Neural G0 GBM specific marker genes describe a cell state that is independently predictive of patient survival.

We also ran the inferCNV algorithm (<https://data.humancellatlas.org/analyze/methods/infer>) to predict copy number alterations for each scRNA-seq dataset to determine whether the cells from each tumor were neoplastic. Cells within a tumor share most of the copy number alterations, so this was a useful method to determine if there was widespread contamination by terminally differentiated cells. We did not find many cells in any of the datasets that did not have a shared copy number mutational profile. We provide an example of this for Darmanis et al., 2017 in Appendix Figure S9.

The exclusion of terminally differentiated cells via marker genes is well described in the main text on Page 14 Lines 16-24, and in Appendix Figures S8A-B. Additionally, we have added an independent method for determining if single cells are neoplastic through copy number analysis. The copy number results can be found on Page 14-15 Lines 24 & 1-4, and in Appendix Figure S9.

4. Minor: It would be an advantage to the community to include an analysis of the 19 samples from Wang Cancer Discovery 2019, as this is the second largest cohort outside of Neftel 2019. I notice that Wang et al. have uploaded a list of phenotype labels to their GEO repo. At any rate it shouldn't be hard to classify them de novo based on PCA or simple markers of mesenchymal (CD44, CHI3L1) and proneural (OLIG2, DLL3) cells.

The Wang et al., Cancer Discovery 2019 dataset (GSE138794) was downloaded and put through the quality control pipeline we developed. The first oddity has to do with the sample meta-information. Based on Table S1 of the manuscript, there should be 15 GBM 10X scRNA-seq profiled samples. Which is at odds with the 19 they claim in the manuscript, and contrary to the description in the main text one of patient tumors (SF11964) is IDH mutant. In the downloaded file from NCBI GEO (GSE138794_RAW.tar) there are only 14 of the fifteen GBM samples listed in Table S1. Furthermore, the list of genes has been heavily filtered such that an intersection of the gene lists across the datasets yields zero genes. The filtering strategy is not explained anywhere and thus is not raw, untouched data as advertised on NCBI GEO. Therefore, these issues suggested something was potentially problematic with this dataset before any data analyses.

As the reviewer suggests, cell type information is now available for 9 of the 14 scRNA-seq profiled tumors. We used the InferCNV package to determine which cells were neoplastic and cross-compared that with the cell types from the meta-data. We found that the endothelial, oligodendrocyte, and astrocyte cells showed patterns of CNV mutation consistent with being neoplastic (**Figure RC1A**) relative to myeloid cells. For comparison, please see the InferCNV analysis applied to Darmanis et al., 2017 where each subpopulation was sorted out using specialized cell type markers using flow-cytometry assisted cell sorting (FACS) (**Figure RC1B**). The Darmanis et al., 2017 data shows that astrocyte, immune, neuron, oligodendrocyte, and OPC cells lack significant CNV mutations, whereas the neoplastic cells have large-scale CNV mutations (**Figure RC1B**).

These discrepancies made Wang et al., Cancer Discovery 2019 fail our quality control check. It is unfortunate but necessary for our study's integrity that we exclude these data at this point. We had hoped the inclusion of an additional 34,702 GBM tumor cells would address the need for more cells in the last submission.

Figure RC1. InferCNV QC analysis of **A.** Wang et al., Cancer Discovery 2019. **B.** Darmanis et al., Cell Reports 2017.

Reviewer #3:

The authors address all my previous concerns and the revised manuscript has significantly improved as compared to the original one.

Only a minor request to further improved clarity would be to include the various marker genes used to identify the clusters and assess the cell cycle phases of the cells in the tSNE plot. This would give the reader an idea of how good the computed cluster boundaries are.

As requested, we have generated t-SNE plots for all of the marker genes in Figure 1F and included this as Appendix Figure S2. Below is the new supplementary figure:

We have also added the following text to the manuscript on Page 6 Lines 6-10 to point readers at this new figure: "We assigned a cell cycle phase to the seven U5-hNSC clusters by analyzing the marker genes (Figure 1F & Appendix Figure S2), cyclin and CDK expression (Figure 1G &

J-K), GO term functional enrichment (Figure 1H; Dataset EV2), and enrichment of genes associated with arrest in specific cell cycle phases (Figure 1I) (Santos *et al*, 2015)."

There are still some typos.

We have run spell checkers and grammar checkers against the document, including the figure legends. Our editorial team read through the manuscript multiple times to capture any additional typos and fix them.

Thank you again for sending us your revised manuscript. We have now heard back from reviewer #2 who was asked to evaluate your revised study. As you will see below, reviewer #2 still raises substantial concerns on your work and does not support publication of the study in Molecular Systems Biology.

Specifically, reviewer #2 is not convinced that the performed revisions have adequately addressed their concerns regarding the relevance of the presented Neural GO signature. This was one of the main issues raised already after the first round of review and was also the main unresolved issue after the previous round of revision. This issue is important as it significantly impacts the broad relevance and potential clinical applications of the presented findings.

Overall, given the remaining substantial concerns and considering that we did already allow an exceptional second round of major revision, I am afraid I see no choice but to return the manuscript with the message that we cannot offer to publish it. I am sorry that the review of your work did not result in a more favorable outcome on this occasion, but I hope that you will not be discouraged from sending your work to Molecular Systems Biology in the future. I hope that you will soon find a suitable venue for publishing your work.

REFeree REPORTS

Reviewer #2:

In their revised manuscript the authors have decided, again, not to retrain their model or adapt their signature in any way. As a consequence, my assessment from the previous round also remains unchanged. The authors rebut my critiques, which were raised in the initial round, but have not altered their signature in any way and instead argue their signature is valid.

The bottom line for me is that the signature the authors have derived is comprised of genes which are very highly expressed in terminally differentiated glia, e.g. Figure 5C-E. Thus, any application of the signature will just pick up differentiated glia and their signature is therefore not useable.

Another example that the Neural GO signature is flawed, which was raised in the original review, is that it is expressed in the majority of GBM cells regardless of cell type (Fig 3K-L This is not surprising, as the core signature contains GFAP which is very broadly expressed in glia. Again, the authors' signature will not be very useful for identifying quiescent stem cells since the signature is highly expressed in the vast majority of glioma cells (including non-malignant tumor-derived glia. To me the lack of utility is obvious and I don't understand the authors' reluctance to retrain the model.

The signature they identify simply anticorrelates with Ki67 and other markers of cell-cycle progression (Figure 5G, hence the correlation with grade (Figure 5F which is defined by mitotic count, and with survival (Figure 5H.

While the underlying goal is laudable and some aspects of the approach are novel the unwillingness to retrain their model on better data or to in any way incorporate feedback into their core signature has left them with a signature that is in my opinion not useable. Since a novel signature for quiescent glioma stem cells is the most significant aspect of the manuscript I do not think these current results will be of broad utility.

We want to appeal the rejection of our paper entitled "Neural G0: a quiescent-like state found in neuroepithelial-derived cells and glioma". This manuscript's rejection is based on the criticism of Reviewer #2.

We feel that Reviewer #2's critique was inappropriate for the work presented in this manuscript. Reviewer #2's "bottom line" critique from the last review, which was the basis for rejection, was that:

"the signature the authors have derived is comprised of genes which are very highly expressed in terminally differentiated glia, e.g. Figure 5C-E. Thus, any application of the signature will just pick up differentiated glia and their signature is therefore not useable."

This reviewer seems to have fundamentally misunderstood the paper, this figure, the data, and the nature of our classifier.

- 1) Our ccAF classifier is derived from in vitro grown human neural stem cells.
- 2) The "Neural G0" category consists of 404 genes that are highly enriched in quiescent neural stem cells (Fig. 2C-D). These genes do not include ones highly expressed in terminally differentiated glial cells, such as GFAP or AQP4 (Dataset EV1).
- 3) In Fig 5 A-B, we leverage the Neural G0 classifier to identify GBM specific genes found in GBM tumor cells. These do include genes like GFAP and AQP4.
- 4) Before identifying GBM specific genes from the patient tumor single-cell datasets, we removed any cells with high expression of differentiated glia, neuronal, or immune cell marker genes and a normal CNV profile (Appendix Figure S8A showing Wang et al, 2020).
- 5) In Fig 5 C-E, we show that while markers like AQP4, GFAP, and PLP1 are indeed found in tumor-associated brain cells such as astrocytes and oligodendrocytes (as we would expect). They are also none-the-less found in the tumor cells themselves as evidenced from scRNA-seq data (Fig 5 C-E). Thus, the tumor cells have a mixture of Neural G0 and differentiated cell markers. We know this is true because we validated that these cells are neoplastic using CNV/SNP analysis of individual cells in the scRNA-seq data (and the tumor cell isolation methods used by Darmanis et al., 2017)
- 6) The reviewer has misunderstood the analysis of TCGA tumor data presented in Fig 5F-H. The reviewer asserts: "The signature they identify simply anticorrelates with Ki67 and other markers of cell-cycle progression (Figure 5G), hence the correlation with grade (Figure 5F) which is defined by mitotic count, and with survival (Figure 5H)." There is a significant anticorrelation between the Neural G0 expression signature and the cell cycle expression signature ($R=-0.48$). However, a correlation coefficient of -0.48 only accounts for 23% of Neural G0 expression signature variance. Indeed, as we described to Reviewer #2 when we include both the Neural G0 and cell cycle expression signatures in a Cox proportional hazards regression model both terms are significantly predictive. That Neural G0 is a significant predictor when the cell cycle is added to the model demonstrates that Neural G0 expression has clinical impact and is not "simply anticorrelation with Ki67 and other markers of cell-cycle progression".
- 7) Further, the reviewer asserts: "Another example that the Neural GO signature is flawed, which was raised in the original review, is that it is expressed in the majority of GBM cells regardless of cell type (Fig 3K-L)." However, again the reviewer seems to be confused. We interpret this as the reviewer implying that the ccAF classifier is simply detecting non-tumor cells that express AQP4 or GFAP. However, the ccAF classifier does not contain these genes. Again, the reviewer continues:
"It is not surprising, as the core signature contains GFAP which is very broadly expressed in glia. Again, the authors' signature will not be very useful for identifying quiescent stem cells since the

signature is highly expressed in the vast majority of glioma cells (including non-malignant tumor-derived glia). To me the lack of utility is obvious and I don't understand the authors' reluctance to retrain the model."

Please note that one of our co-authors Dr. Anoop Patel is a leading expert in glioma biology, single-cell genomic applications for glioma, and is a brain tumor surgeon. Dr. Patel has been instrumental in advising on applications of ccAF to glioma data sets and the possible implications for identifying candidate G0 populations in brain tumors.

In summary, we feel that Reviewer #2 has misconstrued our results and reinterpreted them to fit their own conclusions from our use of our ccAF classifier. Thereby, we request that the MSB editors reconsider the rejection based on the above points regarding Reviewer #2's comments.

Thank you for your message asking us to reconsider our decision on your manuscript MSB-20-9522RR. I have now had the chance to evaluate the points raised in your appeal letter and I have also consulted with reviewer #3. As you will see below, reviewer #3 thinks that some of the remaining issues raised by reviewer #2 seem to stem from misunderstandings and do not require additional revisions. Reviewer #3 thinks that some further discussion would address the issue of "the correlative nature of the Neural G0 signature and a cycling-activity-based (KI67) signature".

As such, we have decided to invite you to perform a final minor revision.

On a more editorial level we would ask you to address the following.

REFEREE REPORTS

Reviewer #3:

Concerning the author's response to the comments made by Reviewer #2 I feel that points 1-5 have been appropriately addressed by the authors and that in particular, Reviewer #2's claims concerning the signature's preponderance to terminally differentiated glia are misguided. Indeed, Reviewer #2 appears to conflate the Neural G0- and GBM-specific-signatures as indicated in points 2 & 3. Hence Reviewer #2's assertion of the signature "just pick[ing] up differentiated glia" is misplaced, as the GFAP-containing signature is intended for GBM-specific applications. The authors address this fact in point 5, referring to Figures 5 C-E, and indeed: scoring for the GBM signature in the combined Darmanis dataset would serve to alleviate my concerns regarding its favoring terminally differentiated glia, as this dataset contains both healthy neurons, oligodendrocytes, and astrocytes, as well as malignant GBM cells.

On the other hand, while I feel that Reviewer #2's concerns regarding the correlative nature of the Neural G0 signature and a cycling-activity-based (ie KI67) signature are justified, and indeed the authors are able to demonstrate the efficacy of cCAF mainly through its non-overlap with a cycling signature (cf. HeLa cell & FUCCI experiments), the significance of the Neural G0 term in the Cox regression model when cell cycle is accounted for demonstrates that it captures information beyond the absence of cycling activity. Hence I feel this model and requisite discussion should be included in the final manuscript.

Points raised by the editor:

Thank you for your message asking us to reconsider our decision on your manuscript MSB-20-9522RR. I have now had the chance to evaluate the points raised in your appeal letter and I have also consulted with reviewer #3. As you will see below, reviewer #3 thinks that some of the remaining issues raised by reviewer #2 seem to stem from misunderstandings and do not require additional revisions. Reviewer #3 thinks that some further discussion would address the issue of "the correlative nature of the Neural G0 signature and a cycling-activity-based (KI67) signature".

As such, we have decided to invite you to perform a final minor revision.

We thank the journal, editor, and reviewer for taking the time to reconsider the decision. We appreciate the opportunity to submit a final minor revision and feel that the request from Reviewer #3 to further discuss the Cox PH model described in the responses was reasonable and makes the manuscript stronger. We have addressed that in the reviewer comment below.

Reviewer #3:

Concerning the author's response to the comments made by Reviewer #2 I feel that points 1-5 have been appropriately addressed by the authors and that in particular, Reviewer #2's claims concerning the signature's preponderance to terminally differentiated glia are misguided. Indeed, Reviewer #2 appears to conflate the Neural G0- and GBM-specific-signatures as indicated in points 2 & 3. Hence Reviewer #2's assertion of the signature "just pick[ing] up differentiated glia" is misplaced, as the GFAP-containing signature is intended for GBM-specific applications. The authors address this fact in point 5, referring to Figures 5 C-E, and indeed: scoring for the GBM signature in the combined Darmanis dataset would serve to alleviate my concerns regarding its favoring terminally differentiated glia, as this dataset contains both healthy neurons, oligodendrocytes, and astrocytes, as well as malignant GBM cells.

On the other hand, while I feel that Reviewer #2's concerns regarding the correlative nature of the Neural G0 signature and a cycling-activity-based (ie KI67) signature are justified, and indeed the authors are able to demonstrate the efficacy of ccAF mainly through it's non-overlap with a cycling signature (cf. HeLa cell & FUCCI experiments), the significance of the Neural G0 term in the Cox regression model when cell cycle is accounted for demonstrates that it captures information beyond the absence of cycling activity. Hence I feel this model and requisite discussion should be included in the final manuscript.

We thank Reviewer #3 for their thoughtful review of the correspondence. The reviewer makes an excellent point that we should include the Cox proportional hazards regression model that shows that Neural G0 eigengene captures information beyond simply being anticorrelated with the cell cycle. We have integrated the Cox model with the Neural G0 and cell cycle eigengenes into the results (Page 19 line 24 – Page 20 line 3):

Additionally, the Neural G0 eigengene was significantly associated with patient survival when the cell cycle eigengene is included in the model (Cox PH coef. = -0.14; and p-value = 9.8×10^{-9}). The Neural G0 has an independent effect beyond the cell cycle effects and therefore the Neural G0 state is not simply the opposite of an actively cycling cell state.

We also added a small section to the discussion as requested (Page 29 line 2-6):

Additionally, the Neural G0 state was shown to account for survival variance that is independent from active cell cycling, which means that the Neural G0 state is not simply the antithesis of

active cell cycle states. Instead, the Neural G0 state has novel biological mechanisms regulating flow into and out of the G0 state that go beyond the biology of the active cell cycle.

Thank you again for sending us your revised manuscript and for performing the requested edits. We are now satisfied with the modifications made and I am pleased to inform you that your paper has been accepted for publication.

Corresponding Author Name: Christopher Plaisier and Patrick Paddison

Manuscript Number: MSB-20-9522